# Towards Understanding Linear Value Decomposition in Cooperative Multi-Agent Q-Learning

## Abstract

Value decomposition is a popular and promising approach to scaling up multi-agent reinforcement learning in cooperative settings. However, the theoretical understanding of such methods is limited. In this paper, we introduce a variant of the fitted Q-iteration framework for analyzing multi-agent Q-learning with value decomposition. Based on this framework, we derive a closed-form solution to the empirical Bellman error minimization with linear value decomposition. With this novel solution, we further reveal two interesting insights: i) linear value decomposition implicitly implements a classical multi-agent credit assignment called *counterfactual difference rewards*; and ii) On-policy data distribution or richer $Q$ function classes can improve the training stability of multi-agent Q-learning. In the empirical study, our experiments demonstrate the realizability of our theoretical closed-form formulation and implications in the didactic examples and a broad set of StarCraft II unit micromanagement tasks, respectively.

## 1 Introduction

Cooperative multi-agent reinforcement learning (MARL) has great promise for addressing coordination problems in a variety of applications, such as robotic systems (Hüttenrauch et al., 2017), autonomous cars (Cao et al., 2012), and sensor networks (Zhang & Lesser, 2011). Such complex tasks often require MARL to learn decentralized policies for agents to jointly optimize a global cumulative reward signal, and post a number of challenges, including multi-agent credit assignment (Wolpert & Tumer, 2002; Nguyen et al., 2018), non-stationarity (Zhang & Lesser, 2010; Song et al., 2019), and scalability (Zhang & Lesser, 2011; Panait & Luke, 2005). Recently, by leveraging the strength of deep learning techniques, cooperative MARL has made a series of great progress (Sunehag et al., 2018; Baker et al., 2020; Wang et al., 2020b;a), particularly in value-based methods that demonstrate state-of-the-art performance on challenging tasks such as StarCraft unit micromanagement (Samvelyan et al., 2019). Sunehag et al. (2018) proposed a popular approach called value-decomposition network (VDN) based on the paradigm of *centralized training with decentralized execution* (CTDE; Foerster et al., 2016). VDN learns a centralized but factorizable joint value function $Q_{\text{tot}}$, represented as the summation of individual value functions $Q_i$. During the execution, decentralized policies can be easily derived for each agent $i$ by greedily selecting actions with respect to its local value function $Q_i$. By utilizing this decomposition structure, an implicit multi-agent credit assignment is realized because $Q_i$ learned by neural network backpropagation from the total temporal-difference error on the single global reward signal, rather than on a local reward signal specific to agent $i$. This decomposition technique significantly improves the scalability of multi-agent Q-learning algorithms and fosters a series of subsequent works, including QMIX (Rashid et al., 2018), QTRAN (Son et al., 2019), and QPLEX (Wang et al., 2020a).

In spite of the empirical success in a broad class of tasks, multi-agent Q-learning with linear value decomposition has not been theoretically well-understood. Because of its limited representation complexity, the standard Bellman update is not a closed operator in the joint action-value function class with linear value decomposition. The approximation error induced by this incompleteness is known as *inherent Bellman error* (Munos & Szepesvári, 2008), which usually deviates Q-learning to an unexpected behavior. To develop a deeper understanding of learning with value decomposition, this paper introduces a multi-agent variant of the popular Fitted Q-Iteration (FQI; Ernst et al., 2005;

Levine et al., 2020) framework and derives a closed-form solution to its empirical Bellman error minimization. To the best of our knowledge, it is the first theoretical analysis that characterizes the underlying mechanism of linear value decomposition in cooperative multi-agent Q-learning, which can serve as a powerful toolkit to establish follow-up profound theories and explore potential insights from different perspectives in this popular value decomposition structure.

By utilizing this novel closed-form solution, this paper formally reveals two interesting insights: 1) Learning linear value decomposition implicitly implements a classical multi-agent credit assignment method called *counterfactual difference rewards* (Wolpert & Tumer, 2002), which draws a connection with COMA (Foerster et al., 2018), a multi-agent policy-gradient method. 2) Multi-agent Q-learning with linear value decomposition potentially suffers from the risk of unbounded divergence from arbitrary initialization. On-policy data distribution or richer $Q$ function classes can provide local or global convergence guarantees for multi-agent Q-learning, respectively.

Finally, we set up an extensive set of experiments to demonstrate the realizability of our theoretical implications. Besides the FQI framework, we also consider deep-learning-based implementations of different multi-agent value decomposition structures. Through didactic examples and the StarCraft II benchmark, we design several experiments to illustrate the consistency of our closed-form formulation with the empirical results, and that online data distribution and richer $Q$ function classes can significantly alleviate the limitations of VDN on the offline training process (Levine et al., 2020).

## 2 RELATED WORK

Deep Q-learning algorithms that use neural networks as function approximators have shown great promise in solving complicated decision-making problems (Mnih et al., 2015). One of the core component of such methods is iterative Bellman error minimization, which can be modelled by a classical framework called Fitted Q-Iteration (FQI; Ernst et al., 2005). FQI utilizes a specific $Q$ function class to iteratively optimize empirical Bellman error on a dataset $D$. Great efforts have been made towards theoretically characterizing the behavior of FQI with finite samples and imperfect function classes (Munos & Szepesvári, 2008; Farahmand et al., 2010; Chen & Jiang, 2019). From an empirical perspective, there is also a growing trend to adopt FQI for empirical analysis of deep offline Q-learning algorithms (Fu et al., 2019; Levine et al., 2020). In MARL, the joint $Q$ function class grows exponentially with the number of agents, leading many algorithms (Sunehag et al., 2018; Rashid et al., 2018) to utilize different value decomposition structures with limited expressiveness to improve scalability. In this paper, we extend FQI to a multi-agent variant as our grounding theoretical framework for analyzing cooperative multi-agent Q-learning with linear value decomposition.

To achieve superior effectiveness and scalability in multi-agent settings, centralized training with decentralized executing (CTDE) has become a popular MARL paradigm (Oliehoek et al., 2008; Kraemer & Banerjee, 2016). *Individual-Global-Max* (IGM) principle (Son et al., 2019) is a critical concept for value-based CTDE (Mahajan et al., 2019), that ensures the consistency between joint and local greedy action selections and enables effective performance in both training and execution phases. VDN (Sunehag et al., 2018) utilizes linear value decomposition to satisfy a sufficient condition of IGM. The simple additivity structure of VDN has achieved excellent scalability and inspired many follow-up methods. QMIX (Rashid et al., 2018) proposes a monotonic $Q$ network structure to improve the expressiveness of the factorized function class. QTRAN (Son et al., 2019) tries to realize the entire IGM function class, but its method is computationally intractable and requires two extra soft regularizations to approximate IGM (which actually loses the IGM guarantee). QPLEX (Wang et al., 2020a) encodes the IGM principle into the $Q$ network architecture and realizes a complete IGM function class, but it may also have potential limitations in scalability. Based on the advantages of VDN's simplicity and scalability, linear value decomposition becomes very popular in MARL (Son et al., 2019; Wang et al., 2020a;d). This paper focuses on the theoretical and empirical understanding of multi-agent Q-learning with linear value decomposition to explore its underlying implications.

## 3 NOTATIONS AND PRELIMINARIES

### 3.1 MULTI-AGENT MARKOV DECISION PROCESS (MMDP)

To support theoretical analysis on multi-agent Q-learning, we adopt the framework of MMDP (Boutilier, 1996), a special case of Dec-POMDP (Oliehoek et al., 2016), to model fully cooperative

multi-agent decision-making tasks. MMDP is defined as a tuple $\mathcal{M} = \langle \mathcal{N}, \mathcal{S}, \mathcal{A}, P, r, \gamma \rangle$. $\mathcal{N} \equiv \{1, \ldots, n\}$ is a finite set of agents. $\mathcal{S}$ is a finite set of global states. $\mathcal{A}$ denotes the action space for an individual agent. The joint action $\mathbf{a} \in \mathbf{A} \equiv \mathcal{A}^n$ is a collection of individual actions $[a_i]_{i=1}^n$. At each timestep $t$, a selected joint action $\mathbf{a}_t$ results in a transition $s_{t+1} \sim P(\cdot|s_t, \mathbf{a}_t)$ and a global reward signal $r(s_t, \mathbf{a}_t)$. $\gamma \in [0, 1)$ is a discount factor. The goal for MARL is to construct a joint policy $\boldsymbol{\pi} = \langle \pi_1, \ldots, \pi_n \rangle$ maximizing expected discounted rewards $V^{\boldsymbol{\pi}}(s) = \mathbb{E}\left[\sum_{t=0}^{\infty} \gamma^t r(s_t, \boldsymbol{\pi}(s_t))|s_0 = s\right]$, where $\pi_i : \mathcal{S} \mapsto \mathcal{A}$ denotes an individual policy of agent $i$. The corresponding action-value function is denoted as $Q^{\boldsymbol{\pi}}(s, \mathbf{a}) = r(s, \mathbf{a}) + \gamma \mathbb{E}_{s' \sim P(\cdot|s,\mathbf{a})}[V^{\boldsymbol{\pi}}(s')]$. We use $Q^*$ and $V^*$ to denote the action-value function and the state-value function corresponding to the optimal policy $\boldsymbol{\pi}^*$, respectively.

Dec-POMDP (Oliehoek et al., 2016) is a generalized model of MMDP with the consideration of partial observability. In Dec-POMDPs, each agent can only access to its local observations rather than the full information of global states. As infinite-horizon Dec-POMDPs is undecidable in general (Madani et al., 1999), this paper focuses theoretical analyses on settings with full observability. In practice, partial observability is not a hard constraint. A Dec-POMDP can be transformed to an MMDP when communication is available. Many prior efforts have been made to construct efficient communication protocols for exchanging information among agents (Foerster et al., 2016; Das et al., 2019; Wang et al., 2020c). By constructing belief states through extended observation scopes, these methods can approximately transform learning problems in Dec-POMDPs to that in MMDPs. From this perspective, we consider MMDP as a simplification of notations to make the underlying insights more accessible.

## 3.2 Centralized Training with Decentralized Execution (CTDE)

Most deep multi-agent Q-learning algorithms with value decomposition adopt the paradigm of centralized training with decentralized execution (Foerster et al., 2016). In the training phase, the centralized trainer can access all global information, including global states, shared global rewards, agents' polices, and value functions. In the decentralized execution phase, every agent makes individual decisions based on its local observations. Note that this paper considers MMDP as a simplified setting which rules out the concerns of partial observability. Thus our notations do not distinguish the concepts of states and observations. *Individual-Global-Max* (IGM) (Son et al., 2019) is a common principle to realize effective decentralized policy execution. It enforces the action selection consistency between the global joint action-value $Q_{\text{tot}}$ and individual action-values $[Q_i]_{i=1}^n$, which are specified as follows:

$$\forall s \in \mathcal{S}, \ \arg\max_{\mathbf{a} \in \mathbf{A}} Q_{\text{tot}}(s, \mathbf{a}) = \left\langle \arg\max_{a_1 \in \mathcal{A}} Q_1(s, a_1), \ldots, \arg\max_{a_n \in \mathcal{A}} Q_n(s, a_n) \right\rangle. \tag{1}$$

As stated in Eq. (2), the additivity constraint adopted by VDN (Sunehag et al., 2018) is a sufficient condition for the IGM constraint stated in Eq. (1). However, this linear decomposition structure is not a necessary condition and induces a limited joint action-value function class because the linear number of individual functions cannot represent a joint action-value function class, which is exponential with the number of agents.

$$\textbf{(Additivity)} \quad Q_{\text{tot}}(s, \mathbf{a}) = \sum_{i=1}^{n} Q_i(s, a_i). \tag{2}$$

## 3.3 Fitted Q-Iteration (FQI) for Multi-Agent Q-Learning

For multi-agent Q-learning with value decomposition, we use $Q_{\text{tot}}$ to denote the global but factorized value function, which can be factorized as a function of individual value functions $[Q_i]_{i=1}^n$. In other words, we can use $[Q_i]_{i=1}^n$ to represent $Q_{\text{tot}}$. For brevity, we overload $Q$ to denote both of them. In the MMDP settings, the shared reward signal can only supervise the training of the joint value function $Q_{\text{tot}}$, which requires us to modify the notation of *Bellman optimality operator* $\mathcal{T}$ as follows:

$$(\mathcal{T}Q)_{\text{tot}}(s, \mathbf{a}) = r(s, \mathbf{a}) + \gamma \mathbb{E}_{s' \sim P(s'|s,\mathbf{a})}\left[\max_{\mathbf{a}' \in \mathbf{A}} Q_{\text{tot}}(s', \mathbf{a}')\right]. \tag{3}$$

Fitted Q-iteration (FQI) (Ernst et al., 2005) provides a unified framework which extends the above operator to solve high-dimensional tasks using function approximation. It follows an iterative optimization framework based on a given dataset $D = \{(s, \mathbf{a}, r, s')\}$,

$$Q^{(t+1)} \leftarrow \underset{Q \in \mathcal{Q}}{\arg\min} \underset{(s,\mathbf{a},r,s') \sim D}{\mathbb{E}} \left[ \left( r + \gamma \max_{\mathbf{a}' \in \mathbf{A}} Q_{\text{tot}}^{(t)}(s', \mathbf{a}') - Q_{\text{tot}}(s, \mathbf{a}) \right)^2 \right], \qquad (4)$$

where an initial solution $Q^{(0)}$ is selected arbitrarily from a function class $\mathcal{Q}$. By constructing a specific function class $\mathcal{Q}$ that only contains instances satisfying the IGM condition stated in Eq. (1) (Sunehag et al., 2018; Rashid et al., 2018), the centralized training procedure in Eq. (4) will naturally produce suitable individual values $[Q_i]_{i=1}^n$, from which individual policies can be derived for decentralized execution.

## 4 MULTI-AGENT Q-LEARNING WITH LINEAR VALUE DECOMPOSITION

In the literature of deep MARL, constructing a specific value function class $\mathcal{Q}$ satisfying the IGM condition is a critical step to realize the paradigm centralized training with decentralized execution. Linear value decomposition proposed by VDN (Sunehag et al., 2018) is a simple yet effective method to implement this paradigm. In this section, we provide theoretical analysis towards a deeper understanding of this popular decomposition structure. Our result is based on a multi-agent variant of fitted Q-iteration (FQI) with linear value decomposition, named FQI-LVD. We derive the closed-form update rule of FQI-LVD, and then reveal the underlying credit assignment mechanism realized by linear value decomposition learning.

### 4.1 MULTI-AGENT FITTED Q-ITERATION WITH LINEAR VALUE DECOMPOSITION (FQI-LVD)

To provide a clear perspective on the effects of linear value decomposition, we make an additional assumption to simplify the notations and facilitate the analysis.

**Assumption 1** (Adequate and Factorizable Dataset). *The dataset $D$ contains all applicable state-action pairs $(s, \mathbf{a})$ whose empirical probability is factorizable with respect to individual behaviors of multiple agents. Formally, let $p_D(\mathbf{a}|s)$ denote the empirical probability of joint action $\mathbf{a}$ executed on state $s$, which can be factorized to the production of individual components,*

$$p_D(\mathbf{a}|s) = \prod_{i \in \mathcal{N}} p_D(a_i|s), \quad \sum_{a_i \in \mathcal{A}} p_D(a_i|s) = 1, \quad p_D(a_i|s) > 0, \qquad (5)$$

*where $p_D(a_i|s)$ denotes the empirical probability of the event that agent $i$ executes $a_i$ on state $s$.*

Assumption 1 is based on the fact that an adequate dataset is necessary for FQI algorithms to find a feasible solution (Farahmand et al., 2010; Chen & Jiang, 2019). In practice, the property of factorizable data distribution can be directly induced by a decentralized data collection procedure. When agents perform fully decentralized execution, the empirical probability of an event $(s, \mathbf{a})$ in the collected dataset $D$ is naturally factorized.

Now we define FQI with linear value decomposition as follows.

**Definition 1** (FQI-LVD). *Given a dataset $D$, FQI-LVD specifies the action-value function class*

$$\mathcal{Q}^{LVD} = \left\{ Q \mid Q_{tot}(\cdot, \mathbf{a}) = \sum_{i=1}^n Q_i(\cdot, a_i), \forall \mathbf{a} \in \mathbf{A} \text{ and } \left[ \forall Q_i \in \mathbb{R}^{|\mathcal{S}||\mathcal{A}|} \right]_{i=1}^n \right\} \qquad (6)$$

*and induces the empirical Bellman operator $\mathcal{T}_D^{LVD}$:*

$$Q^{(t+1)} \leftarrow \mathcal{T}_D^{LVD} Q^{(t)} \equiv \underset{Q \in \mathcal{Q}^{LVD}}{\arg\min} \sum_{(s,\mathbf{a},s') \in \mathcal{S} \times \mathbf{A} \times \mathcal{S}} p_D(\mathbf{a}, s'|s) \left( \hat{y}^{(t)}(s, \mathbf{a}, s') - \sum_{i=1}^n Q_i(s, a_i) \right)^2$$

$$= \underset{Q \in \mathcal{Q}^{LVD}}{\arg\min} \sum_{(s,\mathbf{a}) \in \mathcal{S} \times \mathbf{A}} p_D(\mathbf{a}|s) \left( y^{(t)}(s, \mathbf{a}) - \sum_{i=1}^n Q_i(s, a_i) \right)^2, \qquad (7)$$

*where $\hat{y}^{(t)}(s, \mathbf{a}, s') = r(s, \mathbf{a}) + \gamma \max_{\mathbf{a}'} Q_{tot}^{(t)}(s', \mathbf{a}')$ denotes the sample-based regression target. $y^{(t)}(s, \mathbf{a}) = (\mathcal{T}Q^{(t)})_{tot}(s, \mathbf{a}) = r(s, \mathbf{a}) + \gamma \mathbb{E}_{s' \sim P(\cdot|s,\mathbf{a})} \left[ \max_{\mathbf{a}'} Q_{tot}^{(t)}(s', \mathbf{a}') \right]$ denotes the ground-truth target value derived by Bellman optimality operator. $p_D(\mathbf{a}, s'|s) = p_D(\mathbf{a}|s)P(s'|s, \mathbf{a})$ denotes the empirical probability of the event that agents execute joint action $\mathbf{a}$ on state $s$ and transit to $s'$. $Q_{tot}$ and $[Q_i]_{i=1}^n$ refer to the discussion of CTDE defined in Section 3.3.*

The proof of Eq. (7) is deferred to Lemma 1 in Appendix A. Value-decomposition network (VDN) (Sunehag et al., 2018) provides an implementation of FQI-LVD, in which individual value functions $[Q_i]_{i=1}^n$ are parameterized by deep neural networks, and the joint value function $Q_{\text{tot}}$ can be simply formed by their summation.

## 4.2 Implicit Credit Assignment in Linear Value Decomposition

In the formulation of FQI-LVD, the empirical Bellman error minimization in Eq. (7) can be regarded as a weighted linear least-squares problem, which contains $n|\mathcal{S}||\mathcal{A}|$ variables to form individual value functions $[Q_i]_{i=1}^n$ and $|\mathcal{S}||\mathcal{A}|^n$ data points corresponding to all entries of the regression target $y^{(t)}(s, \mathbf{a})$. To solve this least-squares problem, we derive a closed-form solution stated in Theorem 1, which can be verified through *Moore-Penrose inverse* (Moore, 1920) for weighted linear regression analysis. Proofs for all theorems, lemmas, and propositions in this paper are deferred to Appendix.

**Theorem 1.** *Let $Q^{(t+1)} = \mathcal{T}_D^{LVD} Q^{(t)}$ denote a single iteration of the empirical Bellman operator. Then $\forall i \in \mathcal{N}, \forall (s, \mathbf{a}) \in \mathcal{S} \times \mathbf{A}$, the individual action-value function $Q_i^{(t+1)}(s, a_i) =*

$$\underbrace{\mathbb{E}_{a'_{-i} \sim p_D(\cdot|s)} \left[ y^{(t)} \left( s, a_i \oplus a'_{-i} \right) \right]}_{\text{evaluation of the individual action } a_i} - \frac{n-1}{n} \underbrace{\mathbb{E}_{\mathbf{a}' \sim p_D(\cdot|s)} \left[ y^{(t)} \left( s, \mathbf{a}' \right) \right]}_{\text{counterfactual baseline}} + w_i(s), \tag{8}$$

*where we denote $a_i \oplus a'_{-i} = \langle a'_1, \ldots, a'_{i-1}, a_i, a'_{i+1}, \ldots, a'_n \rangle$. $a'_{-i}$ denotes the action of all agents except for agent $i$. The residue term $\mathbf{w} \equiv [w_i]_{i=1}^n$ is an arbitrary vector satisfying $\forall s, \sum_{i=1}^n w_i(s) = 0$.*

As shown in Theorem 1, the local action-value function $Q_i^{(t+1)}$ consists of three terms. The first term is the expectation of one-step TD target value over the actions of other agents, which evaluates the expected return of executing an individual action $a_i$. The second term is the expectation of one-step target TD values over all joint actions, which can be regarded as a baseline function evaluating the average performance. The arbitrary vector $\mathbf{w}$ indicates the entire valid individual action-value function space. We can ignore this term because $\mathbf{w}$ does not affect the local action selection of each agent and will be eliminated in the summation operator of linear value decomposition (see Eq. (2)), which indicates that joint action-value $Q_{\text{tot}}^{(t+1)} = \sum_i Q_i^{(t+1)}$ has a unique closed-form solution. We compare the theoretical analysis of FQI-LVD with the empirical results of VDN to demonstrate and verify the accuracy of our closed-form updating rule (see Eq. (8)) in Section 6.1.

Note that, if we regard the empirical probability $p_D(\mathbf{a}|s)$ within the dataset $D$ as a *default policy*, the first term of Eq. (8) is the expected value of an individual action $a_i$, and the second term is the expected value of the default policy, which is considered as the *counterfactual baseline*. Their difference corresponds to a credit assignment mechanism called *counterfactual difference rewards*, which has been used by counterfactual multi-agent policy gradient (COMA) (Foerster et al., 2018).

**Implication 1.** *As shown in Eq. (8), linear value decomposition implicitly implements a counterfactual credit assignment mechanism, which is similar to what is used by COMA.*

Compared to COMA, this implicit credit assignment is naturally served by empirical Bellman error minimization through linear value decomposition, which is much more scalable. The extra importance weight $(n-1)/n$ brings our derived credit assignment to be more consistent and meaningful in the sense that all global rewards should be assigned to agents. Consider a simple case where all joint actions generate the same reward signals, Eq. (8) will assign $1/n$ unit of rewards to each agent, but COMA will assign 0. This gap will gradually close when $n$ becomes sufficiently large.

## 5 Improving the Learning Stability of Value Decomposition

In the previous section, we have derived the closed-form update rule of FQI-LVD, which reveals the underlying credit assignment mechanism of linear value decomposition structure. This derivation also enables us to investigate more algorithmic functionalities of linear value decomposition in multi-agent Q-learning. Although linear value decomposition holds superior scalability in multi-agent settings, we find that FQI-LVD has the potential risk of unbounded divergence from arbitrary initialization. To improve the stability of linear value decomposition training, we theoretically demonstrate that on-policy data distribution or richer $Q$ function classes can provide some convergence guarantees. Moreover, we also utilize a concrete MMDP example to visualize our implications.

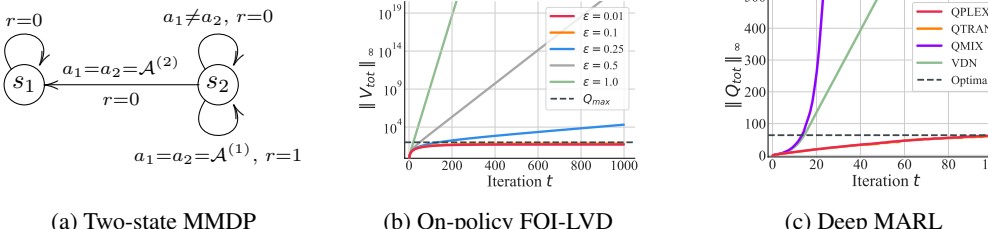

(a) Two-state MMDP      (b) On-policy FQI-LVD      (c) Deep MARL

Figure 1: (a) An MMDP where FQI-LVD will diverge to infinity when $\gamma \in \left(\frac{4}{5}, 1\right)$. $r$ is a shorthand for $r(s, \mathbf{a})$ and the action space for each agent $\mathcal{A} \equiv \left\{\mathcal{A}^{(1)}, \ldots, \mathcal{A}^{(|\mathcal{A}|)}\right\}$. (b) The learning curves of $\|Q_{\text{tot}}\|_\infty$ of on-policy FQI-LVD on the given MMDP where the dataset is generated by different choices of hyper-parameters $\epsilon$ for $\epsilon$-greedy. (c) The learning curves of $\|Q_{\text{tot}}\|_\infty$ while running several deep multi-agent Q-learning algorithms.

## 5.1 UNBOUNDED DIVERGENCE IN OFFLINE TRAINING

We will provide an analysis of the convergence of FQI-LVD with offline training on a dataset $D$.

**Proposition 1.** *The empirical Bellman operator $\mathcal{T}_D^{LVD}$ is not a $\gamma$-contraction, i.e., the following important property of the standard Bellman optimality operator $\mathcal{T}$ does not hold for $\mathcal{T}_D^{LVD}$ anymore.*

$$\forall Q_{tot}, Q'_{tot} \in \mathcal{Q}, \quad \|\mathcal{T}Q_{tot} - \mathcal{T}Q'_{tot}\|_\infty \leq \gamma \|Q_{tot} - Q'_{tot}\|_\infty \tag{9}$$

For the standard *Bellman optimality operator* $\mathcal{T}$ (Sutton & Barto, 2018), $\gamma$-contraction is critical to derive the theoretical guarantee. In the context of FQI-LVD, the additivity constraint limits the joint action-value function class that it can express, which deviates the empirical Bellman operator $\mathcal{T}_D^{LVD}$ from the original *Bellman optimality operator* $\mathcal{T}$ (see Theorem 1). This deviation is induced by the negative importance weight $(n-1)/n$ stated in Eq. (8) and is also known as *inherent Bellman error* (Munos & Szepesvári, 2008), which corrupts a broad set of stability properties, including $\gamma$-contraction.

To serve a concrete example, we construct a simple MMDP with two agents, two global states, and two actions (see Figure 1a). The optimal policy of this MMDP is simply executing the action $\mathcal{A}^{(1)}$ at state $s_2$, which is the only way for two agents to obtain a positive reward. The learning curve of $\epsilon = 1.0$ (green one) in Figure 1b refers to an offline setting with uniform data distribution, in which an unbounded divergence can be observed as depicted by the following proposition.

**Proposition 2.** *There exist MMDPs such that, when using uniform data distribution, the value function of FQI-LVD diverges to infinity from an arbitrary initialization $Q^{(0)}$.*

Note that the unbounded divergence discussed in Proposition 2 would happen to an arbitrary initialization $Q^{(0)}$. To provide an implication for practical scenarios, we also investigate the performance of several deep multi-agent Q-learning algorithms in this MMDP. As shown in Figure 1c, VDN (Sunehag et al., 2018), a deep-learning-based implementation of FQI-LVD, results in unbounded divergence. We postpone the discussion of other deep-learning-based algorithms to the next subsection.

## 5.2 LOCAL AND GLOBAL CONVERGENCE IMPROVEMENTS

To improve the training stability of FQI-LVD, we investigate methods to enable local and global convergence of value decomposition learning, respectively.

**Local Convergence Improvement.** As shown in Theorem 1, the choice of training data distribution affects the output of the empirical Bellman operator $\mathcal{T}_D^{\text{LVD}}$. We find that FQI-LVD has a local convergence property in an on-policy mode, i.e., the dataset $D$ is accumulated by running an $\epsilon$-greedy policy (Mnih et al., 2015). Here we include an informal statement of local stability of FQI-LVD and defer the precise version, its proof, and the algorithm box of on-policy FQI-LVD to Appendix C.1.

**Theorem 2** (Informal). *On-policy FQI-LVD will locally converge to the optimal policy and have a fixed point value function when the hyper-parameter $\epsilon$ is sufficiently small.*

Theorem 2 indicates that multi-agent Q-learning with linear value decomposition has a convergent region, where the value function induces optimal actions. By combining this local stability with

Brouwer's fixed-point theorem (Brouwer, 1911), we can further verify the existence of a fixed-point solution for the on-policy Bellman operator $\mathcal{T}_{D_t}^{\text{LVD}}$. Figure 1b visualizes the performance of on-policy FQI-LVD with different values of the hyper-parameter $\epsilon$. With a smaller $\epsilon$ (such as 0.1 or 0.01), on-policy FQI-LVD demonstrates numerical stability, and their corresponding collected datasets are closer to on-policy data distribution.

**Global Convergence Improvement.** Linear value decomposition structure limits the joint action-value function class $\mathcal{Q}^{\text{LVD}}$, which is the origin of the deviation of the empirical Bellman operator $\mathcal{T}_D^{\text{LVD}}$, discussed in Proposition 1. Another way to improve training stability is to enrich the expressiveness of value decomposition. We consider a multi-agent fitted Q-iteration (FQI) with a full action-value function class derived from IGM, named FQI-IGM, whose action-value function class is as follows:

$$\mathcal{Q}^{\text{IGM}} = \left\{ Q \,\middle|\, Q_{\text{tot}} \in \mathbb{R}^{|\mathcal{S}||\mathcal{A}|^n} \text{ and } \left[ \forall Q_i \in \mathbb{R}^{|\mathcal{S}||\mathcal{A}|} \right]_{i=1}^n \text{ with Eq. (1) is satisfied} \right\}. \tag{10}$$

Note that $\mathcal{Q}^{\text{LVD}} \subset \mathcal{Q}^{\text{IGM}}$ indicates that linear decomposition structure stated in Eq. (2) is a sufficient condition for the IGM constraint. The formal definition of FQI-IGM is deferred to Appendix C.2 and its global convergence property is established by the following theorem.

**Theorem 3.** *FQI-IGM will globally converge to the optimal value function.*

Theorem 3 relies on a fact that $\mathcal{Q}^{\text{IGM}}$ is complete in MMDP settings, i.e., *inherent Bellman errors* discussed in Proposition 1 can reach zero and its empirical Bellman operator $\mathcal{T}_D^{\text{IGM}}$ is a $\gamma$-contraction. Using universal function approximation of neural networks, QPLEX (Wang et al., 2020a), a deep-learning-based implementation of FQI-IGM, theoretically realizes the complete IGM function class. QTRAN (Son et al., 2019) is an approximate implementation of FQI-IGM which uses soft penalties to realize IGM constraints. As the basis of comparison, VDN (Sunehag et al., 2018) is the deep-learning-based implementation of FQI-LVD. An intermediate version, QMIX (Rashid et al., 2018), establishes a non-linear monotonic mapping between local and global value functions. The value function class of QMIX can be summarized as follows:

$$\mathcal{Q}^{\text{QMIX}} = \left\{ Q \,\middle|\, Q_{\text{tot}}(s, \mathbf{a}) = f(s, Q_1(s, a_1), \ldots, Q_n(s, a_n)) \text{ and } f(s, \cdot) \text{ is monotonic} \right\}, \tag{11}$$

which is known to underrepresent the IGM function class since the monotonic correspondence is not necessary for the IGM constraint stated in Eq. (1) (Mahajan et al., 2019). Formally, $\mathcal{Q}^{\text{LVD}} \subset \mathcal{Q}^{\text{QMIX}} \subset \mathcal{Q}^{\text{IGM}}$ is a sequence of strict inclusion relations.

As shown in Figure 1c, QPLEX and QTRAN, two algorithms with representational capacity $\mathcal{Q}^{\text{IGM}}$, perform outstanding numerical stability in the proposed MDP example. By contrast, the phenomenon of unbounded divergence happens to both VDN and QMIX, whose function classes are incomplete in terms of the IGM constraint. This experiment is a didactic study connection between our theoretical implications and practical algorithms. Combining the theoretical and empirical results above, we summarize this section by the following insights.

**Implication 2.** *Multi-agent Q-learning with linear value decomposition potentially suffers from the risk of unbounded divergence from arbitrary initialization. On-policy data distribution or richer Q function classes can improve its local or global convergence, respectively.*

## 6 EMPIRICAL ANALYSIS

In this section, we conduct an empirical study to connect our theoretical implications to practical scenarios of deep multi-agent Q-learning algorithms. An empirical analysis of a didactic example, a two-state MMDP, has been carried out in Section 5, which shows that the linear value decomposition structure needs to improve training stability in offline mode. In order to verify other implications, here we evaluate four state-of-the-art deep-learning-based methods, VDN (Sunehag et al., 2018), QMIX (Rashid et al., 2018), QTRAN (Son et al., 2019), and QPLEX (Wang et al., 2020a) on the matrix game proposed by QTRAN and StarCraft Multi-Agent Challenge (SMAC) benchmark tasks (Samvelyan et al., 2019). The implementation details of four baselines and experimental settings are deferred to Appendix F. We test all experiments with 6 random seeds and demonstrate them with median performance and 25-75% percentiles.

| $a_2$ \ $a_1$ | $\mathcal{A}^{(1)}$ | $\mathcal{A}^{(2)}$ | $\mathcal{A}^{(3)}$ |
|---|---|---|---|
| $\mathcal{A}^{(1)}$ | **8** | -12 | -12 |
| $\mathcal{A}^{(2)}$ | -12 | 0 | 0 |
| $\mathcal{A}^{(3)}$ | -12 | 0 | 0 |

(a) Payoff of matrix game

| $a_2$ \ $a_1$ | $\mathcal{A}^{(1)}$ | $\mathcal{A}^{(2)}$ | $\mathcal{A}^{(3)}$ |
|---|---|---|---|
| $\mathcal{A}^{(1)}$ | -6.22 | -4.89 | -4.89 |
| $\mathcal{A}^{(2)}$ | -4.89 | **-3.56** | -3.56 |
| $\mathcal{A}^{(3)}$ | -4.89 | -3.56 | -3.56 |

(b) $Q_{\text{tot}}$ of FQI-LVD

| $a_2$ \ $a_1$ | $\mathcal{A}^{(1)}$ | $\mathcal{A}^{(2)}$ | $\mathcal{A}^{(3)}$ |
|---|---|---|---|
| $\mathcal{A}^{(1)}$ | -6.23 | -4.90 | -4.90 |
| $\mathcal{A}^{(2)}$ | -4.90 | **-3.57** | -3.57 |
| $\mathcal{A}^{(3)}$ | -4.90 | -3.57 | -3.57 |

(c) $Q_{\text{tot}}$ of VDN

Table 1: (a) Payoff matrix of the one-step game. Boldface means the optimal joint action selection from payoff matrix. (b,c) Joint action-value functions $Q_{\text{tot}}$ of FQI-LVD and VDN. Boldface means the greedy joint action selection from $Q_{\text{tot}}$.

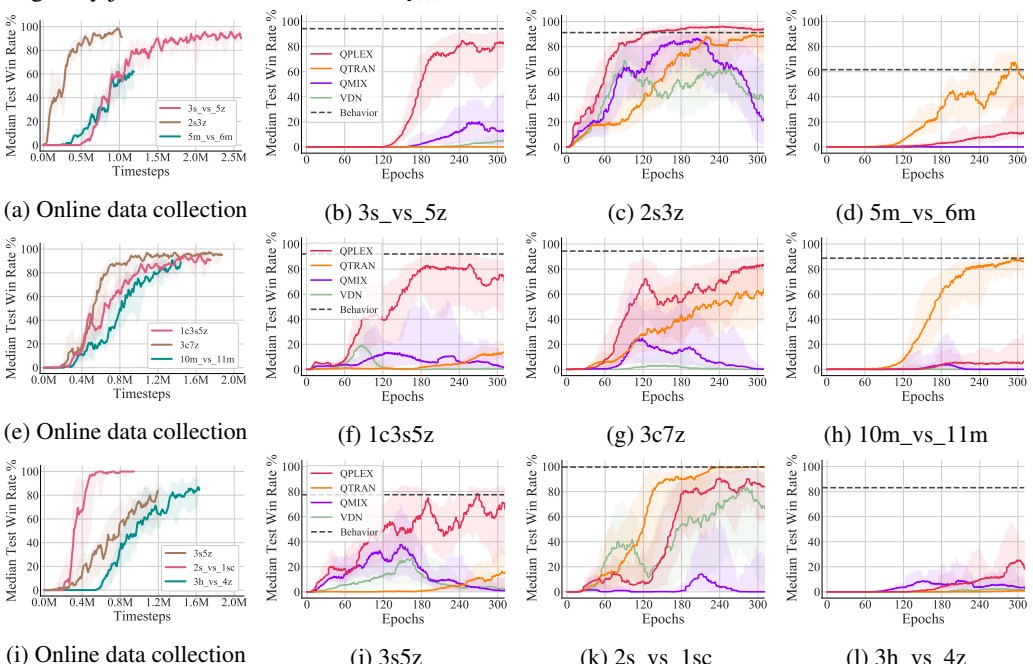

Figure 2: (a,c,i) Constructing datasets using online data collection of VDN. (b-d,f-h,j-l) Evaluating the performance of deep multi-agent Q-learning algorithms with a given static dataset on nine maps.

### 6.1 IS OUR CLOSED-FORM UPDATE RULE OF LINEAR VALUE DECOMPOSITION CONSISTENT WITH THE DEEP-LEARNING-BASED EMPIRICAL RESULTS?

As shown in Theorem 1, we derive the closed-form update rule of FQI-LVD. From an optimization perspective, FQI-LVD and VDN share the same objective function (see Definition 1) but have different optimization methods, i.e., $\arg\min$ vs. gradient descent. Starting from a common matrix game used by QTRAN (Son et al., 2019) and QPLEX (Wang et al., 2020a) stated in Table 1a, we will illustrate the correctness of our closed-form formulation. This matrix game describes a simple cooperative multi-agent, which includes two agents and three actions. Miscoordination penalties are also considered and the optimal strategy for two agents is to perform action $\mathcal{A}^{(1)}$ simultaneously. We adopt a uniform data distribution to conduct this didactic example.

Table 1b and 1c show the joint action-value functions of FQI-LVD and VDN, respectively. Comparing with these two joint action-value functions, we find that the estimation error of VDN is only $\|Q_{\text{tot}}^{\text{FQI-LVD}} - Q_{\text{tot}}^{\text{VDN}}\|_\infty = 0.01$, which corresponds to a 0.2% relative error. This simulation result strongly illustrates the accuracy of Theorem 1. A learning curve of relative error for Table 1c is provided in Appendix G.1. In addition, as discussed by QTRAN and QPLEX, VDN with limited function class cannot learn the optimal policy in this didactic matrix game. The joint action-value functions of QPLEX, QTRAN, and QMIX are deferred to Appendix G.2, where QPLEX and QTRAN can solve this task, but QMIX cannot.

### 6.2 IS LINEAR VALUE DECOMPOSITION LIMITED IN OFFLINE TRAINING?

Section 5 shows that in offline training mode, linear value decomposition is limited in a didactic MMDP task. In order to generalize our implications to complex domains, we investigate the

performance of deep multi-agent Q-learning in the StarCraft II benchmark tasks with offline data collection. Recently, offline reinforcement learning has attracted great attention because it can equip with multi-source datasets and is regarded as a key step towards real-world applications (Dulac-Arnold et al., 2019; Levine et al., 2020). Differing from other related work studying *distributional shift* (Fujimoto et al., 2019; Levine et al., 2020; Yu et al., 2020), we aim to adopt a diverse dataset to investigate the effect of the expressiveness of a value decomposition structure on offline training, i.e., which value decomposition structure is suitable for multi-agent offline reinforcement learning. These datasets are constructed by training a behavior policy of VDN (Sunehag et al., 2018) and collecting a fixed number of experienced episodes during the whole training procedure.

We evaluate the learning curve of StarCraft II on nine common maps. The results are shown in Figure 2. To approximate the MMDP setting, we concatenate the global state with the local observations for each agent to handle partial observability. Figure 2(b-d,f-h,j-l) illustrate that VDN (Sunehag et al., 2018) and QMIX (Rashid et al., 2018) performs poorly and cannot utilize well the offline dataset collected by an unfamiliar behavior policy. In contrast, QPLEX (Wang et al., 2020a) and QTRAN (Son et al., 2019) with richer $Q$ function class perform pretty well, which indicates that the expressiveness of value decomposition structures dramatically affects the performance of multi-agent offline Q-learning. The learning curves of *Behavior* line are shown in Figure 2(a,e,i), which is implemented by VDN with $\epsilon$-greedy online data collection. Figure 2(a,e,i) show that VDN with online data collection can solve these nine tasks, but cannot with offline data collection, that is, there is a considerable gap between online and offline training with linear value decomposition. Although the distribution shift (Levine et al., 2020) can be a potential cause of this gap, the remarkable performance of QPLEX and QTRAN suggests that our datasets should be sufficient for offline training.

In contrast to the theoretical convergence analysis stated in Section 5, in this subsection, empirical experiments aim to conduct the performance of the deep-learning-based implementation of linear value decomposition (i.e., VDN) in the online and offline data collection settings. We have designed several comparative experiments to demonstrate two empirical implications shared with the above theoretical understanding (see Implication 2): 1) VDN with online data collection has superior performance than offline data collection. 2) VDN, a deep-learning-based algorithm with a linear value decomposition structure, has considerable limitations in offline training; while QPLEX and QTRAN, two deep-learning-based algorithms with or approximately with complete IGM function class, are the state-of-the-art value decomposition algorithms for multi-agent offline training.

# 7 CONCLUSION

This paper makes an initial effort to provide theoretical analysis on multi-agent Q-learning with value decomposition. We derive a closed-form solution to the empirical Bellman error minimization with linear value decomposition. Based on this novel result, we reveal the implicit credit assignment mechanism of linear value decomposition learning and provide a formal analysis of its learning stability and convergence. We also formally show that on-policy training or a richer value function class can improve the stability of factorized multi-agent Q-learning. Empirical results are conducted with state-of-the-art deep multi-agent Q-learning with value decomposition and verify theoretical insights in both didactic examples and complex StarCraft II benchmark tasks.

To close this paper, we connect our results with an additional related literature named relative overgeneralization pathology (Wiegand, 2003). In the empirical studies of cooperative learning, the behaviors of individual agents are usually negatively affected by their uncooperative partners. Focusing on a similar issue, our theoretical analysis of the implicit counterfactual credit assignment provides a detailed characterization to understand this pathological phenomenon in linear value decomposition, which also provides insights for the corresponding feasible solutions. Regarding relative overgeneralization pathology, coordination graphs (Böhmer et al., 2020) explore a different methodology for cooperative multi-agent Q-learning. This method allows collaborated action-selection through communications, which does not follow the principle of IGM consistency. In comparison with linear value decomposition, coordination graphs use a higher-order value decomposition structure (Castellini et al., 2019) in the view of function approximation. Supplementary to the results of this paper, the value factorization of coordination graphs provides a different and promising perspective for future studies.

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

## A  OMITTED PROOFS IN SECTION 4

**Definition 1** (FQI-LVD). *Given a dataset D, FQI-LVD specifies the action-value function class*

$$\mathcal{Q}^{LVD} = \left\{ Q \mid Q_{tot}(\cdot, \mathbf{a}) = \sum_{i=1}^{n} Q_i(\cdot, a_i), \forall \mathbf{a} \in \mathbf{A} \text{ and } \left[ \forall Q_i \in \mathbb{R}^{|\mathcal{S}||\mathcal{A}|} \right]_{i=1}^{n} \right\} \tag{6}$$

*and induces the empirical Bellman operator $\mathcal{T}_D^{LVD}$:*

$$Q^{(t+1)} \leftarrow \mathcal{T}_D^{LVD} Q^{(t)} \equiv \underset{Q \in \mathcal{Q}^{LVD}}{\arg\min} \sum_{(s,\mathbf{a},s') \in \mathcal{S} \times \mathbf{A} \times \mathcal{S}} p_D(\mathbf{a}, s'|s) \left( \hat{y}^{(t)}(s, \mathbf{a}, s') - \sum_{i=1}^{n} Q_i(s, a_i) \right)^2$$

$$= \underset{Q \in \mathcal{Q}^{LVD}}{\arg\min} \sum_{(s,\mathbf{a}) \in \mathcal{S} \times \mathbf{A}} p_D(\mathbf{a}|s) \left( y^{(t)}(s, \mathbf{a}) - \sum_{i=1}^{n} Q_i(s, a_i) \right)^2, \tag{7}$$

*where $\hat{y}^{(t)}(s, \mathbf{a}, s') = r(s, \mathbf{a}) + \gamma \max_{\mathbf{a}'} Q_{tot}^{(t)}(s', \mathbf{a}')$ denotes the sample-based regression target. $y^{(t)}(s, \mathbf{a}) = (\mathcal{T}Q^{(t)})_{tot}(s, \mathbf{a}) = r(s, \mathbf{a}) + \gamma \mathbb{E}_{s' \sim P(\cdot|s,\mathbf{a})} \left[ \max_{\mathbf{a}'} Q_{tot}^{(t)}(s', \mathbf{a}') \right]$ denotes the ground-truth target value derived by Bellman optimality operator. $p_D(\mathbf{a}, s'|s) = p_D(\mathbf{a}|s) P(s'|s, \mathbf{a})$ denotes the empirical probability of the event that agents execute joint action $\mathbf{a}$ on state $s$ and transit to $s'$. $Q_{tot}$ and $[Q_i]_{i=1}^{n}$ refer to the discussion of CTDE defined in Section 3.3.*

**Lemma 1.** *The empirical Bellman operator $\mathcal{T}_D^{LVD}$ defined in Definition 1 is equivalent to*

$$\mathcal{T}_D^{LVD} Q^{(t)} \equiv \underset{Q \in \mathcal{Q}^{LVD}}{\arg\min} \sum_{(s,\mathbf{a},s') \in \mathcal{S} \times \mathbf{A} \times \mathcal{S}} p_D(\mathbf{a}, s'|s) \left( \hat{y}^{(t)}(s, \mathbf{a}, s') - \sum_{i=1}^{n} Q_i(s, a_i) \right)^2$$

$$= \underset{Q \in \mathcal{Q}^{LVD}}{\arg\min} \sum_{(s,\mathbf{a}) \in \mathcal{S} \times \mathbf{A}} p_D(\mathbf{a}|s) \left( y^{(t)}(s, \mathbf{a}) - \sum_{i=1}^{n} Q_i(s, a_i) \right)^2. \tag{12}$$

*Proof.* Recall Definition 1,

$$\mathcal{T}_D^{LVD} Q^{(t)} \equiv \underset{Q \in \mathcal{Q}^{LVD}}{\arg\min} \sum_{(s,\mathbf{a},s') \in \mathcal{S} \times \mathbf{A} \times \mathcal{S}} p_D(\mathbf{a}, s'|s) \left( \hat{y}^{(t)}(s, \mathbf{a}, s') - \sum_{i=1}^{n} Q_i(s, a_i) \right)^2$$

$$= \underset{Q \in \mathcal{Q}^{LVD}}{\arg\min} \underset{(s,\mathbf{a},s') \sim D}{\mathbb{E}} \left[ \left( \hat{y}^{(t)}(s, \mathbf{a}, s') - Q_{tot}(s, \mathbf{a}) \right)^2 \right]$$

$$= \underset{Q \in \mathcal{Q}^{LVD}}{\arg\min} \underset{(s,\mathbf{a},s') \sim D}{\mathbb{E}} \left[ \left( \hat{y}^{(t)}(s, \mathbf{a}, s') - y^{(t)}(s, \mathbf{a}) + y^{(t)}(s, \mathbf{a}) - Q_{tot}(s, \mathbf{a}) \right)^2 \right]$$

$$= \underset{Q \in \mathcal{Q}^{LVD}}{\arg\min} \underset{(s,\mathbf{a},s') \sim D}{\mathbb{E}} \left[ \left( \hat{y}^{(t)}(s, \mathbf{a}, s') - y^{(t)}(s, \mathbf{a}) \right)^2 \right]$$

$$+ \underset{(s,\mathbf{a},s') \sim D}{\mathbb{E}} \left[ 2 \left( \hat{y}^{(t)}(s, \mathbf{a}, s') - y^{(t)}(s, \mathbf{a}) \right) \left( y^{(t)}(s, \mathbf{a}) - Q_{tot}(s, \mathbf{a}) \right) \right]$$

$$+ \underset{(s,\mathbf{a},s') \sim D}{\mathbb{E}} \left[ \left( y^{(t)}(s, \mathbf{a}) - Q_{tot}(s, \mathbf{a}) \right)^2 \right] \tag{13}$$

The first term is a constant since $y^{(t)}$ and $\hat{y}^{(t)}$ are fixed targets.

The second term is equal to zero since

$$\underset{(s,\mathbf{a},s') \sim D}{\mathbb{E}} \left[ 2 \left( \hat{y}^{(t)}(s, \mathbf{a}, s') - y^{(t)}(s, \mathbf{a}) \right) \left( y^{(t)}(s, \mathbf{a}) - Q_{tot}(s, \mathbf{a}) \right) \right]$$

$$= 2 \underset{(s,\mathbf{a}) \sim D}{\mathbb{E}} \left[ \underbrace{\underset{s' \sim P(\cdot|s,\mathbf{a})}{\mathbb{E}} \left[ \hat{y}^{(t)}(s, \mathbf{a}, s') - y^{(t)}(s, \mathbf{a}) \right]}_{=0} \left( y^{(t)}(s, \mathbf{a}) - Q_{tot}(s, \mathbf{a}) \right) \right]$$

$$= 0. \tag{14}$$

The third term exactly corresponds to Eq. (12).  □

**Lemma 2.** *Considering following weighted linear regression problem*

$$\min_{\mathbf{x}} \| \sqrt{\mathbf{p}^\top} \cdot (\mathbf{Ax} - \mathbf{b}) \|_2^2 \tag{15}$$

*where $\mathbf{A} \in \mathbb{R}^{m^n \times mn}, \mathbf{x} \in \mathbb{R}^{mn}, \mathbf{b}, \mathbf{p} \in \mathbb{R}^{m^n}$, $m, n \in \mathbb{Z}^+$. Besides, $\mathbf{A}$ is m-ary encoding matrix namely $\forall i \in [m^n], j \in [mn]$*

$$\mathbf{A}_{i,j} = \begin{cases} 1, & if \quad \exists u \in [n], j = m \times u + (\lfloor i/m^u \rfloor \bmod m), \\ 0, & otherwise. \end{cases} \tag{16}$$

*For simplicity, $j^{th}$ row of $\mathbf{A}$ corresponds to a m-ary number $\vec{a}_j = (j)_m$ where $\vec{a} = a_0 a_1 \ldots a_{n-1}$, with $a_u \in [m], \forall u \in [n]$. Assume $\mathbf{p}$ is a positive vector which follows that*

$$\mathbf{p}_j = \mathbf{p}(\vec{a}_j) = \prod_{u \in [n]} p_u(a_{u,j}), \; where \; p_u : [m] \to (0,1) \; and \; \sum_{a_u \in [m]} p_u(a_u) = 1, \forall u \in [n] \tag{17}$$

*The optimal solution of this problem is the following. Denote $i = u \times m + v, v \in [m], u \in [n]$ and an arbitrary vector $\mathbf{w} \in \mathbb{R}^{mn}$*

$$\mathbf{x}_i^* = \sum_{\vec{a}} \frac{\mathbf{p}(\vec{a})}{p_u(a_u)} \mathbf{b}_{\vec{a}} \cdot \mathbf{1}(a_u = v) - \frac{n-1}{n} \mathbf{p}(\vec{a}) \mathbf{b}_{\vec{a}} - \frac{1}{mn} \sum_{i' \in [mn]} \mathbf{w}_{i'} + \frac{1}{m} \sum_{v' \in [m]} \mathbf{w}_{um+v'} \tag{18}$$

*Proof.* For brevity, denote

$$\mathbf{A}^p = \sqrt{\mathbf{p}^\top} \cdot \mathbf{A}, \qquad \mathbf{b}^p = \sqrt{\mathbf{p}^\top} \cdot \mathbf{b} \tag{19}$$

Then the weighted linear regression becomes a standard Linear regression problem w.r.t $\mathbf{A}^p, \mathbf{b}^p$. To compute the optimal solutions, we need to calculate the Moore-Penrose inverse of $\mathbf{A}^p$. The sufficient and necessary condition of this inverse matrix $\mathbf{A}^{p,\dagger} \in \mathbb{R}^{mn \times m^n}$ is the following three statements (Moore, 1920):

$$(1) \; \mathbf{A}^p \mathbf{A}^{p,\dagger} \text{ and } \mathbf{A}^{p,\dagger} \mathbf{A}^p \text{ are self-adjoint} \tag{20}$$

$$(2) \; \mathbf{A}^p = \mathbf{A}^p \mathbf{A}^{p,\dagger} \mathbf{A}^p \tag{21}$$

$$(3) \; \mathbf{A}^{p,\dagger} = \mathbf{A}^{p,\dagger} \mathbf{A}^p \mathbf{A}^{p,\dagger} \tag{22}$$

We consider the following matrix as $\mathbf{A}^{p,\dagger}$ and we prove that it satisfies all three statements. For $\forall i \in [mn], i = u \times m + v, u \in [n], v \in [m], j \in [m^n]$

$$\mathbf{A}_{i,j}^{p,\dagger} = \mathbf{A}_{i,\vec{a}_j}^{p,\dagger}$$

$$= \sqrt{\frac{\mathbf{p}(\vec{a}_{-u,j})}{p_u(a_{u,j})}} \cdot \mathbf{1}(a_{u,j} = v) - \frac{n-1}{n} \sqrt{\mathbf{p}(\vec{a}_j)} - \frac{1}{m} \sqrt{\frac{\mathbf{p}(\vec{a}_{-u,j})}{p_u(a_{u,j})}} + \frac{1}{mn} \sum_{u'=0}^{n-1} \sqrt{\frac{\mathbf{p}(\vec{a}_{-u',j})}{p_{u'}(a_{u',j})}} \tag{23}$$

where $\mathbf{p}(\vec{a}_{-u}) = \prod_{u' \neq u} p_{u'}(a_{u'})$.

First, we verify that $\mathbf{A}^p \mathbf{A}^{p,\dagger}$ is a $m^n \times m^n$ self-adjoint matrix in statement (1). For simplicity, $O(\vec{a}_i, \vec{a}_j) = \{u | a_{u,i} = a_{u,j}, u \in [n]\}$.

$$
(\mathbf{A}^p \mathbf{A}^{p,\dagger})_{i,j} = \sum_{u \in [n]} \sqrt{\mathbf{p}(\vec{a}_i)} \left[ \sqrt{\frac{\mathbf{p}(\vec{a}_{-u,j})}{p_u(a_{u,j})}} \cdot \mathbf{1}(a_{u,j} = a_{u,i}) - \frac{n-1}{n} \sqrt{\mathbf{p}(\vec{a}_j)} - \frac{1}{m} \sqrt{\frac{\mathbf{p}(\vec{a}_{-u,j})}{p_u(a_{u,j})}} \right.
$$

$$
\left. + \frac{1}{mn} \sum_{u'=0}^{n-1} \sqrt{\frac{\mathbf{p}(\vec{a}_{-u',j})}{p_{u'}(a_{u',j})}} \right]
$$

$$
= \sum_{u \in O(\vec{a}_i, \vec{a}_j)} \frac{\sqrt{\mathbf{p}(\vec{a}_j)\mathbf{p}(\vec{a}_i)}}{p_u(a_{u,j})} - \frac{n-1}{n} \sum_{u \in [n]} \sqrt{\mathbf{p}(\vec{a}_i)\mathbf{p}(\vec{a}_j)} - \frac{1}{m} \sum_{u \in [n]} \frac{\sqrt{\mathbf{p}(\vec{a}_j)\mathbf{p}(\vec{a}_i)}}{p_u(a_{u,j})}
$$

$$
+ \sum_{u \in [n]} \frac{1}{mn} \sum_{u'=0}^{n-1} \frac{\sqrt{\mathbf{p}(\vec{a}_j)\mathbf{p}(\vec{a}_i)}}{p_{u'}(a_{u',j})}
$$

$$
= \sum_{u \in O(\vec{a}_i, \vec{a}_j)} \frac{\sqrt{\mathbf{p}(\vec{a}_{j'})\mathbf{p}(\vec{a}_i)}}{p_u(a_{u,j})} - (n-1)\sqrt{\mathbf{p}(\vec{a}_i)\mathbf{p}(\vec{a}_j)} - \frac{1}{m} \sum_{u \in [n]} \frac{\sqrt{\mathbf{p}(\vec{a}_j)\mathbf{p}(\vec{a}_i)}}{p_u(a_{u,j})}
$$

$$
+ \frac{1}{m} \sum_{u \in [n]} \frac{\sqrt{\mathbf{p}(\vec{a}_j)\mathbf{p}(\vec{a}_i)}}{p_u(a_{u,j})}
$$

$$
= \sum_{u \in O(\vec{a}_i, \vec{a}_j)} \frac{\sqrt{\mathbf{p}(\vec{a}_j)\mathbf{p}(\vec{a}_i)}}{p_u(a_{u,j})} - (n-1)\sqrt{\mathbf{p}(\vec{a}_i)\mathbf{p}(\vec{a}_j)} \tag{24}
$$

Observe that $p_u(a_{u,j}) = p_u(a_{u,i})$ if $a_{u,i} = a_{u,j}$, thus $(\mathbf{A}^p \mathbf{A}^{p,\dagger})_{i,j} = (\mathbf{A}^p \mathbf{A}^{p,\dagger})_{j,i}$ for any $i, j \in [m^n]$. This proves that $\mathbf{A}^p \mathbf{A}^{p,\dagger}$ is self-adjoint.

Second, we prove that $\mathbf{A}^{p,\dagger} \mathbf{A}^p$ is a $mn \times mn$ self-adjoint matrix and has surprisingly succinct form. Let $i = u \times m + v, u \in [n], v \in [m]$.

1. $i = i'$. Besides, $O(i) = \{\vec{a} \in [m^n] | a_u = v\}$

$$
(\mathbf{A}^{p,\dagger} \mathbf{A}^p)_{i,i} = \sum_{\vec{a} \in O(i)} \sqrt{\mathbf{p}(\vec{a})} \left[ \sqrt{\frac{\mathbf{p}(\vec{a}_{-u})}{p_u(a_u)}} \cdot \mathbf{1}(a_u = v) - \frac{n-1}{n} \sqrt{\mathbf{p}(\vec{a})} - \frac{1}{m} \sqrt{\frac{\mathbf{p}(\vec{a}_{-u})}{p_u(a_u)}} \right.
$$

$$
\left. + \frac{1}{mn} \sum_{u'=0}^{n-1} \sqrt{\frac{\mathbf{p}(\vec{a}_{-u'})}{p_{u'}(a_{u'})}} \right]
$$

$$
= \sum_{\vec{a} \in O(i)} \frac{\mathbf{p}(\vec{a})}{p_u(a_u)} - \frac{n-1}{n} \mathbf{p}(\vec{a}) - \frac{1}{m} \frac{\mathbf{p}(\vec{a})}{p_u(a_u)} + \frac{1}{mn} \sum_{u'=0}^{n-1} \frac{\mathbf{p}(\vec{a})}{p_{u'}(a_{u'})}
$$

$$
= \sum_{\vec{a} \in O(i)} \left( \mathbf{p}(\vec{a}_{-u}) - \frac{1}{m} \mathbf{p}(\vec{a}_{-u}) + \frac{1}{mn} \sum_{u'=0}^{n-1} \mathbf{p}(\vec{a}_{-u'}) \right) - \frac{n-1}{n} p_u(a_u = v)
$$

$$
= 1 - \frac{1}{m} - \frac{n-1}{n} p_u(a_u = v) + \frac{1}{mn} \sum_{\substack{u' \in [n] \\ u' \neq u}} \sum_{\vec{a} \in O(i)} \mathbf{p}(\vec{a}_{-u'})
$$

$$
+ \frac{1}{mn} \sum_{\vec{a} \in O(i)} \mathbf{p}(\vec{a}_{-u})
$$

$$
= 1 - \frac{1}{m} - \frac{n-1}{n} p_u(a_u = v) + \frac{1}{mn} + \frac{n-1}{mn} m p_u(a_u = v)
$$

$$
= 1 - \frac{1}{m} + \frac{1}{mn} \tag{25}
$$

2. $i = u \times m + v, i' = u \times m + v', v \neq v'$. This implies that $Q(i) \cap O(i') = \emptyset$

$$
\begin{aligned}
(\mathbf{A}^{p,\dagger}\mathbf{A}^p)_{i,i'} &= \sum_{\vec{a} \in O(i')} \sqrt{\mathbf{p}(\vec{a})}[\sqrt{\frac{\mathbf{p}(\vec{a}_{-u})}{p_u(a_u)}} \cdot \mathbf{1}(a_u = v) - \frac{n-1}{n}\sqrt{\mathbf{p}(\vec{a})} \\
&\quad - \frac{1}{m}\sqrt{\frac{\mathbf{p}(\vec{a}_{-u})}{p_u(a_u)}} + \frac{1}{mn}\sum_{u'=0}^{n-1}\sqrt{\frac{\mathbf{p}(\vec{a}_{-u'})}{p_{u'}(a_{u'})}}] \\
&= \sum_{\vec{a} \in O(i) \cap O(i')} \frac{\mathbf{p}(\vec{a})}{p_u(a_u)} - \frac{n-1}{n}\sum_{\vec{a} \in O(i')}\mathbf{p}(\vec{a}) - \frac{1}{m}\sum_{\vec{a} \in O(i')}\frac{\mathbf{p}(\vec{a})}{p_u(a_u)} \\
&\quad + \frac{1}{mn}\sum_{\substack{u' \in [n] \\ u' \neq u}}\sum_{\vec{a} \in O(i')}\frac{\mathbf{p}(\vec{a})}{p_{u'}(a_{u'})} + \frac{1}{mn}\sum_{\vec{a} \in O(i')}\frac{\mathbf{p}(\vec{a})}{p_u(a_u)} \\
&= -\frac{n-1}{n}p_u(a_u = v') - \frac{1}{m} + \frac{n-1}{mn}\sum_{\vec{a} \in O(i')}\mathbf{p}(\vec{a}_{-u'}) + \frac{1}{mn} \\
&= -\frac{1}{m} + \frac{1}{mn}
\end{aligned}
\tag{26}
$$

3. $i = u_1 \times m + v_1, i' = u_2 \times m + v_2, u_1 \neq u_2$.

$$
\begin{aligned}
(\mathbf{A}^{p,\dagger}\mathbf{A}^p)_{i,i'} &= \sum_{\vec{a} \in O(i')} \sqrt{\mathbf{p}(\vec{a})}[\sqrt{\frac{\mathbf{p}(\vec{a}_{-u_1})}{p_{u_1}(a_{u_1})}} \cdot \mathbf{1}(a_{u_1} = v) - \frac{n-1}{n}\sqrt{\mathbf{p}(\vec{a})} \\
&\quad - \frac{1}{m}\sqrt{\frac{\mathbf{p}(\vec{a}_{-u_1})}{p_{u_1}(a_{u_1})}} + \frac{1}{mn}\sum_{u'=0}^{n-1}\sqrt{\frac{\mathbf{p}(\vec{a}_{-u'})}{p_{u'}(a_{u'})}}] \\
&= \sum_{\vec{a} \in O(i) \cap O(i')} \frac{\mathbf{p}(\vec{a})}{p_{u_1}(a_{u_1})} - \frac{n-1}{n}\sum_{\vec{a} \in O(i')}\mathbf{p}(\vec{a}) - \frac{1}{m}\sum_{\vec{a} \in O(i')}\frac{\mathbf{p}(\vec{a})}{p_{u_1}(a_{u_1})} \\
&\quad + \frac{1}{mn}\sum_{\substack{u' \in [n] \\ u' \neq u_2}}\sum_{\vec{a} \in O(i')}\frac{\mathbf{p}(\vec{a})}{p_{u'}(a_{u'})} + \frac{1}{mn}\sum_{\vec{a} \in O(i')}\frac{\mathbf{p}(\vec{a})}{p_{u_2}(a_{u_2})} \\
&= p_{u_2}(a_{u_2}) - \frac{n-1}{n}p_{u_2}(a_{u_2}) - p_{u_2}(a_{u_2}) + \frac{n-1}{mn}mp_{u_2}(a_{u_2}) + \frac{1}{mn} \\
&= \frac{1}{mn}
\end{aligned}
\tag{27}
$$

Observe that $\mathbf{A}^{p,\dagger}\mathbf{A}^p$ is self-adjoint by equation (2,3,4) and the expression is succinct.

Third, we verify statement (2). Since we have computed $\mathbf{A}^{p,\dagger}\mathbf{A}^p$, the verification is straightforward. For brevity, denote $\mathbf{A}^{p,\dagger}\mathbf{A}^p$ as $\mathbf{A}_0^p$

$$
\begin{aligned}
(\mathbf{A}^p\mathbf{A}_0^p)_{\vec{a},i} &= \sqrt{\mathbf{p}(\vec{a})}\sum_{u \in [n]}(\mathbf{A}_0^p)_{um+a_u,i} \\
&= \sqrt{\mathbf{p}(\vec{a})}\left(\mathbf{1}(\exists u \in [n], i = um + a_u) - \frac{1}{m} + \frac{1}{mn} + (n-1)\frac{1}{mn}\right) \\
&= \sqrt{\mathbf{p}(\vec{a})} \cdot \mathbf{1}(\exists u \in [n], i = um + a_u)
\end{aligned}
\tag{28}
$$

Thus, $\mathbf{A}^p\mathbf{A}^{p,\dagger}\mathbf{A}^p = \mathbf{A}^p$.

Similarly, we can verify statement (3). Suppose $i_0 = u_0 \times m + v_0$, we have

$$(\mathbf{A}_0^p \mathbf{A}^{p,\dagger})_{i_0,\vec{a}} = \frac{1}{mn} \sum_{\substack{u \neq u_0 \\ u \in [n]}} \sum_{v \in [m]} [\sqrt{\frac{\mathbf{p}(\vec{a}_{-u})}{p_u(a_u)}} \cdot \mathbf{1}(a_u = v)$$

$$- \frac{n-1}{n} \sqrt{\mathbf{p}(\vec{a})} - \frac{1}{m} \sqrt{\frac{\mathbf{p}(\vec{a}_{-u})}{p_u(a_u)}} + \frac{1}{mn} \sum_{u'=0}^{n-1} \sqrt{\frac{\mathbf{p}(\vec{a}_{-u'})}{p_{u'}(a_{u'})}}]$$

$$+ \sum_{v \in [m]} (\mathbf{1}(v = v_0) - \frac{1}{m} + \frac{1}{mn})[\sqrt{\frac{\mathbf{p}(\vec{a}_{-u_0})}{p_{u_0}(a_{u_0})}} \cdot \mathbf{1}(a_{u_0} = v)$$

$$- \frac{n-1}{n} \sqrt{\mathbf{p}(\vec{a})} - \frac{1}{m} \sqrt{\frac{\mathbf{p}(\vec{a}_{-u_0})}{p_{u_0}(a_{u_0})}} + \frac{1}{mn} \sum_{u'=0}^{n-1} \sqrt{\frac{\mathbf{p}(\vec{a}_{-u'})}{p_{u'}(a_{u'})}}]$$

$$= \frac{1}{mn} \sum_{u \in [n]} \sum_{v \in [m]} [\sqrt{\frac{\mathbf{p}(\vec{a}_{-u})}{p_u(a_u)}} \cdot \mathbf{1}(a_u = v)$$

$$- \frac{n-1}{n} \sqrt{\mathbf{p}(\vec{a})} - \frac{1}{m} \sqrt{\frac{\mathbf{p}(\vec{a}_{-u})}{p_u(a_u)}} + \frac{1}{mn} \sum_{u'=0}^{n-1} \sqrt{\frac{\mathbf{p}(\vec{a}_{-u'})}{p_{u'}(a_{u'})}}]$$

$$+ \sum_{v \in [m]} (\mathbf{1}(v = v_0) - \frac{1}{m})[-\frac{n-1}{n} \sqrt{\mathbf{p}(\vec{a})} - \frac{1}{m} \sqrt{\frac{\mathbf{p}(\vec{a}_{-u_0})}{p_{u_0}(a_{u_0})}}$$

$$+ \frac{1}{mn} \sum_{u'=0}^{n-1} \sqrt{\frac{\mathbf{p}(\vec{a}_{-u'})}{p_{u'}(a_{u'})}}] + \sum_{v \in [m]} (\mathbf{1}(v = v_0) - \frac{1}{m}) \sqrt{\frac{\mathbf{p}(\vec{a}_{-u_0})}{p_{u_0}(a_{u_0})}} \cdot \mathbf{1}(a_{u_0} = v)$$

$$= \frac{1}{mn} \sum_{u \in [n]} \sqrt{\frac{\mathbf{p}(\vec{a}_{-u})}{p_u(a_u)}} - \frac{n-1}{n} \sqrt{\mathbf{p}(\vec{a})}$$

$$+ \frac{1}{n} \sum_{u \in [n]} [-\frac{1}{m} \sqrt{\frac{\mathbf{p}(\vec{a}_{-u})}{p_u(a_u)}} + \frac{1}{mn} \sum_{u'=0}^{n-1} \sqrt{\frac{\mathbf{p}(\vec{a}_{-u'})}{p_{u'}(a_{u'})}}]$$

$$+ \left( \sum_{v \in [m]} (\mathbf{1}(v = v_0) - \frac{1}{m}) \right) [-\frac{n-1}{n} \sqrt{\mathbf{p}(\vec{a})} - \frac{1}{m} \sqrt{\frac{\mathbf{p}(\vec{a}_{-u_0})}{p_{u_0}(a_{u_0})}}$$

$$+ \frac{1}{mn} \sum_{u'=0}^{n-1} \sqrt{\frac{\mathbf{p}(\vec{a}_{-u'})}{p_{u'}(a_{u'})}}] + (\mathbf{1}(a_{u_0} = v_0) - \frac{1}{m}) \sqrt{\frac{\mathbf{p}(\vec{a}_{-u_0})}{p_{u_0}(a_{u_0})}} \tag{29}$$

Clearly, we have the following relations

$$\sum_{u \in [n]} [-\frac{1}{m} \sqrt{\frac{\mathbf{p}(\vec{a}_{-u})}{p_u(a_u)}} + \frac{1}{mn} \sum_{u'=0}^{n-1} \sqrt{\frac{\mathbf{p}(\vec{a}_{-u'})}{p_{u'}(a_{u'})}}] = 0 \tag{30}$$

$$\sum_{v \in [m]} (\mathbf{1}(v = v_0) - \frac{1}{m}) = 0 \tag{31}$$

Thus

$$(\mathbf{A}_0^p \mathbf{A}^{p,\dagger})_{i_0,\vec{a}} = \frac{1}{mn} \sum_{u \in [n]} \sqrt{\frac{\mathbf{p}(\vec{a}_{-u})}{p_u(a_u)}} - \frac{n-1}{n} \sqrt{\mathbf{p}(\vec{a})} + (\mathbf{1}(a_{u_0} = v_0) - \frac{1}{m}) \sqrt{\frac{\mathbf{p}(\vec{a}_{-u_0})}{p_{u_0}(a_{u_0})}} \tag{32}$$

$$= \mathbf{A}_{i_0,\vec{a}}^{p,\dagger} \tag{33}$$

This proves $\mathbf{A}^{p,\dagger} = \mathbf{A}^{p,\dagger}\mathbf{A}^p\mathbf{A}^{p,\dagger}$ in statement (3) and $\mathbf{A}^{p,\dagger}$ is the Moore-Penrose inverse of $\mathbf{A}^p$. Since the optimal solution $\mathbf{x}^* = \mathbf{A}^{p,\dagger}\mathbf{b}^p + (\mathbf{I}_{mn \times mn} - \mathbf{A}^{p,\dagger}\mathbf{A}^p)\mathbf{w}$ where $w \in \mathrm{R}^{mn}$ is any vector (Moore, 1920).

Denote $\mathbf{x}^p = \mathbf{A}^{p,\dagger}\mathbf{b}^p$. We have $\forall i = u \times m + v$

$$\mathbf{x}_i^p = \sum_{\vec{a}} \mathbf{A}_{i,\vec{a}}^{p,\dagger}\sqrt{\mathbf{p}(\vec{a})}\mathbf{b}_{\vec{a}}$$

$$= \sum_{\vec{a}}[\sqrt{\frac{\mathbf{p}(\vec{a}_{-u})}{p_u(a_u)}} \cdot \mathbf{1}(a_u = v) - \frac{n-1}{n}\sqrt{\mathbf{p}(\vec{a})} - \frac{1}{m}\sqrt{\frac{\mathbf{p}(\vec{a}_{-u})}{p_u(a_u)}}$$

$$+ \frac{1}{mn}\sum_{u'=0}^{n-1}\sqrt{\frac{\mathbf{p}(\vec{a}_{-u'})}{p_{u'}(a_{u'})}}]\sqrt{\mathbf{p}(\vec{a})}\mathbf{b}_{\vec{a}}$$

$$= \sum_{\vec{a}}\left[\frac{\mathbf{p}(\vec{a})}{p_u(a_u)} \cdot \mathbf{1}(a_u = v) - \frac{n-1}{n}\mathbf{p}(\vec{a}) - \frac{1}{m}\frac{\mathbf{p}(\vec{a})}{p_u(a_u)} + \frac{1}{mn}\sum_{u'=0}^{n-1}\frac{\mathbf{p}(\vec{a})}{p_{u'}(a_{u'})}\right]\mathbf{b}_{\vec{a}} \qquad (34)$$

From equation (2, 3, 4), we have $i = u \times m + v, i' = u' \times m + v'$

$$(\mathbf{I} - \mathbf{A}^{p,\dagger}\mathbf{A}^p)_{i,i'} = \begin{cases} \frac{1}{m} - \frac{1}{mn} & \text{if } u = u' \\ -\frac{1}{mn} & \text{if } u \neq u' \end{cases} \qquad (35)$$

If we consider $\mathbf{w}$ as the following $i_0 = u_0 \times m + v_0$

$$\mathbf{w}_{i_0} = \sum_{\vec{a} \in O(i_0)}\frac{\mathbf{p}(\vec{a})}{p_{u_0}(a_{u_0})}\mathbf{b}_{\vec{a}} \qquad (36)$$

Then for $i = u \times m + v$

$$((\mathbf{I} - \mathbf{A}^{p,\dagger}\mathbf{A}^p)\mathbf{w})_i = \sum_{\substack{i_0 \in [mn] \\ u \neq u_0}}-\frac{1}{mn}\mathbf{w}_{i_0} + \sum_{i_0:u_0=u}(\frac{1}{m} - \frac{1}{mn})\mathbf{w}_{i_0} \qquad (37)$$

$$= \sum_{\vec{a}}-\frac{1}{mn}\sum_{u'\in[n]}\frac{\mathbf{p}(\vec{a})}{p_{u'}(a_{u'})}\mathbf{b}_{\vec{a}} + \frac{1}{m}\sum_{\vec{a}}\frac{\mathbf{p}(\vec{a})}{p_u(a_u)}\mathbf{b}_{\vec{a}} \qquad (38)$$

Notice that this is exactly the last two terms in equation (5). Therefore, the optimal solutions of this weighted linear regression problem can be written as: $i = u \times m + v, v \in [m], u \in [n]$ and an arbitrary vector $\mathbf{w} \in \mathrm{R}^{mn}$.

$$\mathbf{x}_i^* = \sum_{\vec{a}}\frac{\mathbf{p}(\vec{a})}{p_u(a_u)}\mathbf{b}_{\vec{a}} \cdot \mathbf{1}(a_u = v) - \frac{n-1}{n}\mathbf{p}(\vec{a})\mathbf{b}_{\vec{a}} - \frac{1}{mn}\sum_{i'\in[mn]}\mathbf{w}_{i'} + \frac{1}{m}\sum_{v'\in[m]}\mathbf{w}_{um+v'} \qquad (39)$$

This completes the proof. $\qquad\qquad\qquad\qquad\qquad\qquad\qquad\qquad\qquad\qquad\qquad\qquad\square$

**Theorem 1.** *Let $Q^{(t+1)} = \mathcal{T}_D^{LVD}Q^{(t)}$ denote a single iteration of the empirical Bellman operator. Then $\forall i \in \mathcal{N}, \forall (s, \mathbf{a}) \in \mathcal{S} \times \mathbf{A}$, the individual action-value function $Q_i^{(t+1)}(s, a_i) =$*

$$\underbrace{\mathbb{E}_{a'_{-i}\sim p_D(\cdot|s)}\left[y^{(t)}\left(s, a_i \oplus a'_{-i}\right)\right]}_{\textit{evaluation of the individual action } a_i} - \frac{n-1}{n}\underbrace{\mathbb{E}_{\mathbf{a}'\sim p_D(\cdot|s)}\left[y^{(t)}\left(s, \mathbf{a}'\right)\right]}_{\textit{counterfactual baseline}} + w_i(s), \qquad (8)$$

*where we denote $a_i \oplus a'_{-i} = \langle a'_1, \ldots, a'_{i-1}, a_i, a'_{i+1}, \ldots, a'_n\rangle$. $a'_{-i}$ denotes the action of all agents except for agent $i$. The residue term $\mathbf{w} \equiv [w_i]_{i=1}^n$ is an arbitrary vector satisfying $\forall s, \sum_{i=1}^n w_i(s) = 0$.*

*Proof.* In the formulation of FQI-LVD stated in Definition 1, the empirical Bellman error minimization in Eq. (7) can be regarded as a weighted linear least squares problem as follows: $\forall s \in \mathcal{S}$,

$$\min_{\mathbf{x}} \| \sqrt{\mathbf{p}^\top} \cdot (\mathbf{A}\mathbf{x} - \mathbf{b}) \|_2^2 \tag{40}$$

where let $m, n \in \mathbb{Z}^+$ denote the size of action space $|\mathcal{A}|$ and the number of agents, respectively; $\mathbf{A} \in \mathbb{R}^{m^n \times mn}$ denotes the multi-agent credit assignment coefficient matrix of action-value functions with linear value decomposition; $\mathbf{x} \in \mathbb{R}^{mn}$ denotes individual action-value functions $\left[ Q_i^{(t)}(s, \cdot) \in \mathbb{R}^m \right]_{i=1}^n$ under the empirical Bellman error minimization; according to Lemma 1, $\mathbf{b} \in \mathbb{R}^{m^n}$ denotes the regression target $y^{(t)}(s, \cdot)$ derived by *Bellman optimality operator*; $\mathbf{p} \in \mathbb{R}^{m^n}$ denotes the empirical probability of joint action $\mathbf{a}$ executed on state $s$, $p_D(\mathbf{a}|s)$, which can be factorized to the production of individual components illustrated in Assumption 1.

Besides, $\mathbf{A}$ is m-ary encoding matrix namely $\forall i \in [m^n], j \in [mn]$

$$\mathbf{A}_{i,j} = \begin{cases} 1, & \text{if} \quad \exists u \in [n], j = m \times u + (\lfloor i/m^u \rfloor \bmod m), \\ 0, & \text{otherwise.} \end{cases} \tag{41}$$

For simplicity, $j^{th}$ row of $\mathbf{A}$ corresponds to a m-ary number $\vec{a}_j = (j)_m$ where $\vec{a} = a_0 a_1 \ldots a_{n-1}$, with $a_u \in [m], \forall u \in [n]$. According to the factorizable empirical probability $p_D$ shown in Assumption 1, $\mathbf{p}$ is a corresponding positive vector which follows that

$$\mathbf{p}_j = \mathbf{p}(\vec{a}_j) = \prod_{u \in [n]} p_u(a_{u,j}), \text{ where } p_u : [m] \to (0,1) \text{ and } \sum_{a_u \in [m]} p_u(a_u) = 1, \forall u \in [n] \tag{42}$$

According to Lemma 2, we derive the optimal solution of this problem is the following. Denote $i = u \times m + v, v \in [m], u \in [n]$ and an arbitrary vector $\mathbf{w} \in \mathbb{R}^{mn}$

$$\mathbf{x}_i^* = \sum_{\vec{a}} \frac{\mathbf{p}(\vec{a})}{p_u(a_u)} \mathbf{b}_{\vec{a}} \cdot \mathbf{1}(a_u = v) - \frac{n-1}{n} \mathbf{p}(\vec{a}) \mathbf{b}_{\vec{a}} - \frac{1}{mn} \sum_{i' \in [mn]} \mathbf{w}_{i'} + \frac{1}{m} \sum_{v' \in [m]} \mathbf{w}_{um+v'} \tag{43}$$

which means $\forall i \in \mathcal{N}, \forall (s, \mathbf{a}) \in \mathcal{S} \times \mathbf{A}$, the individual action-value function $Q_i^{(t+1)}(s, a_i) =$

$$\mathbb{E}_{a'_{-i} \sim p_D(\cdot|s)} \left[ y^{(t)} \left( s, a_i \oplus a'_{-i} \right) \right] - \frac{n-1}{n} \mathbb{E}_{\mathbf{a}' \sim p_D(\cdot|s)} \left[ y^{(t)} \left( s, \mathbf{a}' \right) \right] + w_i(s), \tag{44}$$

where we denote $a_i \oplus a'_{-i} = \langle a'_1, \ldots, a'_{i-1}, a_i, a'_{i+1}, \ldots, a'_n \rangle$. $a'_{-i}$ denotes the action of all agents except agent $i$. The residue term $\mathbf{w} \equiv [w_i]_{i=1}^n$ is an arbitrary vector satisfying $\forall s, \sum_{i=1}^n w_i(s) = 0$. $\qquad \square$

## B  OMITTED PROOFS IN SECTION 5.1

**Proposition 1.** *The empirical Bellman operator $\mathcal{T}_D^{LVD}$ is not a $\gamma$-contraction, i.e., the following important property of the standard Bellman optimality operator $\mathcal{T}$ does not hold for $\mathcal{T}_D^{LVD}$ anymore.*

$$\forall Q_{tot}, Q'_{tot} \in \mathcal{Q}, \quad \|\mathcal{T}Q_{tot} - \mathcal{T}Q'_{tot}\|_\infty \le \gamma \|Q_{tot} - Q'_{tot}\|_\infty \tag{9}$$

*Proof.* Assume the empirical Bellman operator $\mathcal{T}_D^{LVD}$ is a $\gamma$-contraction. For any MMDPs, when using a uniform data distribution, the value function of FQI-LVD will converge (Ernst et al., 2005) because of the contraction of the distance (infinity norm) between any pair of $Q$. However, one counterexample is indicated in Proposition 2, which shows that there exists MMDPs such that, when using a uniform data distribution, the value function of FQI-LVD diverges to infinity from an arbitrary initialization $Q^{(0)}$. The assumption of $\gamma$-contraction is not hold and the empirical Bellman operator $\mathcal{T}_D^{LVD}$ is not a $\gamma$-contraction. $\qquad \square$

**Proposition 2.** *There exist MMDPs such that, when using uniform data distribution, the value function of FQI-LVD diverges to infinity from an arbitrary initialization $Q^{(0)}$.*

*Proof.* We consider the following MMDP with 2 agents, 2 states $(s_1, s_2)$ and each agent $(i = 1, 2)$ has 2 actions $\mathcal{A} \equiv \{\mathcal{A}^{(1)}, \mathcal{A}^{(2)}\}$. The reward function is listed below which $r(s_j, \mathbf{a})$ denotes the reward of $(s_j, \mathbf{a})$, where $\mathbf{a} = \langle a_1, a_2 \rangle$.

$$r(s_1) = \begin{pmatrix} 0 & 0 \\ 0 & 0 \end{pmatrix} \quad r(s_2) = \begin{pmatrix} 1 & 0 \\ 0 & 0 \end{pmatrix} \tag{45}$$

Besides, the transition is deterministic.

$$T(s_1) = \begin{pmatrix} s_1 & s_1 \\ s_1 & s_1 \end{pmatrix} \quad T(s_2) = \begin{pmatrix} s_2 & s_2 \\ s_2 & s_1 \end{pmatrix} \tag{46}$$

Furthermore, $\gamma \in (\frac{4}{5}, 1)$. (In practice, $\gamma$ is usually chosen as $0.99$ or $0.95$.) The following proves that this MMDP will diverge for any initialization.

Denote $Q_i^t(s_j, a_i)$ as the decomposed Q-value of agent $i$ after $t^{\text{th}}$ value-iteration at state $s_j$ with action $a_i$. Then, the total Q-value can be described as $Q_{\text{tot}}^t(s_j, \mathbf{a}) = Q_1^t(s_j, a_1) + Q_2^t(s_j, a_2)$. For brevity, $0^{\text{th}}$ Q-value is its initialization.

First, we clarify the process of each iteration. Since the value-iteration for linear decomposed function class is solving the MSE problem in Lemma 2. $\mathbf{b}$ is target one-step TD-value w.r.t the Q-value of the last iteration. Through described in Lemma 2, the optimal solution of this MSE problem is not unique. We can ignore the term of an arbitrary vector $\mathbf{w}$ when considering the joint action-value functions because $\mathbf{w}$ does not affect the local action selection of each agent and will be eliminated in the summation operator of linear value decomposition. In addition, under uniformed sampling, we observe that $p_u(a_u) = \frac{1}{2}$ for any $\vec{a}, u$. Then, in equation 34

$$-\frac{1}{m} \frac{\mathbf{p}(\vec{a})}{p_u(a_u)} + \frac{1}{mn} \sum_{u'=0}^{n-1} \frac{\mathbf{p}(\vec{a})}{p_{u'}(a_{u'})} = 0 \tag{47}$$

Second, we denote $V_{\text{tot}}^t(s_j) = \max_{\mathbf{a}} Q_{\text{tot}}^t(s_j, \mathbf{a})$ and observe that $\forall t \geq 1, s_j$

$$Q_1^t(s_j, a_1) = \frac{1}{2} \sum_{a_2 \in \mathcal{A}} \left( r(s_j, \mathbf{a}) + \gamma V_{\text{tot}}^{t-1}(T(s_j, \mathbf{a})) \right) - \frac{1}{2} \sum_{\mathbf{a} \in \mathcal{A}} \frac{1}{4} \left( r(s_j, \mathbf{a}) + \gamma V_{\text{tot}}^{t-1}(T(s_j, \mathbf{a})) \right) \tag{48}$$

$$= Q_2^t(s_j, a_2) \tag{49}$$

The second equation holds because the transition $T$ and the reward $R$ are symmetric for both agents. Thus, we omit the subscript of local Q-values as $Q^t(s_j, a)$ when $t \geq 1$.

Third, we analyze the Q-values on state $s_1$. Clearly, its iteration is irrelevant to $s_2$. According to equation 48, $\forall a \in \mathcal{A}, t \geq 1$

$$Q^t(s_1, a) = \frac{\gamma}{2} V_{\text{tot}}^{t-1}(s_1) \tag{50}$$

$$= \frac{\gamma}{2} \max_{a_1, a_2 \in \mathcal{A}} \left( Q^{t-1}(s_1, a_1) + Q^{t-1}(s_1, a_2) \right) \tag{51}$$

Clearly, when $t \geq 1, Q^t\left(s_1, \mathcal{A}^{(1)}\right) = Q^t\left(s_1, \mathcal{A}^{(2)}\right)$. Therefore, we observe that $Q^t(s_1, \cdot) = \gamma^t q_1, \forall t \geq 1$ where $q_1$ is determined by the initialization $Q_{\text{tot}}^0(s_1, \mathbf{a}), \forall \mathbf{a} \in \mathcal{A}$.

Last, we consider state $s_2$. It is straightforward to observe the following recursion for $t \geq 2$ from equation 48

$$
\begin{aligned}
Q^t\left(s_2, \mathcal{A}^{(1)}\right) &= \frac{1}{2}(1 + 2\gamma V_{\text{tot}}^{t-1}(s_2)) - \frac{1}{8}[1 + \gamma(3V_{\text{tot}}^{t-1}(s_2) + V_{\text{tot}}^{t-1}(s_1))] \\
&= \frac{5\gamma}{8}V_{\text{tot}}^{t-1}(s_2) + \frac{3}{8} - \frac{1}{4}\gamma^t q_1 \\
&= \frac{5\gamma}{4}\max_{a \in \mathcal{A}} Q^{t-1}(s_2, a) + \frac{3}{8} - \frac{1}{4}\gamma^t q_1 & (52)
\end{aligned}
$$

$$
\begin{aligned}
Q^t\left(s_2, \mathcal{A}^{(2)}\right) &= \frac{1}{2}(\gamma V_{\text{tot}}^{t-1}(s_2) + \gamma V_{\text{tot}}^{t-1}(s_1)) - \frac{1}{8}[1 + \gamma(3V_{\text{tot}}^{t-1}(s_2) + V_{\text{tot}}^{t-1}(s_1))] \\
&= \frac{\gamma}{8}V_{\text{tot}}^{t-1}(s_2) - \frac{1}{8} + \frac{3}{4}\gamma^t q_1 \\
&= \frac{\gamma}{4}\max_{a \in \mathcal{A}} Q^{t-1}(s_2, a) - \frac{1}{8} + \frac{3}{4}\gamma^t q_1 & (53)
\end{aligned}
$$

We consider some $\delta > 0$ and $t_\delta = \left\lceil \log_\gamma \frac{\delta}{6|q_1|} \right\rceil$. Then, $t > t_\delta$

$$
Q^t\left(s_2, \mathcal{A}^{(2)}\right) \geq \frac{\gamma}{4}\max_{a \in \mathcal{A}} Q^{t-1}(s_2, a) - \frac{1+\delta}{8} \geq \frac{\gamma}{4}Q^{t-1}\left(s_2, \mathcal{A}^{(2)}\right) - \frac{1+\delta}{8} \tag{54}
$$

Denote $\widehat{Q}^t\left(s_2, \mathcal{A}^{(2)}\right) = \frac{\gamma}{4}\widehat{Q}^{t-1}\left(s_2, \mathcal{A}^{(2)}\right) - \frac{1+\delta}{8}, \forall t > t_\delta$ and $\widehat{Q}^{t_\delta}\left(s_2, \mathcal{A}^{(2)}\right) = Q^{t_\delta}\left(s_2, \mathcal{A}^{(2)}\right)$. Consequently, $Q^t(s_2, a_2) \geq \widehat{Q}^{t_\delta}\left(s_2, \mathcal{A}^{(2)}\right), \forall t \geq t_\delta$ by equation 54. Since $t \geq t_\delta$

$$
\widehat{Q}^t\left(s_2, \mathcal{A}^{(2)}\right) = \left(\frac{\gamma}{4}\right)^{t-t_\delta}\left(Q^{t_\delta}\left(s_2, \mathcal{A}^{(2)}\right) - \frac{1+\delta}{2\gamma-8}\right) + \frac{1+\delta}{2\gamma-8} \tag{55}
$$

Furthermore, $\gamma \in (\frac{4}{5}, 1)$. There exists some $T_\delta \geq t_\delta$ which

$$
Q^{T_\delta}\left(s_2, \mathcal{A}^{(2)}\right) \geq \widehat{Q}^{T_\delta}\left(s_2, \mathcal{A}^{(2)}\right) \geq \frac{1+2\delta}{2\gamma-8} > -\frac{1+2\delta}{6} \tag{56}
$$

According to equation 52 and let $\delta < \frac{1}{11}$.

$$
\begin{aligned}
Q^{T_\delta+1}\left(s_2, \mathcal{A}^{(1)}\right) &\geq \frac{5\gamma}{4}Q^{T_\delta}\left(s_2, \mathcal{A}^{(2)}\right) + \frac{3}{8} - \frac{1}{4}\gamma^t q_1 & (57) \\
&> -\frac{5+10\delta}{24} + \frac{3}{8} - \frac{1}{24}\delta & (58) \\
&> \frac{1}{8} & (59)
\end{aligned}
$$

Similar to equation 54, we observer from equation 52 that $\forall t > T_{\delta=\frac{1}{11}} + 1$

$$
Q^t\left(s_2, \mathcal{A}^{(1)}\right) \geq \frac{5\gamma}{4}Q^{t-1}\left(s_2, \mathcal{A}^{(1)}\right) + \frac{1}{4} \tag{60}
$$

and

$$
\begin{aligned}
V_{\text{tot}}^t(s_2) &= 2Q^t\left(s_2, \mathcal{A}^{(1)}\right) & (61) \\
&\geq 2\left(\frac{5\gamma}{4}Q^{t-1}\left(s_2, \mathcal{A}^{(1)}\right) + \frac{1}{4}\right) & (62) \\
&= \frac{5\gamma}{4}V_{\text{tot}}^{t-1}(s_2) + \frac{1}{4} & (63)
\end{aligned}
$$

Since $\frac{5\gamma}{4} > 1$ and the initial point at $T_{\delta=\frac{1}{11}} + 1$ is larger than $\frac{1}{8}$, this suggests that $V_{\text{tot}}^t(s_2)$ will eventually diverge.

Noticing that our proof holds with respect to any $\{Q_{\text{tot}}^0(s_j, \mathbf{a}) | \forall j \in \mathcal{S}, \mathbf{a} \in \mathcal{A}\}$. Thus, value-iteration on linear decomposed function class w.r.t this MDP will diverge evnetually under any circumstances. □

## C    OMITTED ALGORITHM BOX, THEOREM, AND DEFINITION IN SECTION 5.2

### C.1    LOCAL CONVERGENCE IMPROVEMENT

---

**Algorithm 1** On-Policy Fitted Q-Iteration with $\epsilon$-greedy Exploration

---

1: Initialize $Q^{(0)}$.
2: **for** $t = 0 \dots T - 1$ **do**                    $\triangleright$ $T$ denotes the computation budget
3:     Construct an exploratory policy $\tilde{\pi}_t$ based on $Q^{(t)}$.            $\triangleright$ i.e., $\epsilon$-greedy exploration

$$\tilde{\pi}_t(a|s) = \prod_{i=1}^{n} \left( \frac{\epsilon}{|\mathcal{A}|} + (1 - \epsilon)\mathbb{I}\left[ a_i = \arg\max_{a'_i \in \mathcal{A}} Q_i^{(t)}(s, a'_i) \right] \right) \tag{64}$$

4:     Collect a new dataset $D_t$ by running $\tilde{\pi}_t$.
5:     Operate an on-policy Bellman operator $Q^{(t+1)} \leftarrow \mathcal{T}_\epsilon^{\text{LVD}} Q^{(t)} \equiv \mathcal{T}_{D_t}^{\text{LVD}} Q^{(t)}$.

---

Algorithm 1 is a variant of fitted Q-iteration which adopts an on-policy sample distribution. At line 3, an exploratory noise is integrated into the greedy policy, since the function approximator generally requires an extensive set of samples to regularize extrapolation values. Particularly, we investigate a standard exploration module called $\epsilon$-greedy, in which every agent takes a small probability to explore actions with non-maximum values. To make the underlying insights more accessible, we assume the data collection procedure at line 4 can obtain infinite samples, which makes the dataset $D_t$ become a sufficient coverage over the state-action space (see Assumption 1). This algorithmic framework serves as a foundation for discussions on local stability.

We consider an additional assumption stated as follows.

**Assumption 2** (Unique Optimal Policy). *The optimal policy $\pi^*$ is unique.*

The intuitive motivation of this assumption is to have the optimal policy $\pi^*$ be a potential stable solution. In situations where the optimal policy is not unique, most Q-learning algorithms will oscillate around multiple optimal policies (Simchowitz & Jamieson, 2019), and Assumption 2 helps us to rule out these non-interesting cases. Based on this setting, the local stability of FQI-LVD can be characterized by the following lemma.

**Lemma 3.** *There exists a threshold $\delta > 0$ such that the on-policy Bellman operator $\mathcal{T}_\epsilon^{\text{LVD}}$ is closed in the following subspace $\mathcal{B} \subset \mathcal{Q}^{\text{LVD}}$, when the hyper-parameter $\epsilon$ is sufficiently small.*

$$\mathcal{B} = \left\{ Q \in \mathcal{Q}^{\text{LVD}} \;\middle|\; \pi_Q = \pi^*, \; \max_{s \in \mathcal{S}} |Q_{tot}(s, \pi^*(s)) - V^*(s)| \leq \delta \right\}$$

*Formally, $\exists \delta > 0$, $\exists \epsilon > 0$, $\forall Q \in \mathcal{B}$, there must be $\mathcal{T}_\epsilon^{\text{LVD}} Q \in \mathcal{B}$.*

Lemma 3 indicates that once the value function $Q$ steps into the subspace $\mathcal{B}$, the induced policy $\pi_Q$ will converge to the optimal policy $\pi^*$. By combining this local stability with Brouwer's fixed-point theorem (Brouwer, 1911), we can further verify the existence of a fixed-point solution for the on-policy Bellman operator $\mathcal{T}_\epsilon^{\text{LVD}}$ (see Theorem 4).

**Theorem 4** (Formal version of Theorem 2). *Besides Lemma 3, Algorithm 1 will have a fixed point value function expressing the optimal policy if the hyper-parameter $\epsilon$ is sufficiently small.*

Theorem 4 indicates that, multi-agent Q-learning with linear value decomposition has a convergent region, where the value function induces optimal actions. Note that $\mathcal{Q}^{\text{LVD}}$ is a limited function class, which even cannot guarantee to contain the one-step TD target $\mathcal{T}_D^{\text{LVD}} Q$. From this perspective, on-policy data distribution becomes necessary to make the one-step TD target projected to a small set of critical state-action pairs, which help construct the stable subspace $\mathcal{B}$ stated in Lemma 3.

### C.2    GLOBAL CONVERGENCE IMPROVEMENT

**Definition 2** (FQI-IGM). *Given a dataset $D$, FQI-IGM specifies the action-value function class*

$$\mathcal{Q}^{IGM} = \left\{ Q \;\middle|\; Q_{tot} \in \mathbb{R}^{|\mathcal{S}||\mathcal{A}|^n} \text{ and } \left[ \forall Q_i \in \mathbb{R}^{|\mathcal{S}||\mathcal{A}|} \right]_{i=1}^{n} \text{ with that Eq. (1) is satisfied} \right\}. \tag{65}$$

*and induces the empirical Bellman operator*

$$Q^{(t+1)} \leftarrow \mathcal{T}_D^{IGM} Q^{(t)} \equiv \underset{Q \in \mathcal{Q}^{IGM}}{\arg\min} \sum_{(s,\mathbf{a}) \in \mathcal{S} \times \mathbf{A}} p_D(\mathbf{a}|s) \left( y^{(t)}(s,\mathbf{a}) - Q_{tot}(s,\mathbf{a}) \right)^2, \qquad (66)$$

*where $y^{(t)}(s,\mathbf{a}) = r(s,\mathbf{a}) + \gamma \max_{\mathbf{a}'} Q_{tot}^{(t)}(s',\mathbf{a}')$ denotes the regression target derived by Bellman optimality operator. $Q_{tot}$ and $[Q_i]_{i=1}^n$ refer to the interfaces of CTDE defined in Section 3.3.*

Compared with FQI-LVD stated in Definition 1, the differences are the $Q$ function class, i.e, $\mathcal{Q}^{IGM}$ vs. $\mathcal{Q}^{LVD}$.

## D    OMITTED PROOFS OF THEOREM 3

**Lemma 4.** *The empirical Bellman operator $\mathcal{T}_D^{IGM}$ stated in Definition 2 is a $\gamma$-contraction, i.e., the following important property of the standard Bellman optimality operator $\mathcal{T}$ will hold for $\mathcal{T}_D^{IGM}$.*

$$\forall Q_{tot}, Q'_{tot} \in \mathcal{Q}, \quad \|\mathcal{T}Q_{tot} - \mathcal{T}Q'_{tot}\|_\infty \leq \gamma \|Q_{tot} - Q'_{tot}\|_\infty \qquad (67)$$

*Proof.* We want to prove

$$\left( \mathcal{T}_D^{IGM} Q \right)_{tot} = r(s,\mathbf{a}) + \gamma \langle P(s,\mathbf{a}), V_Q \rangle, \qquad (68)$$

where $P$ is transition function, $V_Q(\cdot) = \max_{\mathbf{a} \in \mathbf{A}} Q_{tot}(\cdot, \mathbf{a})$, and $\langle \cdot, \cdot \rangle$ is inner product. According to Eq. (68) and Lemma 1.5 in RL textbook (Agarwal et al., 2019), we can prove that $\mathcal{T}_D^{IGM}$ is a $\gamma$-contraction. Eq. (68) indicates that the empirical Bellman error

$$err_D^{IGM} \equiv \min_{Q \in \mathcal{Q}^{IGM}} \sum_{(s,\mathbf{a}) \in \mathcal{S} \times \mathbf{A}} p_D(\mathbf{a}|s) \left( y^{(t)}(s,\mathbf{a}) - Q_{tot}(s,\mathbf{a}) \right)^2 = 0. \qquad (69)$$

Let $\mathbf{a}^{*,(t)} = \left[ a_i^{*,(t)} \right]_{i=1}^n = \arg\max_{\mathbf{a} \in \mathbf{A}} y^{(t)}(s,\mathbf{a})$. Then, $\forall y^{(t)}(s,\cdot)$, we construct $Q_{tot}(s,\mathbf{a}) = y^{(t)}(s,\mathbf{a})$ and its corresponding local action-value functions $[Q_i]_{i=1}^n$ satisfying IGM principle:

$$Q_i(s, a_i) = \begin{cases} 1, & \text{when } a_i = a_i^{*,(t)}, \\ 0, & \text{when } a_i \neq a_i^{*,(t)}. \end{cases} \qquad (70)$$

To avoid the multiple solutions of $\arg\max$ operator in $\mathbf{a}^{*,(t)}$, we consider the lexicographic order of joint actions as the second priority. Thus, we illustrate the completeness of IGM function class in MMDP setting from our construction. Then, Eq. (68) is held, and $\mathcal{T}_D^{IGM}$ is a $\gamma$-contraction in MMDP framework. $\qquad \square$

**Theorem 3.** *FQI-IGM will globally converge to the optimal value function.*

*Proof.* Let $Q^*(s,\mathbf{a}) = max_{\boldsymbol{\pi} \in \Pi} Q^{\boldsymbol{\pi}}(s,\mathbf{a})$ where $\Pi$ is the space of all policies. According to Lemma 4 and Theorem 1.4 in RL textbook (Agarwal et al., 2019), we have that

- There exists a stationary and deterministic policy $\boldsymbol{\pi}$ such that $Q_{tot}^{\boldsymbol{\pi}} = Q_{tot}^*$.

- A vector $Q_{tot} \in \mathbb{R}^{|\mathcal{S}| \times |\mathcal{A}|^n}$ is equal to $Q_{tot}^*$ if and only if it satisfies $Q_{tot} = \left( \mathcal{T}_D^{IGM} Q \right)_{tot}$.

- $\forall Q'_{tot} \in \mathcal{Q}^{IGM}$,

$$\left\| Q_{tot}^* - \left( \mathcal{T}_D^{IGM} Q' \right)_{tot} \right\|_\infty = \left\| \left( \mathcal{T}_D^{IGM} Q^* \right)_{tot} - \left( \mathcal{T}_D^{IGM} Q' \right)_{tot} \right\|_\infty \qquad (71)$$

$$\leq \gamma \left\| Q_{tot}^* - Q'_{tot} \right\|_\infty. \qquad (72)$$

Thus, FQI-IGM will globally converge to optimal value function. $\qquad \square$

# E   OMITTED PROOFS OF APPENDIX C.1

## E.1   SOME NOTATIONS

In this section, we only consider the data distribution generated by the optimal joint policy $\boldsymbol{\pi}^*$.

To simplify the notations, we use $\varepsilon = \frac{\epsilon}{|\mathcal{A}|}$ to reformulate the exploratory policy generated by $\epsilon$-greedy exploration as follows

$$\tilde{\boldsymbol{\pi}}(\mathbf{a}|s) = \prod_{i=1}^{n} \left( \varepsilon + (1 - \hat{\varepsilon})\mathbb{I}\left[ a_i = \underset{a_i' \in \mathcal{A}}{\arg\max}\, Q_i^*(s, a_i') \right] \right) \tag{73}$$

where $\hat{\varepsilon} = (|\mathcal{A}| - 1)\varepsilon$.

In addition, we use $f(s, \cdot, \cdot)$ to denote the corresponding coefficient in the closed-form updating

$$(\mathcal{T}_D^{\text{LVD}} Q)_{\text{tot}}(s, \mathbf{a}) = \sum_{\mathbf{a}' \in \mathcal{A}^n} f(s, \mathbf{a}, \mathbf{a}')(\mathcal{T}Q)_{\text{tot}}(s, \mathbf{a}') \tag{74}$$

where $(\mathcal{T}Q)_{\text{tot}} = r(s, \mathbf{a}') + \gamma V_{\text{tot}}(s')$ denote the precise target values derived by Bellman optimality equation.

Formally, according to Eq. (8),

$$f(s, \mathbf{a}, \mathbf{a}') = \left( \frac{h^{(1)}(s, \mathbf{a}, \mathbf{a}')}{1 - \hat{\varepsilon}} + \frac{h^{(0)}(s, \mathbf{a}, \mathbf{a}')}{\varepsilon} - (n - 1) \right) (1 - \hat{\varepsilon})^{h^{\boldsymbol{\pi}^*}(s, \mathbf{a}')} \varepsilon^{n - h^{\boldsymbol{\pi}^*}(s, \mathbf{a}')}, \tag{75}$$

in which

$$h^{\boldsymbol{\pi}^*}(s, \mathbf{a}) = \sum_{i=1}^{n} \mathbb{I}[a_i = \pi_i^*(s)] \tag{76}$$

$$h^{(1)}(s, \mathbf{a}, \mathbf{a}') = \sum_{i=1}^{n} \mathbb{I}[a_i = \pi_i^*(s)]\mathbb{I}[a_i = a_i'] \tag{77}$$

$$h^{(0)}(s, \mathbf{a}, \mathbf{a}') = \sum_{i=1}^{n} \mathbb{I}[a_i \neq \pi_i^*(s)]\mathbb{I}[a_i = a_i'] \tag{78}$$

As a reference indicating whether the learned value function produces the optimal policy, we denote

$$\mathcal{E}(Q) = \max_{s \in \mathcal{S}} \left[ \max_{\mathbf{a} \in (\mathcal{A}^n \setminus \{\boldsymbol{\pi}^*(s)\})} (Q_{\text{tot}}(s, \boldsymbol{\pi}^*(s)) - Q_{\text{tot}}(s, \mathbf{a})) \right] \tag{79}$$

Notice that $\boldsymbol{\pi}^*$ denotes the optimal policy of the given MDP, so $\mathcal{E}(Q)$ might be negative for a non-optimal or inaccurate value function $Q$.

## E.2   OMITTED PROOFS

**Lemma 5.** *Given a dataset $D$ generated by the optimal policy $\boldsymbol{\pi}^*$ with $\epsilon$-greedy exploration, for any target value function $Q$,*

$$\forall \delta > 0,\ \forall 0 < \varepsilon \leq \frac{\delta}{n^2 |\mathcal{A}|^n 2^{n+1}(R_{max} + \gamma \|V_{tot}\|_\infty)}, \tag{80}$$

*we have*

$$\forall s \in \mathcal{S},\ \left| (\mathcal{T}_D^{LVD} Q)_{tot}(s, \boldsymbol{\pi}^*(s)) - (\mathcal{T}Q)_{tot}(s, \boldsymbol{\pi}^*(s)) \right| \leq \delta, \tag{81}$$

*where $(\mathcal{T}Q)_{tot}(s, \mathbf{a}) = r(s, \mathbf{a}) + \gamma V_{tot}(s')$ denotes the regression target generated by $Q$.*

*Proof.* $\forall s \in \mathcal{S}$,

$$\left| (\mathcal{T}_D^{\text{LVD}} Q)_{\text{tot}}(s, \boldsymbol{\pi}^*(s)) - (\mathcal{T}Q)_{\text{tot}}(s, \boldsymbol{\pi}^*(s)) \right|$$

$$\leq \left| (f(s, \boldsymbol{\pi}^*(s), \boldsymbol{\pi}^*(s)) - 1)(\mathcal{T}Q)_{\text{tot}}(s, \boldsymbol{\pi}^*(s)) \right| + \left| \sum_{a' \in \mathcal{A}^n \setminus \{\boldsymbol{\pi}^*(s)\}} f(s, \boldsymbol{\pi}^*(s), \mathbf{a}')(\mathcal{T}Q)_{\text{tot}}(s, \mathbf{a}') \right|$$

$$\leq \left( |f(s, \boldsymbol{\pi}^*(s), \boldsymbol{\pi}^*(s)) - 1| + \sum_{a' \in \mathcal{A}^n \setminus \{\boldsymbol{\pi}^*(s)\}} |f(s, \boldsymbol{\pi}^*(s), \mathbf{a}')| \right) \|(\mathcal{T}Q)_{\text{tot}}\|_\infty. \tag{82}$$

In the first term, $\forall s \in \mathcal{S}$,

$$|f(s, \boldsymbol{\pi}^*(s), \boldsymbol{\pi}^*(s)) - 1| = \left| \left( \frac{n}{1 - \hat{\varepsilon}} - (n-1) \right) (1 - \hat{\varepsilon})^n - 1 \right|$$

$$= \left| (n - (n-1)(1 - \hat{\varepsilon}))(1 - \hat{\varepsilon})^{n-1} - 1 \right|$$

$$= \left| (1 + (n-1)\hat{\varepsilon})(1 - \hat{\varepsilon})^{n-1} - 1 \right|$$

$$= \left| (1 + (n-1)\hat{\varepsilon}) \left( \sum_{\ell=0}^{n-1} \binom{n-1}{\ell} (-1)^\ell \hat{\varepsilon}^\ell \right) - 1 \right|$$

$$= \left| (1 + (n-1)\hat{\varepsilon}) \left( 1 - (n-1)\hat{\varepsilon} + \sum_{\ell=2}^{n-1} \binom{n-1}{\ell} (-1)^\ell \hat{\varepsilon}^\ell \right) - 1 \right|$$

$$= \left| 1 - (n-1)^2 \hat{\varepsilon}^2 + (1 + (n-1)\hat{\varepsilon}) \left( \sum_{\ell=2}^{n-1} \binom{n-1}{\ell} (-1)^\ell \hat{\varepsilon}^\ell \right) - 1 \right|$$

$$= \left| \hat{\varepsilon}^2 \left( (n-1)^2 - (1 + (n-1)\hat{\varepsilon}) \sum_{\ell=2}^{n-1} \binom{n-1}{\ell} (-1)^\ell \hat{\varepsilon}^{\ell-2} \right) \right|$$

$$\leq |\mathcal{A}|^2 \varepsilon^2 \left( n^2 + 2 \sum_{\ell=2}^{n-1} \binom{n-1}{\ell} \right)$$

$$\leq |\mathcal{A}|^2 \varepsilon^2 \left( n^2 + 2^n \right)$$

$$\leq \varepsilon^2 n^2 |\mathcal{A}|^2 2^n. \tag{83}$$

In the second term, $\forall s \in \mathcal{S}$,

$$\sum_{a' \in \mathcal{A}^n \setminus \{\boldsymbol{\pi}^*(s)\}} |f(s, \boldsymbol{\pi}^*(s), \mathbf{a}')|$$

$$\leq \sum_{a' \in \mathcal{A}^n \setminus \{\boldsymbol{\pi}^*(s)\}} \left| \left( \frac{h^{\boldsymbol{\pi}^*}(s, \mathbf{a}')}{1 - \hat{\varepsilon}} - (n-1) \right) (1 - \hat{\varepsilon})^{h^{\boldsymbol{\pi}^*}(s, \mathbf{a}')} \varepsilon^{n - h^{\boldsymbol{\pi}^*}(s, \mathbf{a}')} \right|$$

$$= \sum_{a' \in \mathcal{A}^n \setminus \{\boldsymbol{\pi}^*(s)\}} \left| \left( h^{\boldsymbol{\pi}^*}(s, \mathbf{a}') - (n-1)(1 - \hat{\varepsilon}) \right) (1 - \hat{\varepsilon})^{h^{\boldsymbol{\pi}^*}(s, \mathbf{a}') - 1} \varepsilon^{n - h^{\boldsymbol{\pi}^*}(s, \mathbf{a}')} \right|$$

$$\leq \sum_{a' \in \mathcal{A}^n \setminus \{\boldsymbol{\pi}^*(s)\}} \left| 2n(1 - \hat{\varepsilon})^{h^{\boldsymbol{\pi}^*}(s, \mathbf{a}') - 1} \varepsilon^{n - h^{\boldsymbol{\pi}^*}(s, \mathbf{a}')} \right|$$

$$\leq \sum_{a' \in \mathcal{A}^n \setminus \{\boldsymbol{\pi}^*(s)\}} 2n\varepsilon$$

$$\leq 2n\varepsilon |\mathcal{A}|^n. \tag{84}$$

Thus $\forall s \in \mathcal{S}$,

$$
\begin{aligned}
&\left|(\mathcal{T}_D^{\text{LVD}}Q)_{\text{tot}}(s, \boldsymbol{\pi}^*(s)) - (\mathcal{T}Q)_{\text{tot}}(s, \boldsymbol{\pi}^*(s))\right| \\
&\leq \left(|f(s, \boldsymbol{\pi}^*(s), \boldsymbol{\pi}^*(s)) - 1| + \sum_{a' \in \mathcal{A}^n \setminus \{\boldsymbol{\pi}^*(s)\}} |f(s, \boldsymbol{\pi}^*(s), \mathbf{a}')|\right) \|(\mathcal{T}Q)_{\text{tot}}\|_\infty \\
&\leq (\varepsilon^2 n^2 |\mathcal{A}|^2 2^n + 2n\varepsilon |\mathcal{A}|^n) \|(\mathcal{T}Q)_{\text{tot}}\|_\infty \\
&\leq \varepsilon n^2 |\mathcal{A}|^n 2^{n+1} \|(\mathcal{T}Q)_{\text{tot}}\|_\infty \\
&\leq \varepsilon n^2 |\mathcal{A}|^n 2^{n+1} (R_{\max} + \gamma \|V_{\text{tot}}\|_\infty) \\
&\leq \delta.
\end{aligned}
\tag{85}
$$

$\square$

**Lemma 6.** *Given a dataset $D$ generated by the optimal policy $\boldsymbol{\pi}^*$ with $\epsilon$-greedy exploration, for any target value function $Q$,*

$$
\forall 0 < \varepsilon \leq \frac{(1-\gamma)\mathcal{E}(Q^*)}{\gamma n^3 |\mathcal{A}|^n 2^{n+4}(R_{max}/(1-\gamma) + \gamma \|V_{tot}^{\boldsymbol{\pi}^*} - V^*\|_\infty)},
\tag{86}
$$

*we have*

$$
\forall s \in \mathcal{S}, \ \left|(\mathcal{T}_D^{LVD}Q)_{tot}(s, \boldsymbol{\pi}^*(s)) - V^*(s)\right| \leq \gamma \|V_{tot}^{\boldsymbol{\pi}^*} - V^*\|_\infty + \frac{1-\gamma}{8n\gamma}\mathcal{E}(Q^*),
\tag{87}
$$

*where $V_{tot}^{\boldsymbol{\pi}^*}(s) = Q_{tot}(s, \boldsymbol{\pi}^*(s))$.*

*Proof.* $\forall s \in \mathcal{S}$,

$$
\begin{aligned}
&\left|(\mathcal{T}_D^{\text{LVD}}Q)_{\text{tot}}(s, \boldsymbol{\pi}^*(s)) - V^*(s)\right| \\
&\leq \left|(\mathcal{T}_D^{\text{LVD}}Q)_{\text{tot}}(s, \boldsymbol{\pi}^*(s)) - (\mathcal{T}Q)_{\text{tot}}(s, \boldsymbol{\pi}^*(s))\right| + \left|(\mathcal{T}Q)_{\text{tot}}(s, \boldsymbol{\pi}^*(s)) - V^*(s)\right| \\
&= \left|(\mathcal{T}_D^{\text{LVD}}Q)_{\text{tot}}(s, \boldsymbol{\pi}^*(s)) - (\mathcal{T}Q)_{\text{tot}}(s, \boldsymbol{\pi}^*(s))\right| + \left|(\mathcal{T}Q)_{\text{tot}}(s, \boldsymbol{\pi}^*(s)) - Q^*(s, \boldsymbol{\pi}^*(s))\right| \\
&= \left|(\mathcal{T}_D^{\text{LVD}}Q)_{\text{tot}}(s, \boldsymbol{\pi}^*(s)) - (\mathcal{T}Q)_{\text{tot}}(s, \boldsymbol{\pi}^*(s))\right| + \left|(\mathcal{T}Q)_{\text{tot}}(s, \boldsymbol{\pi}^*(s)) - (\mathcal{T}Q^*)(s, \boldsymbol{\pi}^*(s))\right| \\
&\leq \left|(\mathcal{T}_D^{\text{LVD}}Q)_{\text{tot}}(s, \boldsymbol{\pi}^*(s)) - (\mathcal{T}Q)_{\text{tot}}(s, \boldsymbol{\pi}^*(s))\right| + \gamma |V_{\text{tot}}(s') - V^*(s')| \\
&\leq \left|(\mathcal{T}_D^{\text{LVD}}Q)_{\text{tot}}(s, \boldsymbol{\pi}^*(s)) - (\mathcal{T}Q)_{\text{tot}}(s, \boldsymbol{\pi}^*(s))\right| + \gamma |Q_{\text{tot}}(s', \boldsymbol{\pi}^*(s')) - V^*(s')| \\
&\leq \left|(\mathcal{T}_D^{\text{LVD}}Q)_{\text{tot}}(s, \boldsymbol{\pi}^*(s)) - (\mathcal{T}Q)_{\text{tot}}(s, \boldsymbol{\pi}^*(s))\right| + \gamma \|V_{\text{tot}}^{\boldsymbol{\pi}^*} - V^*\|_\infty
\end{aligned}
\tag{88}
$$

Let $\delta = \frac{1-\gamma}{8n\gamma}\mathcal{E}(Q^*)$. According to Lemma 5, with the condition

$$
0 < \varepsilon \leq \frac{\delta}{n^2 |\mathcal{A}|^n 2^{n+1}(R_{\max} + \gamma \|V_{\text{tot}}\|_\infty)} = \frac{(1-\gamma)\mathcal{E}(Q^*)/(8n\gamma)}{n^2 |\mathcal{A}|^n 2^{n+1}(R_{\max} + \gamma \|V_{\text{tot}}\|_\infty)},
\tag{89}
$$

we have

$$
\left|(\mathcal{T}_D^{\text{LVD}}Q)_{\text{tot}}(s, \boldsymbol{\pi}^*(s)) - (\mathcal{T}Q)_{\text{tot}}(s, \boldsymbol{\pi}^*(s))\right| \leq \delta = \frac{1-\gamma}{8n\gamma}\mathcal{E}(Q^*).
\tag{90}
$$

Notice that

$$
\|V_{\text{tot}}\|_\infty \leq \|V^*\|_\infty + \|V_{\text{tot}} - V^*\|_\infty
\tag{91}
$$

$$
\leq \frac{R_{\max}}{1-\gamma} + \|V_{\text{tot}}^{\boldsymbol{\pi}^*} - V^*\|_\infty.
\tag{92}
$$

The overall statement is

$$
\forall 0 < \varepsilon \leq \frac{(1-\gamma)\mathcal{E}(Q^*)}{\gamma n^3 |\mathcal{A}|^n 2^{n+4}(R_{\max}/(1-\gamma) + \gamma \|V_{\text{tot}}^{\boldsymbol{\pi}^*} - V^*\|_\infty)} \leq \frac{(1-\gamma)\mathcal{E}(Q^*)/(8n\gamma)}{n^2 |\mathcal{A}|^n 2^{n+1}(R_{\max} + \gamma \|V_{\text{tot}}\|_\infty)}
\tag{93}
$$

we have $\forall s \in \mathcal{S}$,

$$
\begin{aligned}
&\left|(\mathcal{T}_D^{\mathrm{LVD}}Q)_{\mathrm{tot}}(s, \boldsymbol{\pi}^*(s)) - V^*(s)\right| \\
&\leq \left|(\mathcal{T}_D^{\mathrm{LVD}}Q)_{\mathrm{tot}}(s, \boldsymbol{\pi}^*(s)) - (\mathcal{T}Q)_{\mathrm{tot}}(s, \boldsymbol{\pi}^*(s))\right| + \gamma\|V_{\mathrm{tot}}^{\boldsymbol{\pi}^*} - V^*\|_\infty \\
&\leq \gamma\|V_{\mathrm{tot}}^{\boldsymbol{\pi}^*} - V^*\|_\infty + \frac{1-\gamma}{8n\gamma}\mathcal{E}(Q^*).
\end{aligned}
\tag{94}
$$

$\square$

**Lemma 7.** *For any value function $Q$, the corresponding sub-optimality gap satisfies*

$$
\mathcal{E}(\mathcal{T}Q) \geq \mathcal{E}(Q^*) - 2\gamma\|V_{tot} - V^*\|_\infty
\tag{95}
$$

*Proof.* With a slight abuse of notation, let $s_1$ and $s_2$ denote the next states while taking actions $\boldsymbol{\pi}^*(s)$ and $a$ at the state $s$, respectively. According to the definition,

$$
\begin{aligned}
\mathcal{E}(\mathcal{T}Q) &= \max_{(s,\mathbf{a})\in\mathcal{S}\times(\mathcal{A}^n\setminus\{\boldsymbol{\pi}^*(s)\})} ((\mathcal{T}Q)_{\mathrm{tot}}(s, \boldsymbol{\pi}^*(s)) - (\mathcal{T}Q)_{\mathrm{tot}}(s, \mathbf{a})) \\
&\geq \max_{(s,\mathbf{a})\in\mathcal{S}\times(\mathcal{A}^n\setminus\{\boldsymbol{\pi}^*(s)\})} ((\mathcal{T}Q^*)(s, \boldsymbol{\pi}^*(s)) - (\mathcal{T}Q^*)(s, \mathbf{a}) - \gamma\left(|V_{\mathrm{tot}}(s_1) - V^*(s_1)| + |V_{\mathrm{tot}}(s_2) - V^*(s_2)|\right)) \\
&\geq \max_{(s,\mathbf{a})\in\mathcal{S}\times(\mathcal{A}^n\setminus\{\boldsymbol{\pi}^*(s)\})} ((\mathcal{T}Q^*)(s, \boldsymbol{\pi}^*(s)) - (\mathcal{T}Q^*)(s, \mathbf{a}) - 2\gamma\|V_{\mathrm{tot}} - V^*\|_\infty) \\
&= \max_{(s,\mathbf{a})\in\mathcal{S}\times(\mathcal{A}^n\setminus\{\boldsymbol{\pi}^*(s)\})} (Q^*(s, \boldsymbol{\pi}^*(s)) - Q^*(s, \mathbf{a}) - 2\gamma\|V_{\mathrm{tot}} - V^*\|_\infty) \\
&= \mathcal{E}(Q^*) - 2\gamma\|V_{\mathrm{tot}} - V^*\|_\infty
\end{aligned}
\tag{96}
$$

$\square$

**Lemma 8.** *Given a dataset $D$ generated by the optimal policy $\boldsymbol{\pi}^*$ with $\epsilon$-greedy exploration, for any target value function $Q$,*

$$
\forall \delta > 0, \ \forall 0 < \varepsilon \leq \frac{\delta}{n^2|\mathcal{A}|^n 2^n (R_{max}/(1-\gamma) + \gamma\|V_{tot} - V^*\|_\infty)},
\tag{97}
$$

*we have $\forall s \in \mathcal{S}, \ \forall a \in \mathcal{A}^n \setminus \{\boldsymbol{\pi}^*(s)\}$,*

$$
(\mathcal{T}_D^{LVD}Q)_{tot}(s, \mathbf{a}) \leq (\mathcal{T}Q)_{tot}(s, \boldsymbol{\pi}^*(s)) - \mathcal{E}(Q^*) + 2n\gamma\|V_{tot} - V^*\|_\infty + \delta
\tag{98}
$$

*where $(\mathcal{T}Q)_{tot}(s, \mathbf{a}) = r(s, \mathbf{a}) + \gamma V_{tot}(s')$ denotes the regression target generated by $Q$.*

*Proof.* $\forall s \in \mathcal{S}, \forall a \in \mathcal{A}^n \setminus \{\boldsymbol{\pi}^*(s)\}$,

$$
\begin{aligned}
(\mathcal{T}_D^{\mathrm{LVD}}Q)_{\mathrm{tot}}(s, \mathbf{a}) &= \sum_{a'\in\mathcal{A}^n} f(s, \mathbf{a}, \mathbf{a}')(\mathcal{T}Q)_{\mathrm{tot}}(s, \mathbf{a}') \\
&= f(s, \mathbf{a}, \boldsymbol{\pi}^*(s))(\mathcal{T}Q)_{\mathrm{tot}}(s, \boldsymbol{\pi}^*(s)) \\
&\quad + \sum_{a'\in\mathcal{A}^n : h^{\boldsymbol{\pi}^*}(s,\mathbf{a}')=n-1} f(s, \mathbf{a}, \mathbf{a}')(\mathcal{T}Q)_{\mathrm{tot}}(s, \mathbf{a}') \\
&\quad + \sum_{a'\in\mathcal{A}^n : h^{\boldsymbol{\pi}^*}(s,\mathbf{a}')<n-1} f(s, \mathbf{a}, \mathbf{a}')(\mathcal{T}Q)_{\mathrm{tot}}(s, \mathbf{a}')
\end{aligned}
\tag{99}
$$

In the first term,

$$f(s, \mathbf{a}, \boldsymbol{\pi}^*(s))(\mathcal{T}Q)_{\text{tot}}(s, \boldsymbol{\pi}^*(s))$$

$$= \left( \frac{h^{\boldsymbol{\pi}^*}(s, \mathbf{a})}{1 - \hat{\varepsilon}} - (n - 1) \right) (1 - \hat{\varepsilon})^n (\mathcal{T}Q)_{\text{tot}}(s, \boldsymbol{\pi}^*(s))$$

$$= \left( h^{\boldsymbol{\pi}^*}(s, \mathbf{a}) - (n - 1)(1 - \hat{\varepsilon}) \right) (1 - \hat{\varepsilon})^{n-1} (\mathcal{T}Q)_{\text{tot}}(s, \boldsymbol{\pi}^*(s))$$

$$= \left( h^{\boldsymbol{\pi}^*}(s, \mathbf{a}) - (n - 1) + (n - 1)(|\mathcal{A}| - 1)\varepsilon \right) (1 - \hat{\varepsilon})^{n-1} (\mathcal{T}Q)_{\text{tot}}(s, \boldsymbol{\pi}^*(s))$$

$$\leq \left( h^{\boldsymbol{\pi}^*}(s, \mathbf{a}) - (n - 1) \right) (1 - \hat{\varepsilon})^{n-1} (\mathcal{T}Q)_{\text{tot}}(s, \boldsymbol{\pi}^*(s)) + \varepsilon n |\mathcal{A}| \|(\mathcal{T}Q)_{\text{tot}}\|_\infty$$

$$= \left( h^{\boldsymbol{\pi}^*}(s, \mathbf{a}) - (n - 1) \right) (1 + (1 - \hat{\varepsilon})^{n-1} - 1)(\mathcal{T}Q)_{\text{tot}}(s, \boldsymbol{\pi}^*(s)) + \varepsilon n |\mathcal{A}| \|(\mathcal{T}Q)_{\text{tot}}\|_\infty$$

$$\leq \left( h^{\boldsymbol{\pi}^*}(s, \mathbf{a}) - (n - 1) \right) (\mathcal{T}Q)_{\text{tot}}(s, \boldsymbol{\pi}^*(s)) + \left| h^{\boldsymbol{\pi}^*}(s, \mathbf{a}) - (n - 1) \right| |(1 - \hat{\varepsilon})^{n-1} - 1| \|(\mathcal{T}Q)_{\text{tot}}\|_\infty + \varepsilon n |\mathcal{A}| \|(\mathcal{T}Q)_{\text{tot}}\|_\infty$$

$$\leq \left( h^{\boldsymbol{\pi}^*}(s, \mathbf{a}) - (n - 1) \right) (\mathcal{T}Q)_{\text{tot}}(s, \boldsymbol{\pi}^*(s)) + 2n \left| \sum_{\ell=1}^{n-1} \binom{n-1}{\ell} (-1)^\ell \hat{\varepsilon}^\ell \right| \|(\mathcal{T}Q)_{\text{tot}}\|_\infty + \varepsilon n |\mathcal{A}| \|(\mathcal{T}Q)_{\text{tot}}\|_\infty$$

$$\leq \left( h^{\boldsymbol{\pi}^*}(s, \mathbf{a}) - (n - 1) \right) (\mathcal{T}Q)_{\text{tot}}(s, \boldsymbol{\pi}^*(s)) + 2n\hat{\varepsilon} \left( \sum_{\ell=1}^{n-1} \binom{n-1}{\ell} \right) \|(\mathcal{T}Q)_{\text{tot}}\|_\infty + \varepsilon n |\mathcal{A}| \|(\mathcal{T}Q)_{\text{tot}}\|_\infty$$

$$\leq \left( h^{\boldsymbol{\pi}^*}(s, \mathbf{a}) - (n - 1) \right) (\mathcal{T}Q)_{\text{tot}}(s, \boldsymbol{\pi}^*(s)) + \hat{\varepsilon} n 2^n \|(\mathcal{T}Q)_{\text{tot}}\|_\infty + \varepsilon n |\mathcal{A}| \|(\mathcal{T}Q)_{\text{tot}}\|_\infty$$

$$\leq \left( h^{\boldsymbol{\pi}^*}(s, \mathbf{a}) - (n - 1) \right) (\mathcal{T}Q)_{\text{tot}}(s, \boldsymbol{\pi}^*(s)) + \varepsilon n 2^n |\mathcal{A}| \|(\mathcal{T}Q)_{\text{tot}}\|_\infty + \varepsilon n |\mathcal{A}| \|(\mathcal{T}Q)_{\text{tot}}\|_\infty \tag{100}$$

In the second term,

$$\sum_{a' \in \mathcal{A}^n : h^{\boldsymbol{\pi}^*}(s, \mathbf{a}') = n - 1} f(s, \mathbf{a}, \mathbf{a}')(\mathcal{T}Q)_{\text{tot}}(s, \mathbf{a}')$$

$$= \sum_{a' \in \mathcal{A}^n : h^{\boldsymbol{\pi}^*}(s, \mathbf{a}') = n - 1} \left( \frac{h^{(1)}(s, \mathbf{a}, \mathbf{a}')}{1 - \hat{\varepsilon}} + \frac{h^{(0)}(s, \mathbf{a}, \mathbf{a}')}{\varepsilon} - (n - 1) \right) (1 - \hat{\varepsilon})^{n-1} \varepsilon (\mathcal{T}Q)_{\text{tot}}(s, \mathbf{a}')$$

$$= \sum_{a' \in \mathcal{A}^n : h^{\boldsymbol{\pi}^*}(s, \mathbf{a}') = n - 1} \left( h^{(0)}(s, \mathbf{a}, \mathbf{a}')(1 - \hat{\varepsilon})^{n-1} (\mathcal{T}Q)_{\text{tot}}(s, \mathbf{a}') + \left( \frac{h^{(1)}(s, \mathbf{a}, \mathbf{a}')}{1 - \hat{\varepsilon}} - (n - 1) \right) (1 - \hat{\varepsilon})^{n-1} \varepsilon (\mathcal{T}Q)_{\text{tot}}(s, \mathbf{a}') \right)$$

$$\leq \sum_{a' \in \mathcal{A}^n : h^{\boldsymbol{\pi}^*}(s, \mathbf{a}') = n - 1} \left( h^{(0)}(s, \mathbf{a}, \mathbf{a}')(1 - \hat{\varepsilon})^{n-1} (\mathcal{T}Q)_{\text{tot}}(s, \mathbf{a}') + \left| \frac{h^{(1)}(s, \mathbf{a}, \mathbf{a}')}{1 - \hat{\varepsilon}} - (n - 1) \right| (1 - \hat{\varepsilon})^{n-1} \varepsilon \|(\mathcal{T}Q)_{\text{tot}}\|_\infty \right)$$

$$\leq \sum_{a' \in \mathcal{A}^n : h^{\boldsymbol{\pi}^*}(s, \mathbf{a}') = n - 1} \left( h^{(0)}(s, \mathbf{a}, \mathbf{a}')(1 - \hat{\varepsilon})^{n-1} (\mathcal{T}Q)_{\text{tot}}(s, \mathbf{a}') + 2n\varepsilon \|(\mathcal{T}Q)_{\text{tot}}\|_\infty \right)$$

$$= \sum_{a' \in \mathcal{A}^n : h^{\boldsymbol{\pi}^*}(s, \mathbf{a}') = n - 1} \left( h^{(0)}(s, \mathbf{a}, \mathbf{a}') \left( \sum_{\ell=0}^{n-1} \binom{n-1}{\ell} (-1)^\ell \hat{\varepsilon}^\ell \right) (\mathcal{T}Q)_{\text{tot}}(s, \mathbf{a}') + 2n\varepsilon \|(\mathcal{T}Q)_{\text{tot}}\|_\infty \right)$$

$$= \sum_{a' \in \mathcal{A}^n : h^{\boldsymbol{\pi}^*}(s, \mathbf{a}') = n - 1} \left( h^{(0)}(s, \mathbf{a}, \mathbf{a}') \left( 1 + \sum_{\ell=1}^{n-1} \binom{n-1}{\ell} (-1)^\ell \hat{\varepsilon}^\ell \right) (\mathcal{T}Q)_{\text{tot}}(s, \mathbf{a}') + 2n\varepsilon \|(\mathcal{T}Q)_{\text{tot}}\|_\infty \right)$$

$$\leq \sum_{a' \in \mathcal{A}^n : h^{\boldsymbol{\pi}^*}(s, \mathbf{a}') = n - 1} \left( h^{(0)}(s, \mathbf{a}, \mathbf{a}')(\mathcal{T}Q)_{\text{tot}}(s, \mathbf{a}') + \left| \sum_{\ell=1}^{n-1} \binom{n-1}{\ell} (-1)^\ell \hat{\varepsilon}^\ell \right| \|(\mathcal{T}Q)_{\text{tot}}\|_\infty + 2n\varepsilon \|(\mathcal{T}Q)_{\text{tot}}\|_\infty \right)$$

$$= \sum_{a' \in \mathcal{A}^n : h^{\boldsymbol{\pi}^*}(s, \mathbf{a}') = n - 1} \left( h^{(0)}(s, \mathbf{a}, \mathbf{a}')(\mathcal{T}Q)_{\text{tot}}(s, \mathbf{a}') + \hat{\varepsilon} \left| \sum_{\ell=1}^{n-1} \binom{n-1}{\ell} (-1)^\ell \hat{\varepsilon}^{\ell-1} \right| \|(\mathcal{T}Q)_{\text{tot}}\|_\infty + 2n\varepsilon \|(\mathcal{T}Q)_{\text{tot}}\|_\infty \right)$$

$$\leq \sum_{a' \in \mathcal{A}^n : h^{\boldsymbol{\pi}^*}(s, \mathbf{a}') = n - 1} \left( h^{(0)}(s, \mathbf{a}, \mathbf{a}')(\mathcal{T}Q)_{\text{tot}}(s, \mathbf{a}') + \hat{\varepsilon} \left( \sum_{\ell=1}^{n-1} \binom{n-1}{\ell} \right) \|(\mathcal{T}Q)_{\text{tot}}\|_\infty + 2n\varepsilon \|(\mathcal{T}Q)_{\text{tot}}\|_\infty \right)$$

$$\leq \sum_{a' \in \mathcal{A}^n : h^{\boldsymbol{\pi}^*}(s, \mathbf{a}') = n - 1} \left( h^{(0)}(s, \mathbf{a}, \mathbf{a}')(\mathcal{T}Q)_{\text{tot}}(s, \mathbf{a}') + \varepsilon |\mathcal{A}| 2^{n-1} \|(\mathcal{T}Q)_{\text{tot}}\|_\infty + 2n\varepsilon \|(\mathcal{T}Q)_{\text{tot}}\|_\infty \right)$$

$$= \left( \sum_{a' \in \mathcal{A}^n : h^{\boldsymbol{\pi}^*}(s, \mathbf{a}') = n - 1} h^{(0)}(s, \mathbf{a}, \mathbf{a}')(\mathcal{T}Q)_{\text{tot}}(s, \mathbf{a}') \right) + \varepsilon n |\mathcal{A}| 2^{n-1} \|(\mathcal{T}Q)_{\text{tot}}\|_\infty + 2n^2 \varepsilon \|(\mathcal{T}Q)_{\text{tot}}\|_\infty$$

$$\leq \left( \sum_{a' \in \mathcal{A}^n : h^{\boldsymbol{\pi}^*}(s, \mathbf{a}') = n - 1} h^{(0)}(s, \mathbf{a}, \mathbf{a}')(\mathcal{T}Q)_{\text{tot}}(s, \mathbf{a}') \right) + \varepsilon n^2 |\mathcal{A}| 2^n \|(\mathcal{T}Q)_{\text{tot}}\|_\infty \tag{101}$$

In the third term,

$$\sum_{a'\in\mathcal{A}^n:h^{\boldsymbol{\pi}^*}(s,\mathbf{a}')<n-1} f(s,\mathbf{a},\mathbf{a}')(\mathcal{T}Q)_{\text{tot}}(s,\mathbf{a}')$$

$$\leq \sum_{a'\in\mathcal{A}^n:h^{\boldsymbol{\pi}^*}(s,\mathbf{a}')<n-1} |f(s,\mathbf{a},\mathbf{a}')(\mathcal{T}Q)_{\text{tot}}(s,\mathbf{a}')|$$

$$= \sum_{a'\in\mathcal{A}^n:h^{\boldsymbol{\pi}^*}(s,\mathbf{a}')<n-1} \left|\frac{h^{(1)}(s,\mathbf{a},\mathbf{a}')}{1-\hat{\varepsilon}} + \frac{h^{(0)}(s,\mathbf{a},\mathbf{a}')}{\varepsilon} - (n-1)\right| (1-\hat{\varepsilon})^{h^{\boldsymbol{\pi}^*}(s,\mathbf{a}')}\varepsilon^{n-h^{\boldsymbol{\pi}^*}(s,\mathbf{a}')} \left|(\mathcal{T}Q)_{\text{tot}}(s,\mathbf{a}')\right|$$

$$\leq \sum_{a'\in\mathcal{A}^n:h^{\boldsymbol{\pi}^*}(s,\mathbf{a}')<n-1} \left|\frac{h^{(1)}(s,\mathbf{a},\mathbf{a}')}{1-\hat{\varepsilon}} + \frac{h^{(0)}(s,\mathbf{a},\mathbf{a}')}{\varepsilon} + (n-1)\right| (1-\hat{\varepsilon})^{h^{\boldsymbol{\pi}^*}(s,\mathbf{a}')}\varepsilon^{n-h^{\boldsymbol{\pi}^*}(s,\mathbf{a}')} \left|(\mathcal{T}Q)_{\text{tot}}(s,\mathbf{a}')\right|$$

$$\leq \sum_{a'\in\mathcal{A}^n:h^{\boldsymbol{\pi}^*}(s,\mathbf{a}')<n-1} n\left(1+\frac{1}{1-\hat{\varepsilon}}+\frac{1}{\varepsilon}\right)(1-\hat{\varepsilon})^{h^{\boldsymbol{\pi}^*}(s,\mathbf{a}')}\varepsilon^{n-h^{\boldsymbol{\pi}^*}(s,\mathbf{a}')} \left|(\mathcal{T}Q)_{\text{tot}}(s,\mathbf{a}')\right|$$

$$\leq \sum_{a'\in\mathcal{A}^n:h^{\boldsymbol{\pi}^*}(s,\mathbf{a}')<n-1} n\left(1+\frac{2}{\varepsilon}\right)(1-\hat{\varepsilon})^{h^{\boldsymbol{\pi}^*}(s,\mathbf{a}')}\varepsilon^{n-h^{\boldsymbol{\pi}^*}(s,\mathbf{a}')} \left|(\mathcal{T}Q)_{\text{tot}}(s,\mathbf{a}')\right|$$

$$\leq \sum_{a'\in\mathcal{A}^n:h^{\boldsymbol{\pi}^*}(s,\mathbf{a}')<n-1} 3n\varepsilon^{n-h^{\boldsymbol{\pi}^*}(s,\mathbf{a}')-1} \left|(\mathcal{T}Q)_{\text{tot}}(s,\mathbf{a}')\right|$$

$$\leq \sum_{a'\in\mathcal{A}^n:h^{\boldsymbol{\pi}^*}(s,\mathbf{a}')<n-1} 3n\varepsilon\|(\mathcal{T}Q)_{\text{tot}}\|_\infty$$

$$\leq 3n\varepsilon|\mathcal{A}|^n\|(\mathcal{T}Q)_{\text{tot}}\|_\infty \tag{102}$$

Combining the above terms, we can get

$$(\mathcal{T}_D^{\text{LVD}}Q)_{\text{tot}}(s,\mathbf{a})$$
$$= f(s,\mathbf{a},\boldsymbol{\pi}^*(s))(\mathcal{T}Q)_{\text{tot}}(s,\boldsymbol{\pi}^*(s)) + \sum_{a'\in\mathcal{A}^n:h^{\boldsymbol{\pi}^*}(s,\mathbf{a}')=n-1} f(s,\mathbf{a},\mathbf{a}')(\mathcal{T}Q)_{\text{tot}}(s,\mathbf{a}')$$
$$+ \sum_{a'\in\mathcal{A}^n:h^{\boldsymbol{\pi}^*}(s,\mathbf{a}')<n-1} f(s,\mathbf{a},\mathbf{a}')(\mathcal{T}Q)_{\text{tot}}(s,\mathbf{a}')$$
$$\leq \left(h^{\boldsymbol{\pi}^*}(s,\mathbf{a})-(n-1)\right)(\mathcal{T}Q)_{\text{tot}}(s,\boldsymbol{\pi}^*(s)) + \varepsilon n 2^n|\mathcal{A}|\|(\mathcal{T}Q)_{\text{tot}}\|_\infty + \varepsilon n|\mathcal{A}|\|(\mathcal{T}Q)_{\text{tot}}\|_\infty$$
$$+ \left(\sum_{a'\in\mathcal{A}^n:h^{\boldsymbol{\pi}^*}(s,\mathbf{a}')=n-1} h^{(0)}(s,\mathbf{a},\mathbf{a}')(\mathcal{T}Q)_{\text{tot}}(s,\mathbf{a}')\right) + \varepsilon n^2|\mathcal{A}|2^n\|(\mathcal{T}Q)_{\text{tot}}\|_\infty + 3n\varepsilon|\mathcal{A}|^n\|(\mathcal{T}Q)_{\text{tot}}\|_\infty$$
$$\leq \left(h^{\boldsymbol{\pi}^*}(s,\mathbf{a})-(n-1)\right)(\mathcal{T}Q)_{\text{tot}}(s,\boldsymbol{\pi}^*(s)) + \left(\sum_{a'\in\mathcal{A}^n:h^{\boldsymbol{\pi}^*}(s,\mathbf{a}')=n-1} h^{(0)}(s,\mathbf{a},\mathbf{a}')(\mathcal{T}Q)_{\text{tot}}(s,\mathbf{a}')\right)$$
$$+ \varepsilon n^2|\mathcal{A}|^n 2^n\|(\mathcal{T}Q)_{\text{tot}}\|_\infty \tag{103}$$

in which

$$\sum_{a'\in\mathcal{A}^n:h^{\boldsymbol{\pi}^*}(s,\mathbf{a}')=n-1} h^{(0)}(s,\mathbf{a},\mathbf{a}')(\mathcal{T}Q)_{\text{tot}}(s,\mathbf{a}')$$

$$\leq \left(\sum_{a'\in\mathcal{A}^n:h^{\boldsymbol{\pi}^*}(s,\mathbf{a}')=n-1} h^{(0)}(s,\mathbf{a},\mathbf{a}')\right) \max_{a'\in\mathcal{A}^n:h^{\boldsymbol{\pi}^*}(s,\mathbf{a}')=n-1}(\mathcal{T}Q)_{\text{tot}}(s,\mathbf{a}')$$

$$= (n-h^{\boldsymbol{\pi}^*}(s,\mathbf{a})) \max_{a'\in\mathcal{A}^n:h^{\boldsymbol{\pi}^*}(s,\mathbf{a}')=n-1}(\mathcal{T}Q)_{\text{tot}}(s,\mathbf{a}')$$

$$\leq (n-h^{\boldsymbol{\pi}^*}(s,\mathbf{a})) \max_{a'\in\mathcal{A}^n\setminus\{\boldsymbol{\pi}^*(s)\}}(\mathcal{T}Q)_{\text{tot}}(s,\mathbf{a}')$$

$$= (n-h^{\boldsymbol{\pi}^*}(s,\mathbf{a}))\left((\mathcal{T}Q)_{\text{tot}}(s,\boldsymbol{\pi}^*)-\mathcal{E}(\mathcal{T}Q)\right) \tag{104}$$

Thus $\forall s \in \mathcal{S}, \forall \mathbf{a} \in \mathcal{A}^n \setminus \{\boldsymbol{\pi}^*(s)\}$,

$$(\mathcal{T}_D^{\text{LVD}} Q)_{\text{tot}}(s, \mathbf{a})$$

$$\leq \left(h^{\boldsymbol{\pi}^*}(s, \mathbf{a}) - (n-1)\right)(\mathcal{T}Q)_{\text{tot}}(s, \boldsymbol{\pi}^*(s)) + \left(\sum_{\mathbf{a}' \in \mathcal{A}^n : h^{\boldsymbol{\pi}^*}(s, \mathbf{a}') = n-1} h^{(0)}(s, \mathbf{a}, \mathbf{a}')(\mathcal{T}Q)_{\text{tot}}(s, \mathbf{a}')\right)$$

$$+ \varepsilon n^2 |\mathcal{A}|^n 2^n \|(\mathcal{T}Q)_{\text{tot}}\|_\infty$$

$$\leq \left(h^{\boldsymbol{\pi}^*}(s, \mathbf{a}) - (n-1)\right)(\mathcal{T}Q)_{\text{tot}}(s, \boldsymbol{\pi}^*(s)) + (n - h^{\boldsymbol{\pi}^*}(s, \mathbf{a}))\left((\mathcal{T}Q)_{\text{tot}}(s, \boldsymbol{\pi}^*) - \mathcal{E}(\mathcal{T}Q)\right) + \varepsilon n^2 |\mathcal{A}|^n 2^n \|(\mathcal{T}Q)_{\text{tot}}\|_\infty$$

$$= (\mathcal{T}Q)_{\text{tot}}(s, \boldsymbol{\pi}^*(s)) - (n - h^{\boldsymbol{\pi}^*}(s, \mathbf{a}))\mathcal{E}(\mathcal{T}Q) + \varepsilon n^2 |\mathcal{A}|^n 2^n \|(\mathcal{T}Q)_{\text{tot}}\|_\infty \tag{105}$$

According to Lemma 7, $\mathcal{E}(\mathcal{T}Q) \geq \mathcal{E}(Q^*) - 2\gamma \|V_{\text{tot}} - V^*\|_\infty$. So $\forall s \in \mathcal{S}, \forall \mathbf{a} \in \mathcal{A}^n \setminus \{\boldsymbol{\pi}^*(s)\}$,

$$\begin{aligned}
(\mathcal{T}_D^{\text{LVD}} Q)_{\text{tot}}(s, \mathbf{a}) &\leq (\mathcal{T}Q)_{\text{tot}}(s, \boldsymbol{\pi}^*(s)) - (n - h^{\boldsymbol{\pi}^*}(s, \mathbf{a}))\mathcal{E}(\mathcal{T}Q) + \varepsilon n^2 |\mathcal{A}|^n 2^n \|(\mathcal{T}Q)_{\text{tot}}\|_\infty \\
&\leq (\mathcal{T}Q)_{\text{tot}}(s, \boldsymbol{\pi}^*(s)) - (n - h^{\boldsymbol{\pi}^*}(s, \mathbf{a}))\left(\mathcal{E}(Q^*) - 2\gamma\|V_{\text{tot}} - V^*\|_\infty\right) + \varepsilon n^2 |\mathcal{A}|^n 2^n \|(\mathcal{T}Q)_{\text{tot}}\|_\infty \\
&\leq (\mathcal{T}Q)_{\text{tot}}(s, \boldsymbol{\pi}^*(s)) - \mathcal{E}(Q^*) + 2n\gamma\|V_{\text{tot}} - V^*\|_\infty + \varepsilon n^2 |\mathcal{A}|^n 2^n \|(\mathcal{T}Q)_{\text{tot}}\|_\infty \\
&\leq (\mathcal{T}Q)_{\text{tot}}(s, \boldsymbol{\pi}^*(s)) - \mathcal{E}(Q^*) + 2n\gamma\|V_{\text{tot}} - V^*\|_\infty + \varepsilon n^2 |\mathcal{A}|^n 2^n (R_{\max} + \gamma\|V_{\text{tot}}\|_\infty) \\
&\leq (\mathcal{T}Q)_{\text{tot}}(s, \boldsymbol{\pi}^*(s)) - \mathcal{E}(Q^*) + 2n\gamma\|V_{\text{tot}} - V^*\|_\infty + \varepsilon n^2 |\mathcal{A}|^n 2^n (R_{\max} + \gamma\|V^*\|_\infty + \gamma\|V_{\text{tot}} - V^*\|_\infty) \\
&\leq (\mathcal{T}Q)_{\text{tot}}(s, \boldsymbol{\pi}^*(s)) - \mathcal{E}(Q^*) + 2n\gamma\|V_{\text{tot}} - V^*\|_\infty + \varepsilon n^2 |\mathcal{A}|^n 2^n (R_{\max}/(1-\gamma) + \gamma\|V_{\text{tot}} - V^*\|_\infty) \\
&\leq (\mathcal{T}Q)_{\text{tot}}(s, \boldsymbol{\pi}^*(s)) - \mathcal{E}(Q^*) + 2n\gamma\|V_{\text{tot}} - V^*\|_\infty + \delta
\end{aligned} \tag{106}$$

$\square$

**Lemma 9.** *Let $\mathcal{B}$ denote a subspace of value functions*

$$\mathcal{B} = \left\{ Q \in \mathcal{Q}^{LVD} \,\middle|\, \mathcal{E}(Q) \geq 0, \ \|V_{tot} - V^*\|_\infty \leq \frac{1}{8n\gamma}\mathcal{E}(Q^*) \right\} \tag{107}$$

*Given a dataset $D$ generated by the optimal policy $\boldsymbol{\pi}^*$ with $\epsilon$-greedy exploration,*

$$\forall 0 < \varepsilon \leq \frac{(1-\gamma)\mathcal{E}(Q^*)}{n^3 |\mathcal{A}|^n 2^{n+4}(R_{max}/(1-\gamma) + \mathcal{E}(Q^*)/(8n))} \tag{108}$$

*we have $\forall Q \in \mathcal{B}, \mathcal{T}_D^{LVD} Q \in \hat{\mathcal{B}} \subset \mathcal{B}$ where*

$$\hat{\mathcal{B}} = \left\{ Q \in \mathcal{Q}^{LVD} \,\middle|\, \mathcal{E}(Q) > 0, \ \|V_{tot} - V^*\|_\infty \leq \frac{1}{8n\gamma}\mathcal{E}(Q^*) \right\} \tag{109}$$

*Proof.* According to Lemma 5, with the condition

$$0 < \varepsilon \leq \frac{\mathcal{E}(Q^*)/4}{n^2 |\mathcal{A}|^n 2^{n+1}(R_{\max}/(1-\gamma) + \mathcal{E}(Q^*)/(8n))} \leq \frac{\mathcal{E}(Q^*)/4}{n^2 |\mathcal{A}|^n 2^{n+1}(R_{\max} + \gamma\|V_{\text{tot}}\|_\infty)} \tag{110}$$

we have $\forall Q \in \mathcal{B}, \forall s \in \mathcal{S}$,

$$\left|(\mathcal{T}_D^{\text{LVD}} Q)_{\text{tot}}(s, \boldsymbol{\pi}^*(s)) - (\mathcal{T}Q)_{\text{tot}}(s, \boldsymbol{\pi}^*(s))\right| \leq \frac{1}{4}\mathcal{E}(Q^*) \tag{111}$$

which implies $\forall Q \in \mathcal{B}, \forall s \in \mathcal{S}$,

$$(\mathcal{T}_D^{\text{LVD}} Q)_{\text{tot}}(s, \boldsymbol{\pi}^*(s)) \geq (\mathcal{T}Q)_{\text{tot}}(s, \boldsymbol{\pi}^*(s)) - \frac{1}{4}\mathcal{E}(Q^*). \tag{112}$$

According to Lemma 8, with the condition

$$\begin{aligned}
0 < \varepsilon &\leq \frac{\mathcal{E}(Q^*)/4}{n^2 |\mathcal{A}|^n 2^n (R_{\max}/(1-\gamma) + \mathcal{E}(Q^*)/(8n))} \\
&\leq \frac{\mathcal{E}(Q^*)/4}{n^2 |\mathcal{A}|^n 2^n (R_{\max}/(1-\gamma) + \gamma\|V_{\text{tot}} - V^*\|_\infty)}
\end{aligned} \tag{113}$$

we have $\forall Q \in \mathcal{B}, \forall s \in \mathcal{S}, \forall a \in \mathcal{A}^n \setminus \{\boldsymbol{\pi}^*(s)\}$,

$$(\mathcal{T}_D^{\text{LVD}}Q)_{\text{tot}}(s, \mathbf{a}) \leq (\mathcal{T}Q)_{\text{tot}}(s, \boldsymbol{\pi}^*(s)) - \mathcal{E}(Q^*) + 2n\gamma\|V_{\text{tot}} - V^*\|_\infty + \frac{1}{4}\mathcal{E}(Q^*)$$

$$\leq (\mathcal{T}Q)_{\text{tot}}(s, \boldsymbol{\pi}^*(s)) - \mathcal{E}(Q^*) + \frac{1}{4}\mathcal{E}(Q^*) + \frac{1}{4}\mathcal{E}(Q^*)$$

$$= (\mathcal{T}Q)_{\text{tot}}(s, \boldsymbol{\pi}^*(s)) - \frac{1}{2}\mathcal{E}(Q^*)$$

$$< (\mathcal{T}_D^{\text{LVD}}Q)_{\text{tot}}(s, \boldsymbol{\pi}^*(s)) \tag{114}$$

which implies $\mathcal{E}(\mathcal{T}_D^{\text{LVD}}Q) > 0$.

According to Lemma 6, with the condition

$$0 < \varepsilon \leq \frac{(1-\gamma)\mathcal{E}(Q^*)}{\gamma n^3|\mathcal{A}|^n 2^{n+4}(R_{\max}/(1-\gamma) + \mathcal{E}(Q^*)/(8n))}$$

$$\leq \frac{(1-\gamma)\mathcal{E}(Q^*)}{\gamma n^3|\mathcal{A}|^n 2^{n+4}(R_{\max}/(1-\gamma) + \gamma\|V_{\text{tot}}^{\boldsymbol{\pi}^*} - V^*\|_\infty)}, \tag{115}$$

we have $\forall Q \in \mathcal{B}, \forall s \in \mathcal{S}$,

$$\left|(\mathcal{T}_D^{\text{LVD}}V)(s) - V^*(s)\right| = \left|(\mathcal{T}_D^{\text{LVD}}Q)_{\text{tot}}(s, \boldsymbol{\pi}^*(s)) - V^*(s)\right| \tag{116}$$

$$\leq \gamma\|V_{\text{tot}}^{\boldsymbol{\pi}^*} - V^*\|_\infty + \frac{1-\gamma}{8n\gamma}\mathcal{E}(Q^*) \leq \frac{1}{8n\gamma}\mathcal{E}(Q^*). \tag{117}$$

Combing Eq. (110), (113), and (115), the overall condition is

$$0 < \varepsilon \leq \frac{(1-\gamma)\mathcal{E}(Q^*)}{n^3|\mathcal{A}|^n 2^{n+4}(R_{\max}/(1-\gamma) + \mathcal{E}(Q^*)/(8n))} \tag{118}$$

$\square$

**Lemma 3.** *There exists a threshold $\delta > 0$ such that the on-policy Bellman operator $\mathcal{T}_\epsilon^{LVD}$ is closed in the following subspace $\mathcal{B} \subset \mathcal{Q}^{LVD}$, when the hyper-parameter $\epsilon$ is sufficiently small.*

$$\mathcal{B} = \left\{ Q \in \mathcal{Q}^{LVD} \ \middle|\ \boldsymbol{\pi}_Q = \boldsymbol{\pi}^*, \ \max_{s \in \mathcal{S}}|Q_{tot}(s, \boldsymbol{\pi}^*(s)) - V^*(s)| \leq \delta \right\}$$

*Formally, $\exists \delta > 0, \exists \epsilon > 0, \forall Q \in \mathcal{B}$, there must be $\mathcal{T}_\epsilon^{LVD}Q \in \mathcal{B}$.*

*Proof.* It is implied by Lemma 9. $\square$

**Theorem 4** (Formal version of Theorem 2). *Besides Lemma 3, Algorithm 1 will have a fixed point value function expressing the optimal policy if the hyper-parameter $\epsilon$ is sufficiently small.*

*Proof.* Notice that the state value function is sufficient to determine the target values, so the subspace $\mathcal{B}$ defined in Lemma 9 is a compact and convex space in terms of $V_{\text{tot}}$. The operator $\mathcal{T}_D^{\text{LVD}}$ is a continuous mapping because it only involves elementary functions. According to Brouwer's Fixed Point Theorem (Brouwer, 1911), there exist $Q \in \mathcal{B}$ satisfying $\mathcal{T}_D^{\text{LVD}}Q \in \mathcal{B}$. In addition, according to the definition stated in Eq. (109), the fixed point must represent the unique optimal policy since it cannot lie on the boundary with $\mathcal{E}(Q) = 0$. $\square$

## F EXPERIMENT SETTINGS AND IMPLEMENTATION DETAILS

### F.1 IMPLEMENTATION DETAILS

We adopt the PyMARL (Samvelyan et al., 2019) implementation with default hyper-parameters to investigate state-of-the-art multi-agent Q-learning algorithms: VDN (Sunehag et al., 2018), QMIX (Rashid et al., 2018), QTRAN (Son et al., 2019), and QPLEX (Wang et al., 2020a). The training time of these algorithms on an NVIDIA RTX 2080TI GPU is about 4 hours to 12 hours, which

| Map Name | Replay Buffer Size | Behaviour Test Win Rate | Behaviour Policy |
|----------|--------------------|-------------------------|------------------|
| 2s3z | 20k episodes | 91.2% | VDN |
| 3s5z | 20k episodes | 77.5% | VDN |
| 2s_vs_1sc | 20k episodes | 99.6% | VDN |
| 3s_vs_5z | 20k episodes | 94.2% | VDN |
| 1c3s5z | 30k episodes | 92.1% | VDN |
| 3c7z | 30k episodes | 94.4% | VDN |
| 5m_vs_6m | 50k episodes | 61.7% | VDN |
| 10m_vs_11m | 50k episodes | 88.7% | VDN |
| 3h_vs_4z | 50k episodes | 83.1% | VDN |

Table 2: The dataset configurations of offline data collection setting.

is depended on the number of agents and the episode length limit of each map. The performance measure of StarCraft II tasks is the percentage of episodes in which RL agents defeat all enemy units within the limited time constraints, called *test win rate*. The dataset providing off-policy exploration is constructed by training a behavior policy of VDN and collecting its 20k, 30k or 50k experienced episodes. The dataset configurations are shown in Table 2. We investigate five multi-agent Q-learning algorithms over 6 random seeds, which includes 3 different datasets and evaluates two seeds on each dataset. We train 300 epochs to evaluate the learning performance with a given static dataset, of which 32 episodes are trained in each update, and 160k transitions are trained for each epoch totally. Moreover, the training process of behavior policy is the same as that discussed in PyMARL (Samvelyan et al., 2019), which has collected a total of 2 million timestep data and anneals the hyper-parameter $\epsilon$ of $\epsilon$-greedy exploration strategy linearly from 1.0 to 0.05 over 50k timesteps. The target network will be updated periodically after training every 200 episodes. We call this period of 200 episodes an *Iteration*, which corresponds to an iteration of FQI-LVD (see Definition 1).

### F.2 Two-State MMDP

In the two-state MMDP shown in Figure 1a, due to the GRU-based implementation of the finite-horizon paradigm in the above five deep multi-agent Q-learning algorithms, we assume that two agents starting from state $s_2$ have 100 environmental steps executed by a uniform $\epsilon$-greedy exploration strategy (*i.e.*, $\epsilon = 1$). We use this long-term horizon pattern and uniform $\epsilon$-greedy exploration methods to approximate an infinite-horizon MMDP paradigm with uniform data distribution. We adopt $\gamma = 0.9$ to implement FQI-LVD and deep MARL algorithms. In the FQI-LVD framework, $V_{max} = \frac{1}{1-\gamma} = 100$ as shown in Figure 1b. Figure 1c demonstrates that *Optimal* line is approximately $\sum_{i=0}^{99} \gamma^i = 63.4$ in one episode of 100 timesteps.

### F.3 StarCraft II

StarCraft II unit micromanagement tasks consider a combat game of two groups of agents, where StarCraft II takes built-in AI to control enemy units, and MARL algorithms can control each ally unit to fight the enemies. Units in two groups can contain different types of soldiers, but these soldiers in the same group should belong to the same race. The action space of each agent includes no-op, move [direction], attack [enemy id], and stop. At each timestep, agents choose to move or attack in continuous maps. MARL agents will get a global reward equal to the amount of damage done to enemy units. Moreover, killing one enemy unit and winning the combat will bring additional bonuses of 10 and 200, respectively. The maps of SMAC challenges in this paper are introduced in Table 3 in the episodes of 100 timesteps.

| Map Name | Ally Units | Enemy Units |
|----------|-----------|-------------|
| 2s3z | 2 Stalkers & 3 Zealots | 2 Stalkers & 3 Zealots |
| 3s5z | 3 Stalkers & 5 Zealots | 3 Stalkers & 5 Zealots |
| 2s_vs_1sc | 2 Stalkers | 1 Spine Crawler |
| 3s_vs_5z | 3 Stalkers | 5 Zealots |
| 1c3s5z | 1 Colossus, 3 Stalkers & 5 Zealots | 1 Colossus, 3 Stalkers & 5 Zealots |
| 3c7z | 3 Colossi & 7 Zealots | 3 Colossi & 7 Zealots |
| 5m_vs_6m | 5 Marines | 6 Marines |
| 10m_vs_11m | 10 Marines | 11 Marines |
| 3h_vs_4z | 3 Hydralisks | 4 Zealots |

Table 3: SMAC challenges.

# G    DEFERRED TABLES AND FIGURES IN SECTION 6

## G.1    THE LEARNING CURVE OF TABLE 1C

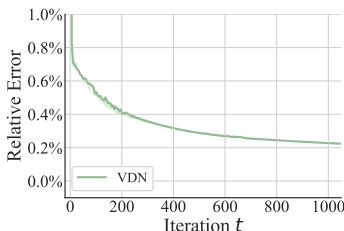

Figure 3: The learning curve of Table 1c. Every iteration contains 200 gradient steps. The relative error is defined as $\|Q_{\text{tot}}^{\text{FQI-LVD}} - Q_{\text{tot}}^{\text{VDN}}\|_\infty / \|Q_{\text{tot}}^{\text{FQI-LVD}}\|_\infty$.

## G.2    DEFERRED TABLES IN SECTION 6.1

| $a_2$ \ $a_1$ | $\mathcal{A}^{(1)}$ | $\mathcal{A}^{(2)}$ | $\mathcal{A}^{(3)}$ |
|------|------|------|------|
| $\mathcal{A}^{(1)}$ | **7.98** | -12.09 | -12.10 |
| $\mathcal{A}^{(2)}$ | -12.18 | -0.02 | -0.02 |
| $\mathcal{A}^{(3)}$ | -12.11 | -0.03 | -0.03 |

(a) $Q_{\text{tot}}$ of QPLEX

| $a_2$ \ $a_1$ | $\mathcal{A}^{(1)}$ | $\mathcal{A}^{(2)}$ | $\mathcal{A}^{(3)}$ |
|------|------|------|------|
| $\mathcal{A}^{(1)}$ | **8.00** | -12.00 | -12.00 |
| $\mathcal{A}^{(2)}$ | -12.00 | -0.00 | 0.00 |
| $\mathcal{A}^{(3)}$ | -12.00 | 0.00 | 0.00 |

(b) $Q_{\text{tot}}$ of QTRAN

| $a_2$ \ $a_1$ | $\mathcal{A}^{(1)}$ | $\mathcal{A}^{(2)}$ | $\mathcal{A}^{(3)}$ |
|------|------|------|------|
| $\mathcal{A}^{(1)}$ | -7.98 | -7.98 | -7.98 |
| $\mathcal{A}^{(2)}$ | -7.98 | -0.00 | **-0.00** |
| $\mathcal{A}^{(3)}$ | -7.98 | -0.00 | -0.00 |

(c) $Q_{\text{tot}}$ of QMIX

Table 4: (a-c) Joint action-value functions $Q_{\text{tot}}$ of QPLEX, QTRAN, and QMIX. Boldface means the greedy joint action selection from $Q_{\text{tot}}$.

# H  ABLATION STUDIES ON NETWORK CAPACITY

## H.1  ABLATION STUDIES IN MATRIX GAME

| $a_2$ \ $a_1$ | $\mathcal{A}^{(1)}$ | $\mathcal{A}^{(2)}$ | $\mathcal{A}^{(3)}$ |
|---|---|---|---|
| $\mathcal{A}^{(1)}$ | **8** | -12 | -12 |
| $\mathcal{A}^{(2)}$ | -12 | 0 | 0 |
| $\mathcal{A}^{(3)}$ | -12 | 0 | 0 |

| $a_2$ \ $a_1$ | $\mathcal{A}^{(1)}$ | $\mathcal{A}^{(2)}$ | $\mathcal{A}^{(3)}$ |
|---|---|---|---|
| $\mathcal{A}^{(1)}$ | **7.98** | -12.09 | -12.10 |
| $\mathcal{A}^{(2)}$ | -12.18 | -0.02 | -0.02 |
| $\mathcal{A}^{(3)}$ | -12.11 | -0.03 | -0.03 |

| $a_2$ \ $a_1$ | $\mathcal{A}^{(1)}$ | $\mathcal{A}^{(2)}$ | $\mathcal{A}^{(3)}$ |
|---|---|---|---|
| $\mathcal{A}^{(1)}$ | **8.00** | -12.00 | -12.00 |
| $\mathcal{A}^{(2)}$ | -12.00 | -0.00 | 0.00 |
| $\mathcal{A}^{(3)}$ | -12.00 | 0.00 | 0.00 |

(a) Payoff of matrix game  (b) $Q_{\text{tot}}$ of QPLEX  (c) $Q_{\text{tot}}$ of QTRAN

| $a_2$ \ $a_1$ | $\mathcal{A}^{(1)}$ | $\mathcal{A}^{(2)}$ | $\mathcal{A}^{(3)}$ |
|---|---|---|---|
| $\mathcal{A}^{(1)}$ | -6.24 | -4.90 | -4.90 |
| $\mathcal{A}^{(2)}$ | -4.90 | **-3.57** | -3.57 |
| $\mathcal{A}^{(3)}$ | -4.90 | -3.57 | -3.57 |

| $a_2$ \ $a_1$ | $\mathcal{A}^{(1)}$ | $\mathcal{A}^{(2)}$ | $\mathcal{A}^{(3)}$ |
|---|---|---|---|
| $\mathcal{A}^{(1)}$ | -8.03 | -8.03 | -8.03 |
| $\mathcal{A}^{(2)}$ | -8.03 | -0.01 | **-0.01** |
| $\mathcal{A}^{(3)}$ | -8.03 | -0.01 | -0.01 |

(d) $Q_{\text{tot}}$ of Large-VDN  (e) $Q_{\text{tot}}$ of Large-QMIX

Table 5: (a-c) The ground-truth payoff matrix and the joint action-value functions of QPLEX and QTRAN. (d-e) The joint action-value functions $Q_{\text{tot}}$ of Large-VDN and Large-QMIX. Boldface means the greedy joint action selection from $Q_{\text{tot}}$.

To address the concern that QPLEX naturally uses more hidden parameters than VDN and QMIX, which may also improve its representational capacity. To demonstrate that the performance gap between QPLEX and other methods does not come from the difference in term of the number of parameters, we increase the number of neurons in VDN and QMIX so that they have comparable number of parameters as QPLEX. Formally, Large-VDN and Large-QMIX have similar number of parameters as QPLEX. The experiment results are presented in Table 5, both the "Large-" versions of VDN and QMIX cannot represent an accurate value function in this matrix game. Increasing the number of parameters cannot address the limitations of VDN and QMIX on representational capacity.

## H.2  ABLATION STUDIES IN STARCRAFT II BENCHMARK TASKS

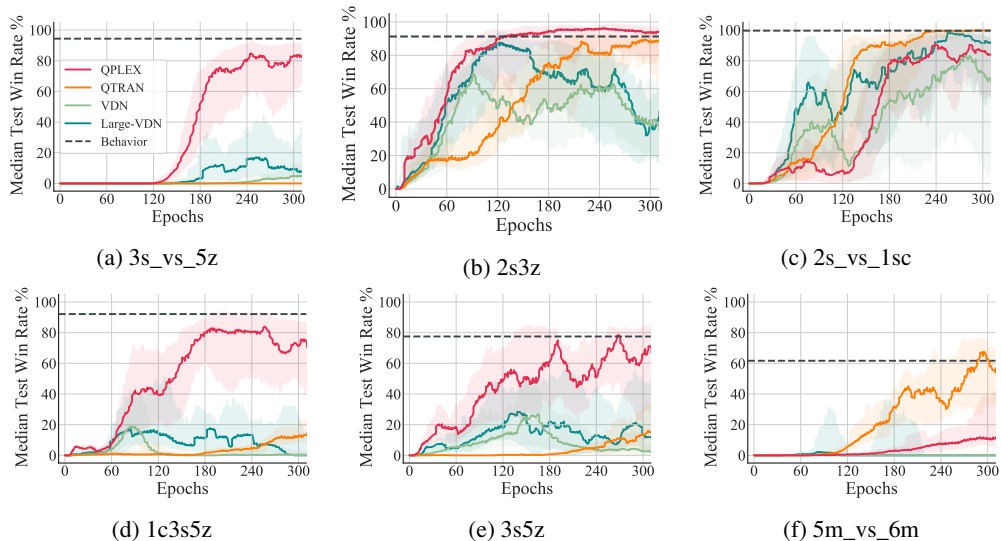

Figure 4: Evaluating the performance of Large-VDN with a given static dataset.

In addition to the ablation study in the matrix game, Figure 4 and Figure 5 present the ablation studies in StarCraft II benchmark tasks with offline data collection. In comparison to the standard versions of VDN and QMIX, we introduce Large-VDN and Large-QMIX which have similar number of parameters as QPLEX. As shown in Figure 4, increasing parameters can benefit VDN in some

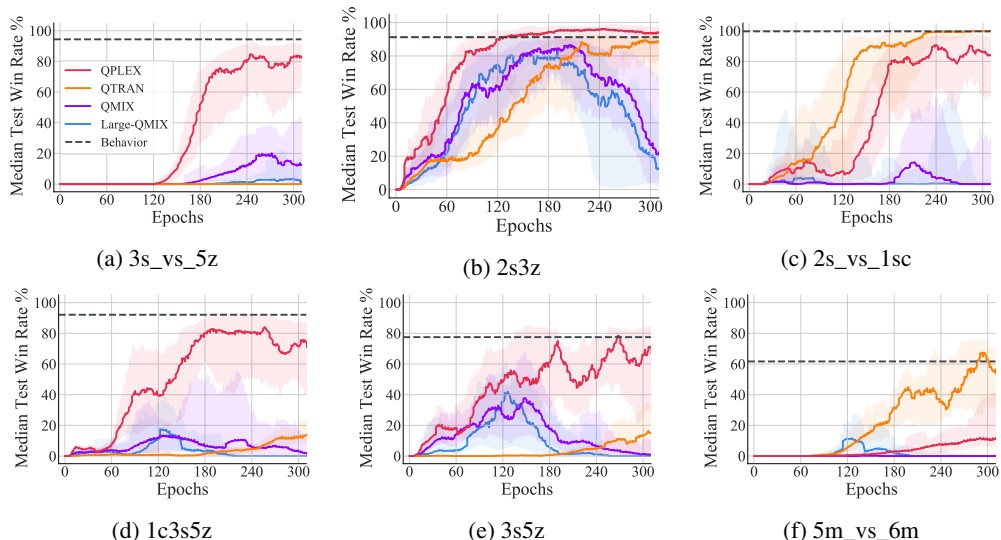

Figure 5: Evaluating the performance of Large-QMIX with a given static dataset.

easy maps such as 2s3z and 2s_vs_1sc, but it cannot provide fundamental improvement in harder tasks. As shown in Figure 5, the effects of increasing parameters are rather weak for QMIX. These experiments demonstrate that increasing the number of parameters cannot address the limitations of VDN and QMIX on representational capacity.

# I  ADDITIONAL EXPERIMENTS ON MMDP EXAMPLE

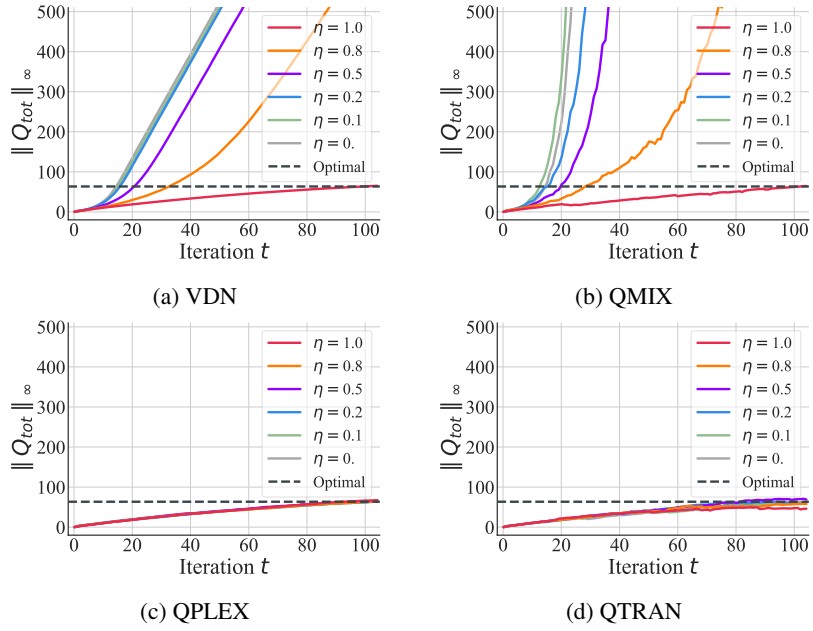

Figure 6: The learning curves of $\|Q_{\text{tot}}\|_\infty$ while running several deep multi-agent Q-learning algorithms with an unfactorizable dataset.

**Remark**  Assumption 1 on the factorizable dataset does not require the factorizability of the underlying transition and reward functions or the decomposability of the joint action-value function. On the contrary, all our theorems and examples focus on the situations where the joint Q-function

cannot be perfectly factorized. In addition, Assumption 1 can be naturally satisfied when the dataset is collected by decentralized execution of agents' policies, e.g., an on-policy dataset collected using $\epsilon$-greedy exploration policies or an offline dataset collected by given decentralized policies of agents. All algorithms discussed in paper, including VDN, QMIX, QTRAN, and QPLEX, learn decentralized policies, which are executed in a decentralized manner. The theoretical implications derived in this paper are applicable whenever such factorizable data collection procedures are carried out.

To investigate the dependency of our theoretical implications on Assumption 1, we provide an experiment to evaluate the performance of deep multi-agent Q-learning algortihms on unfactorizable datasets. Figure 6 present the learning curves of VDN, QMIX, QPLEX, and QTRAN in the example MMDP shown in Figure 1a with an unfactorizable dataset $D$ constructed by a parameter $\eta$ as follows:

$$\forall s \in \mathcal{S}, \quad p_D(\mathcal{A}^{(1)}, \mathcal{A}^2 \mid s) = \begin{pmatrix} 0.5\eta + 0.25(1 - \eta) & 0.25(1 - \eta) \\ 0.25(1 - \eta) & 0.5\eta + 0.25(1 - \eta) \end{pmatrix}.$$

As shown in Figure 6, the choice of parameter $\eta$ has no impacts on the performance of QPLEX and QTRAN, which matches the fact that Theorem 3 does not rely on the assumption of factorizable dataset. As the extension of Proposition 2, VDN and QMIX empirically suffer from unbounded divergence when the dataset is not factorizable. The only exception is the case of $\eta = 1$, in which the dataset only contains two kinds of joint actions. In this case, the given MMDP degenerates to a single-agent MDP because agents only perform the same actions in the dataset. As a result, VDN and QMIX would not diverge in this special situation.

