# OpenReview forum: "Towards Understanding Linear Value Decomposition in Cooperative Multi-Agent Q-Learning"
_ICLR.cc/2021/Conference — Reject_

### Official Review · AnonReviewer2 · 2020-10-25
**Interesting paper but has issues.**

**Rating:** 5
**Confidence:** 4

**Review:**

This paper analyzes the behavior of Bellman update in cooperative multi-agent setting, when the value function has the form of a linear value decomposition of individual Q function per agent. The paper assumes no partial observability (i.e., all agents can see all states) and shows that under deterministic dynamics and factorizable Dataset, after one update, the individual Q function has close form, building connection with COMA. The paper then shows with this function class, the Bellman update might diverge in some MMDP. By extending to a broader function class, the Bellman update is close again.

The theory proposed in the paper looks interesting. I appreciate that the authors make an attempt to create a theory to analyze how the value decomposition fits to the Bellman equation, which is a fundamental question to ask. It can be a good contribution to the research community. However, there are several questions:

Questions:

The assumption of factorizable data and no partial observability seems to be super strong. What is the reason behind the two assumptions? If we remove them, to how much extent you conclusion still holds?

I don’t see very strong connections between the theoretical analysis and the empirical studies. Why “|Q_tot^{FQI-LVD} - Q_tot^{VDN}| = 0.22 strongly illustrates the accuracy of Theorem 1”? Shouldn’t we compare the two terms with relative error? Note that FQI-IGM basically removes the constraint of linear value decomposition. It looks like QMix is in FQI-IGM (the Q combination is nonlinear) and it should work according to the theory, but why it doesn’t work in Matrix Game?

The StarCraft experiment is kind of confusing. How is it connected to the theories? Basically you just show methods with more rich neural network models work better, which usually holds without the theory. Note that the dataset is constructed with a pre-trained policy (by VDN), I am not sure whether the factorizable assumption holds here? Did you check? It looks like a lot of ablation studies are needed to make the connection clear.

=========

After rebuttal I will still keep the score.

The Assumption 1 is still quite strong. While it is true that $p(\mathbf{a}|s) = \prod_i p(a_i|s)$ holds for any policy with decentralized execution, the assumption of full observability is really strong and it doesn't seem that it can be got rid of easily. With this assumption, $p(\mathbf{a}|s) = \prod_i p(a_i|s)$ is actually a trivial assumption, since all information needed to make a decision of action $a_i$ for agent $i$ is already contained in the full state $s$ and can be determined independent of each other. In the revised paper, the authors suggest that communication can solve it but this would require thorough communication over the entire MDP, which can be hard to achieve (and if that's easily achievable then there is no need to study Dec-POMDP anymore). Without relaxing this assumption, I have concerns that the theory is not substantial (which are also concerns from other reviewers like R3). One baby step is to at least assume each agent may receive a noisy version of the full state $s$, and see what's going on.

I thank the authors for additional experiments. Note that in addition to the proposed theory, there are many possible explanations of the empirical results presented by the authors. E.g., as a general rule of thumb in RL training, using offline data is often worse than using online data. Without detailed analysis, it is hard to tell. I would rather use an environment that is more complicated than the matrix game, but much more simpler than SMAC, which is partial observable and has too many moving parts.

---

> ### Author Response · Authors · 2020-11-21
> **Response to Reviewer 2 (Part 1)**
>
> Thanks for your comments. In the following Q\&A, we provide clarification to your questions and concerns. We appreciate if you have any further questions or comments.
>
> **Q1**: The assumptions of deterministic dynamics and factorizable dataset.
>
> The assumption of deterministic dynamics has been released in the latest revision. By introducing Lemma 1 in Appendix A, our theoretical results can generalize to stochastic environments.
>
> The assumption of a factorizable dataset is quite gentle, especially for MARL with the paradigm of centralized training with decentralized execution. When the dataset is collected by decentralized execution of agents' fixed policies, e.g., on-policy ($\epsilon$-greedy exploration) data collection or offline dataset with given multi-agent decentralized policies, it can be naturally factorized. As we know, VDN, QMIX, QTRAN, and QPLEX all learn decentralized policies, which are executed in a decentralized manner.
>
> When the dataset is not factorizable, Theorem 1 (implicit counterfactual credit assignment) might not hold, but Theorem 3 (global convergence of FQI-IGM) still holds. This theorem implies that QPLEX, the deep-learning-based implementation of FQI-IGM, is an excellent choice in the arbitrary data collection procedure.
>
> In Appendix I of the revision, we included an additional experiment on the didactic two-state MMDP with an unfactorizable dataset. The experiment shows that VDN and QMIX still suffer from unbounded divergence when the dataset is no longer factorizable. In comparison, the excellent performance of QTRAN and QPLEX is not sensitive to the data distribution, which matches our argument that Theorem 3 (global convergence of FQI-IGM) does not rely on the factorizability assumption.
>
> **Q2**: The assumption of full observability.
>
> This work studies learning in infinite-horizon multi-agent decision-making problems. As infinite-horizon Dec-POMDPs is undecidable in general (Madani et al., 1999), it is sensible to start a MARL formal analysis with full observability. In practice, partial observability is not a hard constraint. When agents can only access their local observations, learning communication is a common approach to reducing partial observability issues. e.g., NDQ (Wang et al., 2020) aims to learn an efficient communication protocol that transmits useful and necessary information to address partial observability. Our results can provide insights into those application scenarios with communications.
>
> We included a discussion of these considerations in the preliminary section of the latest revision. Please refer to the introduction of MMDP in Section 2 for more detailed discussions.
>
> **Q3**: I don’t see very strong connections between the theoretical analysis and the empirical studies. Why “|Q\_tot\^{FQI-LVD} - Q\_tot\^{VDN}| = 0.22 strongly illustrates the accuracy of Theorem 1”? Shouldn’t we compare the two terms with relative error?
>
> The experiments in Section 6.1 are set to verify the theoretical findings in practical deep MARL algorithms. The reviewer is right that it is more sensible to compare with relative error. The absolute error of 0.22 is equivalent to 3.7\% relative error, and the expected value is over 6. In the latest revision, we fixed a minor bug in the simulation of VDN on the matrix game. The value decomposition derived by VDN nearly matches the theoretical analysis (reaching absolute error of 0.01 or 0.2\% relative error after 1000 iterations). This close match implies that the theoretical result of Theorem 1 for linear value decomposition can generalize to its practical deep MARL counterpart. Please refer to Table 1 in Section 6.1 and Figure 3 in Appendix G.
>
> Empirical studies on a more complex MARL domain (i.e., StarCraft II benchmark) in Section 6.2 also have strong connections to theoretical results. Please refer to the discussion in Q5 below.
>
> [1] *Madani O, Hanks S, Condon A. On the undecidability of probabilistic planning and infinite-horizon partially observable Markov decision problems. InAAAI/IAAI 1999 Jul 18 (pp. 541-548).*
>
> [2] *Wang T, Wang J, Zheng C, et al. Learning nearly decomposable value functions via communication minimization. International Conference on Learning Representations, 2020.*

---

> > ### Author Response · Authors · 2020-11-21
> > **Response to Reviewer 2 (Part 2)**
> >
> > **Q4**: It looks like QMix is in FQI-IGM (the Q combination is nonlinear) and it should work according to the theory, but why it doesn’t work in Matrix Game?
> >
> > Although the Q combination of QMIX is nonlinear, the monotonic value decomposition of QMIX is not equivalent to, but a proper subset of $Q^{IGM}$ used by FQI-IGM. Please refer to Section 5.2 of the latest revision for details.
> >
> > The IGM function class contains all action-value functions that satisfy the consistency of maximum-value actions between global and local value functions. The monotonic mixing network of QMIX only implements a proper subset of $Q^{IGM}$. The matrix game presented in this paper is not a monotonic game. Due to its limited representation capability, QMIX will fit its local action-value functions to the sub-optimal solutions to minimize empirical Bellman errors with uniform data distribution. As a result, QMIX fails in this matrix game. This limitation of QMIX has been discussed in previous work and becomes a major motivation of many advanced methods such as QPLEX, QTRAN, and MAVEN (Mahajan et al., 2019).
> >
> > **Q5**: The StarCraft experiment is kind of confusing. How is it connected to the theories?
> >
> > The main purpose of conducting StarCraft experiments is not to verify the correctness of the proposed theories, but to investigate whether the proposed theoretical implications can generalize to practical problems.
> >
> > In Section 6.2, we generalize the experimental setting to a complex MARL domain (i.e., StarCraft II benchmark) and investigate two common data collection settings, i.e., online and offline data collection. By comparing the performance of VDN with online and offline data collection procedures, we empirically demonstrate two implications derived from the theoretical understanding (see Implication 2 in Section 5): 1) VDN with online data collection has superior performance over that of offline data collection. 2) VDN (linear value decomposition) has considerable limitations in offline training, which can be addressed by improving representational capacity as QTRAN and QPLEX.
> >
> > **Q6**: More rich neural network models work better.
> >
> > The "richness" of the factorized joint action-value function class mentioned in this paper is different from the term "network capacity" (e.g., the number of layers and neurons) in other deep learning applications. As discussed in Q4, the joint value function class of QMIX is a strict subset of that of QPLEX, even if both of them use an infinite number of neurons and layers. Our experiments in Section 5 and 6 are conducted to investigate the effects of the structural representational capacity of deep MARL methods on their performance.
> >
> > In Appendix H of the revision, we included an additional ablation study on the number of parameters used by VDN and QMIX. This experiment indicates two points: (1) In matrix games, both VDN and QMIX still cannot represent the payoff matrix after increasing the number of neurons. (2) In StarCraft benchmark tasks, increasing the number of neurons also cannot help VDN or QMIX achieve comparable performance to QPLEX. These results support our claim that the critical difference between QMIX and QPLEX is their structural representational capacity rather than the number of network parameters.
> >
> > **Q7**: Note that the dataset is constructed with a pre-trained policy (by VDN), I am not sure whether the factorizable assumption holds here?
> >
> > As justified in Q1, when pre-trained decentralized policies of agents collect the dataset (e.g., learned by VDN), it can be naturally factorized. The factorizable dataset's distribution-based assumption is appropriate modeling of deep MARL algorithms with centralized training and decentralized execution, including VDN, QMIX, QTRAN, and QPLEX. However, we do not intend to restrict this assumption exactly in StarCraft experiments since the practical problem has only finite samples for a high-dimensional continuous state space. As discussed in Q5, these experiments' motivation is to demonstrate that the implications derived under mild assumptions (i.e., factorizable dataset) can generalize to practical problems in deep MARL.
> >
> > [3] *Mahajan A, Rashid T, Samvelyan M, et al. Maven: Multi-agent variational exploration. Advances in Neural Information Processing Systems. 2019: 7613-7624.*

---

### Official Review · AnonReviewer1 · 2020-10-28
**interesting paper analyzing the success and failures of current value decomposition schemes**

**Rating:** 6
**Confidence:** 4

**Review:**

This paper is largely a theoretical undertaking and focuses on bringing new insights into the currently popular value decomposition schemes like VDN, QMIX etc. for multi-agent reinforcement learning. They find two major implications: 1) linear value decomposition leads to implicit difference based credit assignment, and 2) richer Q function classes like those of QTRAN, QPLEX can improve convergence because otherwise there is a risk of unbounded divergence.

The paper was an enjoyable read (although there are a few grammar errors here and there).
Instead of the theoretically more complicated online RL problem, it uses the offline RL setting to analyze various properties of value decomposition schemes, especially the possibility of them not being a proper $\gamma$-contraction.

There are some minor issues in the presentation. In fig 1b, the $\epsilon$ term hasn't been defined. One can guess that it's likely the $\epsilon$-greedy policy for data collection, but greek variables should be clarified before they are used in general.

Given QTRAN should have a harder time enforcing IGM constraint, it's a bit surprising in Fig 2, that it works so much better than QPLEX on the harder problems. Any speculations into why?

It would be useful to also draw connections to coordination graphs and their pairwise and higher order interactions of value functions (like that of DCG for example [1]).

[1] https://arxiv.org/abs/1910.00091

Post discussion: I'll defer to R2 and R4 for judging the theoretical contributions and am convinced that they are not as significant.

---

> ### Author Response · Authors · 2020-11-21
> **Response to Reviewer 1**
>
> Thanks for your inspiring comments. We appreciate if you have any further questions or comments.
>
> **Q1**: The comparison between QTRAN and QPLEX.
>
> QPLEX outperforms QTRAN in most tasks because it uses duplex dueling architecture to realize a strict constraint of IGM, while QTRAN's soft constraint may not be able to enforce IGM. On occasion, QPLEX may suffer from optimization instability since it implements the advantage-reference decomposition with the reference's maximum value. In this variant of dueling architecture, the learned advantage function needs to update simultaneously when the value of the optimal action changes. This issue has been observed and discussed by Dueling DQN (Wang et al., 2016) and QPLEX itself.
>
> **Q2**: The connection to coordination graphs.
>
> Thanks for your suggestion on discussing coordination graphs (Böhmer et al., 2020) as a supplementary to the related literature.
>
> The coordination graph is a different framework from the algorithms discussed in this paper. This paper focuses on the paradigm of centralized training with decentralized execution. The methodology of coordination graphs is not in this case since it allows collaborated action-selection through several rounds of communications and interactions. Meanwhile, the value factorization used by coordination graphs does not follow the principle of IGM consistency, which is its main difference between coordination graphs and VDN, QMIX, QTRAN.
>
> Despite these differences, the coordination graph is indeed a promising method for multi-agent reinforcement learning. Compared with linear value decomposition, coordination graphs use a higher-order value decomposition structure in the view of function approximation. This higher-order value factorization provides a different way to address linear value decomposition limitations on representational capacity. It is an interesting and important future direction to study and understand this higher-order value decomposition structure's algorithmic properties.
> We included this discussion in Section 7 of the latest revision.
>
> [1] *Wang Z, Schaul T, Hessel M, et al. Dueling network architectures for deep reinforcement learning. International conference on machine learning. 2016: 1995-2003.*
>
> [2] *Böhmer W, Kurin V, Whiteson S. Deep coordination graphs. In International Conference on Machine Learning 2020.*

---

### Official Review · AnonReviewer4 · 2020-10-29
**Interesting work**

**Rating:** 5
**Confidence:** 5

**Review:**

The theoretical understanding of MARL with linear value decomposition is limited at the moment. Due to the limited representation of LVD (linear value decomposition), the standard Bellman update is not a closed operator in the joint action-value function class with LVD.

This paper is inspired by the current limit of the understanding and it provides a series of theoretical characterizations. The first result is that the updated Q_i(s, a_i) can be expressed by the decomposed target (equation (8)) up to some term in s with assumptions on discrete state and action spaces. The analysis is mostly based on weighted linear regression. This theorem has a moderate implication that the decomposed target is a counterfactual credit assignment. The second result is that the update (7) is closed, even with a sufficiently small eps for eps-greedy. Without eps this result is obvious, but with an eps it requires some tweak to analyze it. The third result is that if the linear decomposition is replaced with consistency on argmax (likewise in QMIX). This result is more or less intuitive and obvious.

The greatest contribution of this paper lies in that it proposes a closed-form solution to the Bellman error minimization derived in FQI-LVD up to some term in s. Implication 1 is rather obvious, and Implication 2 is an interesting explanation of the current limitation of linear decomposition. As the paper does not propose a new algorithmic approach, these theoretical implications can be limited. I believe that the information included in this paper might not be very sufficient.

Pros:

1. this paper is of clear logic. For example, a). the research problems are well explained with strong motivation. b). The results and conclusions are solid verified by comparing the performance of different algorithms.
2. the findings are meaningful. The closed-form solution can serve as a powerful mathematical toolkit, which encourages follow-up profound theories and explores potential insights from different perspectives.
3. this paper is theoretically and mathematically rigorous. It gave more than 20 pages of proofs for nearly the lemmas and theorems

Questions:

1. The author didn't interpret the credit assignment part very well when discussing the closed-form solution to Bellman error minimization. For example, the paper states that one of the contributions is that it implicitly implements a classical multi-agent credit assignment. However, in the literature review part, VDN also realized a multi-agent credit assignment. Is there an overlap? Or, is there a connection or difference between the credit assignment of the two solutions? Also, this paper gave examples of how the credit assignment was done when all joint actions generate the same reward signals. What if joint actions generate different rewards (and obviously this case is more close to reality)? How would the closed-form solution assign credits intuitively? I think the author should make more elaborations here.
2. On page 7, the table shows a cooperative game. From table(a), we can see that the optimal policy is both agents take A1 simultaneously. However, in Table (b) and (c), both FQI-LVD and VDN algorithm choose the optimal policy of A2. I wonder why the two algorithms fail to find the optimal policy here?

---

> ### Author Response · Authors · 2020-11-21
> **Response to Reviewer 4 (Part 1)**
>
> Thanks for your comments. In the following Q\&A, we provide clarification to your questions and concerns. We appreciate if you have any further questions or comments.
>
> **Q1**: Implication 1 is rather obvious.
>
> We are curious why the reviewer thinks that Implication 1 is obvious. To our best knowledge, this paper presents the first result revealing what is the credit assignment mechanism that deep multi-agent Q-learning with linear value decomposition (e.g., VDN) realizes. Without our proposed Theorem 1, the credit assignment mechanism of VDN used to be regarded as a black box. Please also refer to Q3.
>
> **Q2**: As the paper does not propose a new algorithmic approach, these theoretical implications can be limited.
>
> We have to disagree with the reviewer that every paper needs to propose a new algorithmic approach. A paper can be significant and important for a research area if it provides formal and empirical insights on why existing algorithms work and when they do not work. This is exactly what this work aims for.
>
> This work introduces formal multi-agent FQI frameworks and reveals how some recent deep MARL algorithms (e.g., VDN) works and when they may not work. Our theoretical results can provide insights into developing better practical algorithms. This paper also conducts an extensive empirical study toward verifying the derived theoretical implications in deep MARL algorithms (see Section 6). These empirical results indicate a notable limitation of VDN and QMIX, i.e., their performance might be limited in offline settings. Regarding this issue, our analysis can provide two potential solutions, i.e., using on-policy training and using a richer function class (e.g., by QTRAN/QPLEX), whose positive impacts on learning performance have also been demonstrated in experiments.
>
> **Q3**: The author didn't interpret the credit assignment part very well when discussing the closed-form solution to Bellman error minimization. VDN also realized a multi-agent credit assignment. Is there an overlap? Or, is there a connection or difference between the credit assignment of the two solutions?
>
>
> The reviewer misunderstood the relation between FQI-LVD and VDN. This paper does not implement a new credit assignment by introducing the FQI-LVD framework. Instead, the purpose of FQI-LVD is to analyze VDN, e.g., revealing what its underlying credit assignment mechanism is and what is its convergence property.
>
> For details, as we discussed in Section 3.2, VDN can be regarded as a deep-learning-based implementation of FQI-LVD. The credit assignment mechanism of VDN is represented by the hidden parameters of a neural network, which is end-to-end learned through back-propagation from TD-learning on the joint Q-function. Theorem 1 provides a formal and intuitive interpretation of this black-box credit assignment realized by VDN, i.e., a popular heuristic counterfactual credit assignment mechanism used by COMA. This result helps us understand why VDN works in some challenging coordination tasks.
>
> **Q4**: This paper gave examples of how the credit assignment was done when all joint actions generate the same reward signals. What if joint actions generate different rewards (and obviously this case is more close to reality)? How would the closed-form solution assign credits intuitively?
>
> As discussed in two paragraphs following Theorem 1, the closed-form solution reveals that multi-agent linear value decomposition (e.g., VDN) realizes a counterfactual credit assignment. The purpose of using the example, where all joint-action generate the same reward signals, is to easily illustrate why this counterfactual credit assignment mechanism is similar to but better than the counterfactual credit assignment used by COMA.
>
> As discussed in Q3, FQI-LVD and VDN share the same counterfactual credit assignment mechanism stated as the closed-form solution in Theorem 1. With a counterfactual credit assignment, the credit assigned to an agent's action is proportional to its advantage value over a baseline policy, where the baseline policy corresponds to the state-action distribution in the dataset. Please refer to Eq.(8) and the related discussions in the paper.
>
> Regarding the question "What if joint actions generate different rewards," the reward function setting does not affect the comparison with COMA credit assignment. As discussed in the paper, the only difference between the credit assignment used by VDN and COMA is the coefficient of a state-dependent term. This state-dependent term is the expected value of the baseline policy (see Eq.(8)), which does not affect the relative credit among actions or agents.

---

> > ### Author Response · Authors · 2020-11-21
> > **Response to Reviewer 4 (Part 2)**
> >
> > **Q5**: The failures of FQI-LVD and VDN in cooperative matrix game in Table 1.
> >
> > This is because FQI-LVD and VDN use linear value decomposition, which has a limited function class and cannot exactly represent the joint value function in this didactic matrix game. Because of the uniform data distribution in the dataset, FQI-LVD and VDN will fit its local Q-values towards sub-optimal solutions, which choose actions A2 and A3. This limitation of VDN has been widely discussed by papers such as QTRAN and QPLEX, which motivated their works.
> >
> > **Q6**: A mistake in the review summary.
> >
> > The review summary stated that "The third result is that if the linear decomposition is replaced with consistency on argmax (likewise in QMIX). This result is more or less intuitive and obvious."
> >
> > However, the monotonic value decomposition of QMIX is not equivalent to $Q^{IGM}$ used by FQI-IGM. Please refer to Section 5.2 of the latest revision. This capacity gap is the major motivation of many advanced methods such as QPLEX, QTRAN, and MAVEN (Mahajan et al., 2019).
> >
> > [1] *Mahajan A, Rashid T, Samvelyan M, et al. Maven: Multi-agent variational exploration. Advances in Neural Information Processing Systems. 2019: 7613-7624.*

---

### Official Review · AnonReviewer3 · 2020-10-29
**A hard look at the latest MARL value function factorization methods**

**Rating:** 5
**Confidence:** 4

**Review:**

The paper takes a step back to examine some of the latest MARL methods in value function factorization by employing a classical framework, FQI. With some assumptions, the authors provide their analysis of the examined methods in the Bellman error minimization context and aligns them through a coherent unifying perspective.


Strengths:

+ The paper captures an important essence of lasting and meaningful research to deconstruct and provide some understanding of some of the latest MARL methods. In that aspect, the reviewer sees the need for more papers like this.

+ As with successful papers with a similar theoretical approach, this paper carries a well-targeted array of relevant works and aptly incorporates the study of each in the analysis.


Major Concerns:

- Even as a "first-step" analysis of related works, the two assumptions presented are, to my knowledge, unprecedentedly strong in the context of related works. It reads as though the assumptions are excessively strong to the point that the theoretical results that follow are natural corollaries. For example, if the transitions do not carry any stochasticity (as in Assumption 1), what need would there be for any non-linear value decomposition? In fact, if Assumption 2, also, further holds, then would there be a need for any value decomposition in the first place? The reviewer would very much like to be presented with explanation beyond "decomposition for the sake of decomposition". What kinds of insight can we gather from making a rather strong assumption and further deciding to carry out value decomposition?

- The connection made between FQI and the more recent works should be explained in more detail. In other words, Section 5 could be better written with comparative analyses of VDN, QMIX, QTRAN, etc. in the eyes of linear value decomposition. When the two assumptions hold, how do those works compare against each other? What do the comparative analysis results mean?

---

> ### Author Response · Authors · 2020-11-21
> **Response to Reviewer 3**
>
> Thanks for your comments. We have released the assumption of deterministic dynamics and explained the implication of Assumption 2, as discussed in the following Q\&A. We appreciate if you have any further questions or comments.
>
> **Q1**: The assumption of deterministic dynamics.
>
> Our theoretical results can generalize to stochastic environments. In the latest revision, we released the assumption of deterministic transitions by introducing Lemma 1 in Appendix A. Other derivations are not changed.
>
> **Q2**: The meaning and insights of value decomposition under the assumption of the factorizable dataset.
>
> The reviewer might misunderstand the implication and the generality of Assumption 2 about the factorizable dataset. This assumption does not require the factorization of the underlying transition and reward functions or the decomposability of the joint action-value function. On the contrary, all our theorems and examples focus on the situations where the joint Q-function cannot be perfectly factorized.
>
> Assumption 2 about the factorizable dataset can be satisfied when the dataset is collected by decentralized execution of agents' policies, e.g., an on-policy dataset collected using $\epsilon$-greedy exploration policies or an offline dataset collected by given decentralized policies of agents. As we know, VDN, QMIX, QTRAN, and QPLEX all learn decentralized policies, which are executed in a decentralized manner. Our theoretical implications are applicable whenever such factorizable data collection procedures are carried out.
>
> **Q3**: Section 5 could be better written with comparative analyses of VDN, QMIX, QTRAN, etc. in the eyes of linear value decomposition.
>
> Section 5 provides theoretical analyses on the learning stability of different value decomposition methods and also empirical evaluations on their deep-learning-based counterparts. VDN, QMIX, QTRAN, and QPLEX correspond to the deep-learning-based implementations of FQI with three function capacity hierarchies, i.e., $Q^{LVD} \subset Q^{QMIX} \subset Q^{IGM}$, which are linear, monotonic, and IGM value function classes, respectively. QTRAN realizes an approximate $Q^{IGM}$ and QPLEX realizes an exact one. Our empirical studies indicate that, the superior function capacity of QTRAN and QPLEX helps them achieve better learning stability over VDN and QMIX in the didactic MMDP example, which matches the theoretical implications of Proposition 2 and Theorem 3.
>
> In Section 5.2 of the latest revision, we included more detailed descriptions and discussions to compare these algorithms.
>
> **Q4**: When the two assumptions hold, how do those works compare against each other? What do the comparative analysis results mean?
>
> We released Assumption 1, as discussed in Q1, and explained the implication and generality of Assumption 2 in Q2. We conducted experiments, where Assumption 2 of the factorizable dataset holds, in Section 5.2 and Section 6.1. Empirical results of VDN, QTRAN, and QPLEX matches our theoretical analyses of their corresponding FQI variants. As suggested by Proposition 2 and Theorem 3, our empirical results show that VDN and QMIX suffer from the risk of unbounded divergence, while QPLEX and QTRAN perform outstanding numerical stability in these tasks. Also, as suggested by Theorem 2, VDN and QMIX with online data collection can perform much better than what they do in offline settings.
>
> These theoretical and empirical results can provide insights on why existing algorithms work and when they may not work, which can also indicate critical components for developing better practical algorithms.

---

### Author Response · Authors · 2020-11-23
**Summary of the Revision**

Thank you to all the reviewers for thoughtful comments, which helped us improve this work. Here is a brief summary of major updates made to the revision:

1. Released the assumption of deterministic transitions and generalized theoretical results to stochastic environments by introducing Lemma 1 in Appendix A. Other derivations are not changed.

2. Added an ablation study on the network capacity of VDN and QMIX in Appendix H. These experiments indicate that the limitations of VDN and QMIX on representational capacity cannot be addressed by simply increasing the number of neurons.

3. Clarified the generality of the assumption about factorized dataset and added experiments on an unfactorizable dataset in Appendix I. The results show that VDN and QMIX still suffer from unbounded divergence without the factorizability assumption. By contrast, QTRAN and QPLEX are not sensitive to the data distribution, which remarks the generality of Theorem 3 on the global convergence of FQI-IGM.

4. Added discussions in Section 5.2 to remark the difference between the monotonic value decomposition of QMIX and the complete IGM function class.

5. Added discussions about partial observability in Section 3.1.

6. Added discussions about related work on coordination graphs in Section 7.

We hope that our response and revision address your concerns and questions. We are happy to provide further clarification if you have any additional concerns or comments.

---

### Decision · Program_Chairs · 2021-01-07
**Final Decision**

**Decision:**

Reject

**Comment:**

This paper begins to formalize a connection between value decomposition and difference rewards. Whilst we are in agreement with the authors that papers do not need to make new algorithmic contribution and purely theoretical papers that deepen our understanding of established methods can be significant contributions, all reviewers had doubts on the maturity of the theoretical contribution of this paper.

Given the concerns raised by the authors for the attention of the area chair, I would like to reassure the authors that the majority of reviewers engaged in discussion after the rebuttal but remained unconvinced of the significance of the theoretical results. As these are representative of the potential audience at ICLR, it is clear further improvements to the motivation given in the paper and/or weakening of the assumptions within the theory are needed to engage the interest of the wider machine learning community.

The empirical studies in the paper also seem disconnected from the theoretical contribution and more like a continuation of the paper "Qplex: Duplex dueling multi-agent q-learning." Given the theoretical connection to difference rewards (e.g. COMA as explicitly noted by the authors in Implication 1) I would expect these methods to be included in the experiments to demonstrate how this theoretical connection affects performance in practical applications.